# Feature Normalization Prevents Collapse of Non-contrastive Learning Dynamics

## Abstract

Contrastive learning is a self-supervised representation learning framework, where two positive views generated through data augmentation are made similar by an attraction force in a data representation space, while a repulsive force makes them far from negative examples. Non-contrastive learning, represented by BYOL and SimSiam, further gets rid of negative examples and improves computational efficiency. While learned representations may collapse into a single point due to the lack of the repulsive force at first sight, [TCG21] revealed through the learning dynamics analysis that the representations can avoid collapse if data augmentation is sufficiently stronger than regularization. However, their analysis does not take into account commonly-used *feature normalization*, a normalizer before measuring the similarity of representations, and hence excessively strong regularization may collapse the dynamics, which is an unnatural behavior under the presence of feature normalization. Therefore, we extend the previous theory based on the L2 loss by considering the cosine loss, which involves feature normalization. We show that the cosine loss induces sixth-order dynamics (while the L2 loss induces a third-order one), in which a stable equilibrium dynamically emerges even if there are only collapsed solutions with given initial parameters. Thus, we offer a new understanding that feature normalization plays an important role in robustly preventing the dynamics collapse.

## 1 Introduction

Modern machine learning often owes to the success of self-supervised representation learning, which attempts to capture the underlying data structure useful for downstream tasks by solving an auxiliary learning task. Among self-supervised learning, contrastive learning is a popular framework, in which data augmentation generates two positive views from the original data and their encoded features are contrasted with background negative samples [CHL05, vdOLV18]. In particular, [CKNH20] conducted large-scale contrastive learning with 10K+ negative samples to establish comparable downstream classification performance even to supervised vision learners. The benefit of large-scale negative samples has been observed both theoretically [NS21, BNN22] and empirically [CH21, TBM$^+$22], but it is disadvantageous in terms of computational efficiency.

By contrast, non-contrastive learning trains a feature encoder with only positive views, leveraging additional implementation tricks. The seminal work [GSA$^+$20] proposed BYOL (Bootstrap Your Own Latent) to introduce the momentum encoder and apply gradient stopping for one encoder branch only. The follow-up work [CH21] showed that gradient stopping brings success into non-contrastive learning via a simplified architecture SimSiam (Simple Siamese representation learning). Despite their empirical successes, non-contrastive learning lacks the repulsive force induced by negative samples and learned representations may trivially collapse to a constant with only the attractive force between positive views. According to folklore, the success is attributed to asymmetric architectures between the two branches [WFT$^+$22]. [TCG21] first tackled the question *why non-contrastive learning does not collapse*, by specifically studying the learning dynamics of BYOL. They tracked the eigenmodes of the encoder parameters and found that the eigenmode dynamics have non-trivial equilibriums unless the regularization is overly strong. To put it differently, the balance between data augmentation and regularization controls the existence of non-trivial solutions. However, this analysis dismisses *feature normalization* practically added to normalize the encoded positive views before computing their similarity. As feature normalization blows up when encoded features approach zero, the analysis of [TCG21] may fail to explain the behavior of the non-contrastive learning dynamics with strong regularization. Indeed, our pilot study (Fig. 1) reveals that SimSiam learning dynamics remains to stabilize with much heavier regularization than the default strength $\rho = 10^{-4}$.

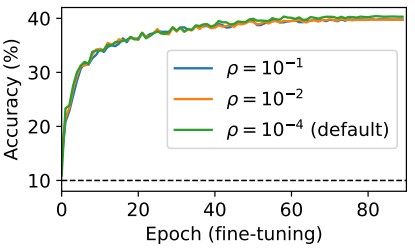

**Figure 1:** Linear probing accuracy of SimSiam representations of the CIFAR-10 dataset [Kri09] is indifferent to the weight decay intensity $\rho$. The vertical axis indicates fine-tuning epochs of the linear classifier. For non-contrastive pre-training, we used the ResNet-18 model [HZRS16] with the initial learning rate $5 \times 10^{-6}$, 500 epochs, and different weight decay intensities ($\rho$) indicated in the legends. Other parameters and setup were inherited from the official implementation [CH21].

Therefore, we study the non-contrastive learning dynamics with feature normalization: an encoded feature $\mathbf{\Phi x}$ for an input $\mathbf{x} \in \mathbb{R}^d$ and encoder $\mathbf{\Phi} \in \mathbb{R}^{h \times d}$ is normalized as $\mathbf{\Phi x}/ \|\mathbf{\Phi x}\|_2$. The main challenge is that the normalization yields a highly nonlinear dynamics because parameter norms appear in the denominator of a loss. This is a major reason why the existing studies on non-contrastive learning sticks to the L2-loss dynamics without the normalization [TCG21, WCDT21, PTLR22, WL22, LLUT23, TGR$^+$23]. Instead, we consider the high-dimensional limit $d, h \rightarrow \infty$, where the feature norm $\|\mathbf{\Phi x}\|_2$ concentrates around a constant with proper initialization. In this way, we can analyze the learning dynamics with feature normalization. Under the setup of synthetic data, we derive the learning dynamics of encoder parameters (Section 4), and disentangle it into the eigenmode dynamics with further assumptions (Section 5.1). The eigenmode dynamics is sixth-order, and we find that a stable equilibrium emerges even if there is no stable equilibrium with the initial parametrization and regularization strength (Section 5.2). This dynamics behavior is in contrast to the third-order dynamics of [TCG21], compared in Section 5.3. We further observe the above findings in numerical simulation (Section 5.4). Overall, we demonstrate how feature normalization prevents the collapse using a synthetic model. We believe that our techniques open a new direction to understanding self-supervised representation learning.

## 2 RELATED WORK

Recent advances in contrastive learning can be attributed to the InfoNCE loss [vdOLV18], which can be regarded as a multi-sample mutual information estimator between the two views [POvdO$^+$19, SE20]. [CKNH20] showed that large-scale contrastive representation learning can potentially perform comparably to supervised vision learners. This empirical success owes to a huge number of negative samples, forming a repulsive force in contrastive learning. Follow-up studies confirmed that larger negative samples are generally beneficial for downstream performance [CH21, TBM$^+$22], and the phenomenon has been verified through theoretical analysis of the downstream classification error [NS21, WZW$^+$22, BNN22, ADK22], whereas larger negative samples require heavier computation.

Non-contrastive learning is yet another stream of contrastive learning, without requiring any negative samples. Although it may fail due to lack of the repulsive force, additional tricks in architectures assist the learned representation avoiding a trivial solution. BYOL [GSA$^+$20] is the initial attempt by introducing the momentum encoder and gradient stopping to make two encoder branches asymmetric. Later, SimSiam [CH21] revealed that gradient stopping is dominant. Both BYOL and SimSiam emphasize the importance of asymmetric architectures. Other recent approaches to non-contrastive learning are to conduct representation learning and clustering iteratively (e.g., SwAV [CMM$^+$20] and TCR [LCLS22]), to impose regularization on the representation covariance matrix (e.g., Barlow Twins [ZJM$^+$21], Whitening MSE [ESSS21], and VICReg [BPL22]), and to leverage knowledge distillation (e.g., DINO [CTM$^+$21]). While these methods empirically succeed, theoretical understanding of the mechanism of non-contrastive learning still falls behind. In particular, we need to answer *why* the non-contrastive dynamics does not collapse without the repulsive force, and *what* the non-contrastive dynamics learns. For the latter question, recent studies revealed that it implicitly learns a subspace [WCDT21], sparse signals [WL21], a permutation matrix over latent variables [PTLR22], and a low-pass filter of parameter spectra [ZWMW23]. Besides, contrastive supervision is theoretically useful for downstream classification under a simplified setup [BNS18, BSX$^+$22].

Why does non-contrastive dynamics remain stable? The seminal work [TCG21] analyzed the BYOL/SimSiam dynamics with a two-layer network and found that data augmentation behaves as a repulsive force to prevent eigenmodes of network parameters from collapsing if augmentation is sufficiently stronger than regularization. We closely follow this analysis to delineate that feature normalization serves as another repulsive force and regularization may not destroy the dynamics. Our

focus is to understand how a non-trivial equilibrium emerges in self-supervised learning dynamics, whereas several prior studies revealed the importance of normalization for supervised learning by investigating when and how fast general gradient descent dynamics with weight normalization converges [DGM20, WZZS21] and how normalization prevents rank collapse of nonlinear MLPs at the infinite-depth limit via isometry [JDB23]. Further, [WL22] analyzed the SimSiam dynamics with a trainable prediction head to reveal the conditions preventing representation collapse. [TGR$^+$23] investigated the same phenomenon in a reinforcement learning setup. While we have less understanding of other non-contrastive dynamics, [LLUT23] showed that some non-contrastive dynamics including VICReg may cause dimensional collapse. Notably, a concurrent work [HLZ23] studied implicit bias of non-contrastive learning with the cosine loss and showed that the *non-zero* eigenmodes converges closely to each other, whereas how the complete collapse is avoided remains unclear.

## 3   MODEL AND LOSS FUNCTIONS

**Notations.**   The $n$-dimensional Euclidean space and hypersphere are denoted by $\mathbb{R}^n$ and $\mathbb{S}^{n-1}$, respectively. The L2, Frobenius, and spectral norms are denoted by $\|\cdot\|_2$, $\|\cdot\|_F$, and $\|\cdot\|$, respectively. The $n \times n$ identity matrix is denoted by $\mathbf{I}_n$, or by $\mathbf{I}$ whenever clear from the context. For two vectors $\mathbf{u}, \mathbf{v} \in \mathbb{R}^n$, $\langle \mathbf{u}, \mathbf{v} \rangle = \mathbf{u}^\top \mathbf{v}$ denotes the inner product. For two matrices $\mathbf{A}, \mathbf{B} \in \mathbb{R}^{n_1 \times n_2}$, $\langle \mathbf{A}, \mathbf{B} \rangle_F = \sum_{i,j} A_{i,j} B_{i,j}$ denotes the Frobenius inner product. For a time-dependent matrix $\mathbf{A}$ (such as network parameters), we make the time dependency explicitly by $\mathbf{A}(t)$ if necessary. The Moore–Penrose inverse of a matrix $\mathbf{A}$ is denoted by $\mathbf{A}^\dagger$. The set of $n \times n$ symmetric matrices is denoted by $\mathbb{Sym}_n := \left\{ \mathbf{A} \in \mathbb{R}^{n \times n} | \mathbf{A} = \mathbf{A}^\top \right\}$. The upper and lower asymptotic orders are denoted by $\mathcal{O}(\cdot)$ and $\Omega(\cdot)$, respectively. The little-o and little-$\omega$ are denoted in the same way. The stochastic orders of boundedness and convergence indexed by $h$ are denoted by $\mathcal{O}_\mathbb{P}(\cdot)$ and $o_\mathbb{P}(\cdot)$, respectively.

**Model.**   In this work, we focus on the SimSiam model [CH21] as a non-contrastive learner and consider the following two-layer linear network, following the analysis of [TCG21]. We first sample a $d$-dimensional input feature $\mathbf{x}_0 \sim \mathcal{D}$ as an anchor and apply a data augmentation to obtain two views $\mathbf{x}, \mathbf{x}' \sim \mathcal{D}^{\text{aug}}_{\mathbf{x}_0}$, where $\mathcal{D}^{\text{aug}}_{\mathbf{x}_0}$ is the augmentation distribution. While affine transforms or random maskings of input images are common as data augmentation [CKNH20, HCX$^+$22], we assume the isotropic Gaussian augmentation distribution $\mathcal{D}^{\text{aug}}_{\mathbf{x}_0} = \mathcal{N}(\mathbf{x}_0, \sigma^2 \mathbf{I})$ to simplify and let $\sigma^2$ represent the augmentation intensity. For the input distribution, we suppose the multivariate Gaussian $\mathcal{D} = \mathcal{N}(\mathbf{0}, \boldsymbol{\Sigma})$ to devote ourselves to understanding dynamics, as in [SMG14, TCG21].

Our neural network encoder consists of two linear layers without biases: representation net $\boldsymbol{\Phi} \in \mathbb{R}^{h \times d}$ and projection head $\mathbf{W} \in \mathbb{R}^{h \times h}$ as the first and second layers, respectively, where $h$ is the representation dimension. For the two views $\mathbf{x}, \mathbf{x}'$, we obtain *online* representation $\boldsymbol{\Phi}\mathbf{x} \in \mathbb{R}^h$ and *target* representation $\boldsymbol{\Phi}\mathbf{x}' \in \mathbb{R}^h$, and predict the target from the online representation by $\mathbf{W}\boldsymbol{\Phi}\mathbf{x} \in \mathbb{R}^h$. Here, we use the same representation parameters $\boldsymbol{\Phi}$ for both views without the exponential moving average [GSA$^+$20] as this ablation reportedly works comparably in SimSiam [CH21].

**Loss functions.**   BYOL/SimSiam introduce *asymmetry* of the two branches with the stop gradient operator, denoted by $\text{StopGrad}(\cdot)$, where parameters are regarded as constants during backpropagation [CH21]. [TCG21] used the following *L2 loss* to describe non-contrastive dynamics:

$$\mathcal{L}_{\text{sq}}(\boldsymbol{\Phi}, \mathbf{W}) := \frac{1}{2} \mathbb{E}_{\mathbf{x}_0} \mathbb{E}_{\mathbf{x}, \mathbf{x}' | \mathbf{x}_0} [\| \mathbf{W}\boldsymbol{\Phi}\mathbf{x} - \text{StopGrad}(\boldsymbol{\Phi}\mathbf{x}') \|_2^2], \tag{1}$$

where the expectations are taken over $\mathbf{x}, \mathbf{x}' \sim \mathcal{D}^{\text{aug}}_{\mathbf{x}_0}$ and $\mathbf{x}_0 \sim \mathcal{D}$. Thanks to the simple closed-form solution, the L2 loss has been used in most of the existing analyses of self-supervised learning dynamics [WCDT21, TGR$^+$23, ZWMW23].

We instead focus on the following *cosine loss* to take feature normalization into account, which is a key factor in the success of contrastive representation learning [WI20]:

$$\mathcal{L}_{\text{cos}}(\boldsymbol{\Phi}, \mathbf{W}) := \mathbb{E}_{\mathbf{x}_0} \mathbb{E}_{\mathbf{x}, \mathbf{x}' | \mathbf{x}_0} \left[ -\frac{\langle \mathbf{W}\boldsymbol{\Phi}\mathbf{x}, \text{StopGrad}(\boldsymbol{\Phi}\mathbf{x}') \rangle}{\|\mathbf{W}\boldsymbol{\Phi}\mathbf{x}\|_2 \|\text{StopGrad}(\boldsymbol{\Phi}\mathbf{x}')\|_2} \right]. \tag{2}$$

Importantly, the cosine loss has been used in most practical implementations [GSA$^+$20, CH21], including a reproductive research [HMW22] of simulations in [TCG21]. Subsequently, the weight decay $R(\boldsymbol{\Phi}, \mathbf{W}) := \frac{\rho}{2}(\|\boldsymbol{\Phi}\|_F^2 + \|\mathbf{W}\|_F^2)$ is added with a regularization strength $\rho > 0$.

## 4 NON-CONTRASTIVE DYNAMICS IN THERMODYNAMICAL LIMIT

Let us focus on the cosine loss and derive its non-contrastive dynamics via the gradient flow. See Appendix B for the proofs of lemmas provided subsequently. As the continuous limit of the gradient descent where learning rates are taken to be infinitesimal [SMG14], we characterize time evolution of the network parameters by the following simultaneous ordinal differential equation:

$$\dot{\boldsymbol{\Phi}} = -\nabla_{\boldsymbol{\Phi}}\{\mathcal{L}_{\cos}(\boldsymbol{\Phi}, \mathbf{W}) + R(\boldsymbol{\Phi}, \mathbf{W})\}, \quad \dot{\mathbf{W}} = -\nabla_{\mathbf{W}}\{\mathcal{L}_{\cos}(\boldsymbol{\Phi}, \mathbf{W}) + R(\boldsymbol{\Phi}, \mathbf{W})\}. \quad (3)$$

To derive the dynamics, several assumptions are imposed.

**Assumption 1** (Symmetric projection). *$\mathbf{W} \in \mathbb{Sym}_h$ holds during time evolution.*

**Assumption 2** (Input distribution). *$\boldsymbol{\Sigma} = \mathbf{I}$, namely, $\mathcal{D} = \mathcal{N}(\mathbf{0}, \mathbf{I})$.*

**Assumption 3** (Thermodynamical limit). *$d, h \to \infty$, and $d/h \to \alpha$ for some $\alpha \in (0, 1)$.*

**Assumption 4** (Parameter initialization). *$\boldsymbol{\Phi}$ is initialized with $\sqrt{d} \cdot \boldsymbol{\Phi}(0)_{ij} \sim \mathcal{N}(0, 1)$ for $i \in [h], j \in [d]$. $\mathbf{W}$ is initialized with $\sqrt{h} \cdot \mathbf{W}(0)_{ij} \sim \mathcal{N}(0, 1)$ for $i, j \in [h]$.*

Assumptions 1 and 2 are borrowed from [TCG21] and simplify subsequent analyses. We empirically verify that the non-contrastive dynamics maintains the symmetry of $\mathbf{W}$ during the training later (Section 5.4). Assumption 3 is a cornerstone to our analysis: the high-dimensional limit makes Gaussian random vectors concentrate on a sphere, which leads to a closed-form solution for the cosine loss dynamics. We suppose that the common hidden unit size $h = 512$ (used in SimSiam) is sufficient to bring into the high-dimensional limit—though the high-dimensional regime of representations would be arguable with the low-dimensional manifold assumption being in one's mind. Assumption 4 is a standard initialization scale empirically in the He initialization [HZRS15] and theoretically in the neural tangent kernel regime [JGH18]. This initialization scale maintains norms of the random matrices $\boldsymbol{\Phi}$ and $\mathbf{W}\boldsymbol{\Phi}$ without vanishing or exploding under the thermodynamical limit.

**Lemma 1.** *Parameter matrices $\mathbf{W}$ and $\boldsymbol{\Phi}$ evolve as follows:*

$$\mathbf{W}^\top \dot{\mathbf{W}} = \mathbf{H} - \rho \mathbf{W}\mathbf{W}^\top,$$
$$\dot{\boldsymbol{\Phi}}\boldsymbol{\Phi}^\top \mathbf{W}^\top = \mathbf{W}^\top \mathbf{H} - \rho \boldsymbol{\Phi}\boldsymbol{\Phi}^\top \mathbf{W}^\top, \quad (4)$$

*where $\mathbf{H} := \mathbb{E}[\mathbf{z}'\boldsymbol{\omega}^\top - (\boldsymbol{\omega}^\top \mathbf{z}')\boldsymbol{\omega}\boldsymbol{\omega}^\top]$, $\mathbf{z} := \boldsymbol{\Phi}\mathbf{x}'/\|\boldsymbol{\Phi}\mathbf{x}'\|_2$, and $\boldsymbol{\omega} := \mathbf{W}\boldsymbol{\Phi}\mathbf{x}/\|\mathbf{W}\boldsymbol{\Phi}\mathbf{x}\|_2$. The expectation in $\mathbf{H}$ is taken over $\mathbf{x}_0$, $\mathbf{x}$, and $\mathbf{x}'$.*

We will analyze Eq. (4) to see when the dynamics stably converges to a non-trivial solution. To solve it, we need to evaluate $\mathbf{H}$ first. This involves expectations with $\mathbf{z}'$ and $\boldsymbol{\omega}$, which are normalized Gaussian vectors and cannot be straightforwardly evaluated. Here, we take a step further by considering the thermodynamical limit (Assumption 3), where norms of Gaussian vectors are concentrated. This regime allows us to directly evaluate Gaussian random vectors instead of the normalized ones.

**Lemma 2.** *Under Assumptions 1 to 4, for a fixed $\mathbf{x}_0$, the norms of $\boldsymbol{\Phi}\mathbf{x}$ and $\mathbf{W}\boldsymbol{\Phi}\mathbf{x}$ (as well as $\boldsymbol{\Phi}\mathbf{x}'$ and $\mathbf{W}\boldsymbol{\Phi}\mathbf{x}'$) are concentrated:*

$$\left\|\tfrac{1}{\sqrt{h\sigma^2}}\boldsymbol{\Phi}\mathbf{x}\right\|_2^2 = \left\|\tfrac{1}{\sqrt{h}}\boldsymbol{\Phi}\right\|_F^2 + \left\|\tfrac{1}{\sqrt{h\sigma^2}}\boldsymbol{\Phi}\mathbf{x}_0\right\|_2^2 + o_{\mathbb{P}}(1),$$

$$\left\|\tfrac{1}{\sqrt{h^2\sigma^2}}\mathbf{W}\boldsymbol{\Phi}\mathbf{x}\right\|_2^2 = \left\|\tfrac{1}{\sqrt{h^2}}\mathbf{W}\boldsymbol{\Phi}\right\|_F^2 + \left\|\tfrac{1}{\sqrt{h^2\sigma^2}}\mathbf{W}\boldsymbol{\Phi}\mathbf{x}_0\right\|_2^2 + o_{\mathbb{P}}(1).$$

**Lemma 3.** *Under Assumptions 1 to 4, the following concentrations are established:*

$$\left\|\tfrac{1}{\sqrt{h\sigma^2}}\boldsymbol{\Phi}\mathbf{x}_0\right\|_2 = \left\|\tfrac{1}{\sqrt{h\sigma^2}}\boldsymbol{\Phi}\right\|_F + o_{\mathbb{P}}(1), \qquad \left\|\tfrac{1}{\sqrt{h^2\sigma^2}}\mathbf{W}\boldsymbol{\Phi}\mathbf{x}_0\right\|_2 = \left\|\tfrac{1}{\sqrt{h^2\sigma^2}}\mathbf{W}\boldsymbol{\Phi}\mathbf{x}_0\right\|_F + o_{\mathbb{P}}(1).$$

Lemmas 2 and 3 are based on the *Hanson–Wright inequality* [Ver18, Theorem 6.3.2], a concentration inequality for order-2 Gaussian chaos, with an additional effort to control norms of random matrices $\mathbf{W}$ and $\boldsymbol{\Phi}$. By combining Lemmas 2 and 3, we can express normalizers $\|\boldsymbol{\Phi}\mathbf{x}'\|_2^{-1}$ and $\|\mathbf{W}\boldsymbol{\Phi}\mathbf{x}\|_2^{-1}$ in $\mathbf{H}$ into simpler forms, and obtain a concise expression of $\mathbf{H}$ consequently.

**Lemma 4.** *Let $\boldsymbol{\Psi} := \mathbf{W}\boldsymbol{\Phi}$. Assume that $\|\boldsymbol{\Phi}\|_F$ and $\|\boldsymbol{\Psi}\|_F$ are bounded away from zero. Under Assumptions 1 to 4, $\mathbf{H}$ can be expressed as follows:*

$$\mathbf{H} = \frac{1}{1+\sigma^2}\left\{\tilde{\boldsymbol{\Phi}}\tilde{\boldsymbol{\Psi}}^\top - 2\tilde{\boldsymbol{\Psi}}\tilde{\boldsymbol{\Phi}}^\top\tilde{\boldsymbol{\Psi}}\tilde{\boldsymbol{\Psi}}^\top - \operatorname{tr}(\tilde{\boldsymbol{\Phi}}^\top\tilde{\boldsymbol{\Psi}})\tilde{\boldsymbol{\Psi}}\tilde{\boldsymbol{\Psi}}^\top\right\} + o_{\mathbb{P}}(1),$$

*where $\tilde{\boldsymbol{\Phi}} := \boldsymbol{\Phi}/\|\boldsymbol{\Phi}\|_F$ and $\tilde{\boldsymbol{\Psi}} := \boldsymbol{\Psi}/\|\boldsymbol{\Psi}\|_F$.*

## 5 ANALYSIS OF NON-CONTRASTIVE DYNAMICS

To analyze the dynamics (4), the main obstacle is the normalizers $\|\mathbf{\Phi}\|_{\mathrm{F}}^{-1}$ and $\|\mathbf{\Psi}\|_{\mathrm{F}}^{-1}$ in $\mathbf{H}$, which makes the dynamics highly nonlinear and challenging to solve directly. Instead, we consider the equilibrium state $\|\mathbf{\Phi}\|_{\mathrm{F}} \to N_\Phi$ and $\|\mathbf{\Psi}\|_{\mathrm{F}} \to N_\Psi$ with $N_\Phi, N_\Psi > 0$. This allows us to focus on the parameter values $\mathbf{W}$ and $\mathbf{\Phi}$ at equilibrium. We impose the next assumption.

**Assumption 5** (Norms remain stable). $\|\mathbf{\Phi}\|_{\mathrm{F}} \equiv N_\Phi$, $\|\mathbf{\Psi}\|_{\mathrm{F}} \equiv N_\Psi$, and $\mathrm{tr}(\tilde{\mathbf{\Phi}}^\top \tilde{\mathbf{\Psi}}) \equiv N_\times$ for *sufficiently long time.*

In Section 5.4, we will numerically see that the three quantities may not drastically change during time evolution. Indeed, learning dynamics analysis of weight normalization often posits a similar one so that parameter norms remain the same globally [vL17]. We conjecture that this assumption can be replaced with the local stability as in the previous convergence analysis of weight-norm dynamics [WZZS21]; nevertheless, we choose to assume the global stability for simplicity to concentrate on the equilibrium analysis. Under Assumption 5, $\mathbf{H}$ can be expressed as follows:

$$\mathbf{H} = \frac{1}{1+\sigma^2}\left(\frac{\mathbf{FW}}{N_\Phi N_\Psi} - \frac{2\mathbf{WFWFW}}{N_\Phi N_\Psi^3} - \frac{N_\times \mathbf{WFW}}{N_\Phi N_\Psi}\right)(=: \hat{\mathbf{H}}), \tag{5}$$

where $\mathbf{F} := \mathbf{\Phi\Phi}^\top$ and we drop the negligible term $o_{\mathbb{P}}(1)$ for simplicity.

### 5.1 EIGENMODE DECOMPOSITION OF DYNAMICS

To analyze the stability of the dynamics (4), we disentangle it into the eigenmodes. We first show the condition where the eigenspaces of $\mathbf{W}$ and $\mathbf{F}$ align with each other. Note that two commuting matrices can be simultaneously diagonalized.

**Proposition 1.** *Suppose $\mathbf{W}$ is non-singular. Under the dynamics* (4) *with $\mathbf{H} = \hat{\mathbf{H}}$, the commutator $\mathbf{L}(t) := [\mathbf{F}, \mathbf{W}] := \mathbf{FW} - \mathbf{WF}$ satisfies $\frac{\mathrm{dvec}(\mathbf{L}(t))}{\mathrm{d}t} = -\mathbf{K}(t)\mathrm{vec}(\mathbf{L}(t))$, where*

$$\mathbf{K}(t) := 2\frac{\mathbf{W} \oplus \mathbf{WFW} + \mathbf{W}^2(\mathbf{FW} \oplus \mathbf{I}_d)}{(1+\sigma^2)N_\Phi N_\Psi^3} + \frac{(\mathbf{W}^{-1}) \oplus \mathbf{F} - (\mathbf{W} - N_\times \mathbf{W}^2) \oplus \mathbf{I}_d}{(1+\sigma^2)N_\Phi N_\Psi} + 3\rho\mathbf{I}_d,$$

*and $\mathbf{A} \oplus \mathbf{B} := \mathbf{A} \otimes \mathbf{B} + \mathbf{B} \otimes \mathbf{A}$ denotes the sum of the two Kronecker products.*

*If $\inf_{t\geq 0} \lambda_{\min}(\mathbf{K}(t)) \geq \lambda_0 > 0$ for some $\lambda_0 > 0$, then $\|\mathbf{L}(t)\|_{\mathrm{F}} \to 0$ as $t \to \infty$.*

Proposition 1 is a variant of [TCG21, Theorem 3] for the dynamics (4). Consequently, we see that $\mathbf{W}$ and $\mathbf{F}$ are simultaneously diagonalizable at the equilibrium $\|\mathbf{L}(t)\|_{\mathrm{F}} = \|[\mathbf{F}, \mathbf{W}]\|_{\mathrm{F}} = 0$. We then approximately deal with the dynamics (4).

**Assumption 6** (Always commutative). $\|[\mathbf{F}, \mathbf{W}]\|_{\mathrm{F}} \equiv 0$ *for $\forall t \geq 0$.*

We verify the validity of the assumption in Section 5.4, where we see that the commutator remains to be nearly zero.

Let $\mathbf{U}$ be the common eigenvectors of $\mathbf{F}$ and $\mathbf{W}$, then $\mathbf{W} = \mathbf{U}\mathbf{\Lambda}_W \mathbf{U}^\top$ and $\mathbf{F} = \mathbf{U}\mathbf{\Lambda}_F \mathbf{U}^\top$, where $\mathbf{\Lambda}_W = \mathrm{diag}[p_1, p_2, \ldots, p_d]$ and $\mathbf{\Lambda}_F = \mathrm{diag}[s_1, s_2, \ldots, s_d]$. By extending the discussion of [TCG21, Appendix B.1], we can show that $\mathbf{U}$ would not change over time.

**Proposition 2.** *Suppose $\mathbf{W}$ is non-singular. Under the dynamics* (4) *with $\mathbf{H} = \hat{\mathbf{H}}$, we have $\dot{\mathbf{U}} = \mathbf{O}$.*

With Assumptions 5 and 6 and Proposition 2, we decompose (4) with $\mathbf{H} = \hat{\mathbf{H}}$ into the eigenmodes.

$$\dot{p}_j = -\frac{1}{(1+\sigma^2)N_\Phi N_\Psi}\left(\frac{2}{N_\Psi^2}s_j^2 p_j^2 + N_\times s_j p_j - s_j\right) - \rho p_j,$$

$$\dot{s}_j = -\frac{2}{(1+\sigma^2)N_\Phi N_\Psi}\left(\frac{2}{N_\Psi^2}s_j^2 p_j^3 + N_\times s_j p_j^2 - s_j p_j\right) - 2\rho s_j. \tag{6}$$

The eigenmode dynamics (6) is far more interpretable than the matrix dynamics (4) and amenable to further understanding. Subsequently, we analyze the eigenmode dynamics to investigate the number of equilibrium points and their stability.

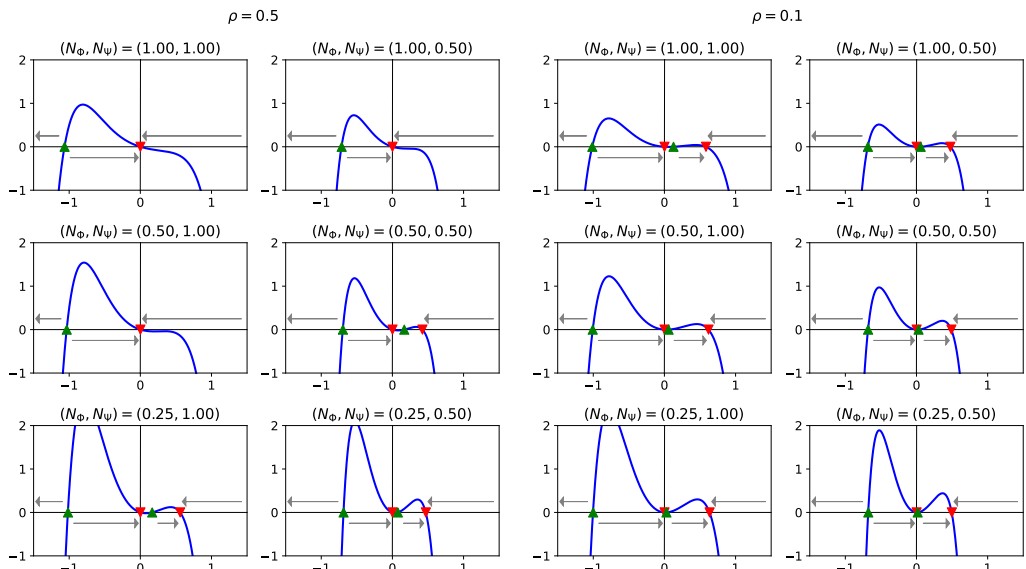

**Figure 2:** Numerical illustrations of the dynamics Eq. (8) with different values of $(\rho, N_\Phi, N_\Psi)$, where vertical and horizontal axes denote $p_j$ and $\dot{p}_j$, respectively. The left two columns are illustrated for $\rho = 0.5$, while right two columns for $\rho = 0.1$. Red ▼ and green ▲ indicate stable (namely, $\dot{p}_j < 0$) and unstable equilibrium (namely, $\dot{p}_j > 0$) points, respectively [HSD12]. For other parameters, we chose $N_\times = 1$ and $\sigma^2 = 0.1$ for illustration.

## 5.2 EQUILIBRIUM ANALYSIS OF EIGENMODE DYNAMICS

We are interested in how the eigenmode avoids collapse with feature normalization. For this purpose, we investigate the equilibrium points of the eigenmode dynamics (6).

**Invariant parabola.** By simple algebra, $\dot{s}_j - 2p_j\dot{p}_j = -2\rho(s_j - p_j^2)$. Noting that $\frac{d}{dt}(s_j - p_j^2) = \dot{s}_j - 2p_j\dot{p}_j$ and integrating both ends, we encounter the following relation:

$$s_j(t) = p_j^2(t) + c_j \exp(-2\rho t), \tag{7}$$

where $c_j \coloneqq s_j(0) - p_j^2(0)$ is the initial condition. Equation (7) elucidates that the dynamics of $(p_j(t), s_j(t))$ asymptotically converges to the parabola $s_j(t) = p_j^2(t)$ as $t \to \infty$ when regularization $\rho > 0$ exists. The information of initialization $c_j$ shall be forgotten. Stronger regularization yields faster convergence to the parabola. We reasonably expect that this exponential convergence is much faster than the drifts of $\|\Phi\|_F$, $\|\Psi\|_F$, and $\mathrm{tr}(\tilde{\Phi}^\top \Psi)$ so that Assumption 5 holds.

**Dynamics on invariant parabola.** We now focus on the dynamics on the invariant parabola. Substituting $s_j(t) = p_j^2(t)$ into $p_j$-dynamics in Eq. (6) yields the following dynamics:

$$\dot{p}_j = -\frac{2}{(1+\sigma^2)N_\Phi N_\Psi^3}p_j^6 - \frac{N_\times}{(1+\sigma^2)N_\Phi N_\Psi}p_j^3 + \frac{1}{(1+\sigma^2)N_\Phi N_\Psi}p_j^2 - \rho p_j. \tag{8}$$

We illustrate the dynamics (8) with different parameter values in Fig. 2. This dynamics always has $p_j = 0$ as an equilibrium point, and the number of equilibrium points varies between two and four. Notably, Eq. (8) is a *sixth-order* non-linear ODE (in $p_j$), whereas the L2 loss dynamics [TCG21, Eq. (16)] induces a *third-order* non-linear eigenmode dynamics, as we will recap in Section 5.3. From Fig. 2, we can classify into three regimes (refer to Fig. 3 together; more formally, confer Appendix C):

- **(Collapse)** When all of $\rho$, $N_\Phi$, $N_\Psi$ are large, the dynamics only has two equilibrium points. See the plots with $(\rho, N_\Phi, N_\Psi) \in \{(0.5, 1.0, 1.0), (0.5, 1.0, 0.5), (0.5, 0.5, 1.0)\}$. In this regime, $p_j = 0$ is the only stable equilibrium, causing the collapsed dynamics. This regime is brittle because the stable equilibrium $p_j = 0$ blows up the normalizers $\|\Phi\|_F^{-1}$ and $\|\Psi\|_F^{-1}$ in the original cosine loss dynamics. As $p_j$ shrinks, the values $N_\Phi$ and $N_\Psi$ shrink together, too, which brings the dynamics into the next two regimes.

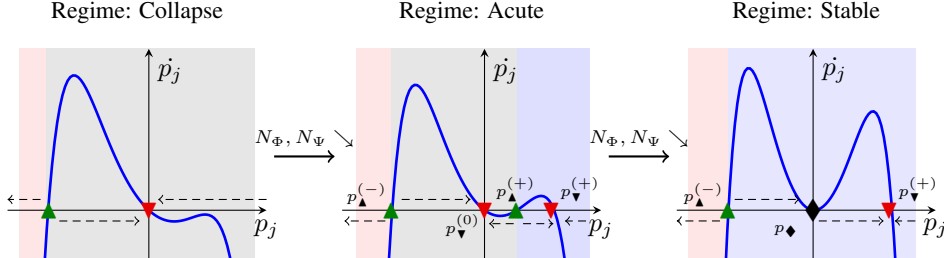

**Figure 3:** Schema of Collapse, Acute, and Stable regimes of the eigenmode dynamics Eq. (8). Red ▼ and green ▲ indicate stable (namely, $\dot{p}_j < 0$) and unstable equilibrium (namely, $\dot{p}_j > 0$) points, respectively. The black ♦ denotes the saddle point. Red , gray , and blue backgrounds indicate ranges where the eigenmode will diverge to $-\infty$, collapse to 0, and converge to the stable equilibrium, respectively. As $N_\Phi$ and $N_\Psi$ become smaller, the regime shifts in the direction «Collapse → Acute → Stable», and as $N_\Phi$ and $N_\Psi$ become larger, the regime shifts in the opposite direction «Stable → Acute → Collapse».

- **(Acute)** When $\rho$, $N_\Phi$, and $N_\Psi$ become smaller than those in Collapse, two new equilibrium points emerge and the number of equilibrium points is four in total. See the plots with $(\rho, N_\Phi, N_\Psi) \in \{(0.5, 0.5, 0.5), (0.5, 0.25, 1.0), (0.1, 1.0, 1.0)\}$. Let $p_\blacktriangle^{(-)}$, $p_\blacktriangledown^{(0)}(= 0)$, $p_\blacktriangle^{(+)}$, and $p_\blacktriangledown^{(+)}$ denote the equilibrium points from smaller to larger ones, respectively, namely, $p_\blacktriangle^{(-)} < p_\blacktriangledown^{(0)} = 0 < p_\blacktriangle^{(+)} < p_\blacktriangledown^{(+)}$ (see Fig. 3). Note that $p_j = p_\blacktriangle^{(-)}, p_\blacktriangle^{(+)}$ are unstable and $p_j = p_\blacktriangledown^{(0)}, p_\blacktriangledown^{(+)}$ are stable [HSD12]. In this regime, the eigenmode initialized larger than $p_\blacktriangle^{(+)}$ converge to non-degenerate point $p_\blacktriangledown^{(+)}$. However, the eigenmode degenerates to $p_\blacktriangledown^{(0)}$ if initialization is in the range $[p_\blacktriangle^{(-)}, p_\blacktriangle^{(+)}]$ (close to zero), and diverges if initialization has large negative value $< p_\blacktriangle^{(-)}$. If the eigenmode degenerates, the values $N_\Phi$ and $N_\Psi$ further shrink and then the regime enters the final one; if the eigenmode diverges, $N_\Phi$ and $N_\Psi$ inflate and the regime goes back to the previous Collapse.

- **(Stable)** When $\rho$, $N_\Phi$, and $N_\Psi$ are further smaller than those in Acute, the middle two equilibrium points $p_\blacktriangledown^{(0)}$ and $p_\blacktriangle^{(+)}$ approaches and form a saddle point. See the plots with $(\rho, N_\Phi, N_\Psi) \in \{(0.5, 0.25, 0.5), (0.1, 0.25, 1.0), (0.1, 0.25, 0.5)\}$. Denote this saddle point by $p_\blacklozenge$. The dynamics has a unstable equilibrium $p_\blacktriangle^{(-)}$, a saddle point $p_\blacklozenge$, and a stable equilibrium $p_\blacktriangledown^{(+)}$, from smaller to larger ones. In this regime, the eigenmode stably converges to the non-degenerate point $p_j = p_\blacktriangledown^{(+)}$ unless the initialization is smaller than $p_\blacktriangle^{(-)}$.

  (*Remark*: $p_\blacktriangledown^{(0)} = p_\blacktriangle^{(+)}$ never occurs because the dynamics diverges as $N_\Phi, N_\Psi \to 0$. Nonetheless, this approximately occurs with realistic parameters such as $(\rho, N_\Phi, N_\Psi) = (0.1, 0.25, 0.5)$.)

**Three regimes prevent degeneration.** We illustrate the relationship among the three regimes in Fig. 3. As we see in the numerical experiments (Section 5.4), the parameter initialization (Assumption 4) hardly makes the initial eigenmode smaller than $p_\blacktriangle^{(-)}$: indeed, we simulated the initial eigenmode distributions in Fig. 4, which indicates that the eigenmodes are sufficiently larger than $p_\blacktriangle^{(-)}$. Therefore, the learning dynamics has stable equilibriums and successfully stabilizes.

Importantly, this cosine loss dynamics stabilizes and would not collapse to zero regardless of the regularization strength $\rho$, which is in stark contrast to the L2 loss dynamics, as detailed in Section 5.3. This observation tells us the importance of feature normalization to prevent representation collapse in non-contrastive self-supervised learning.

## 5.3 COMPARISON WITH L2 LOSS DYNAMICS

Whereas we mainly focused on the study of the cosine loss dynamics, [TCG21] (and many earlier studies) engaged in the L2 loss dynamics, which does not entail feature normalization. Here, we compare the cosine and L2 loss dynamics to see how feature normalization plays a crucial role.

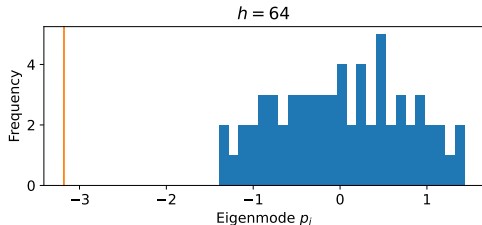 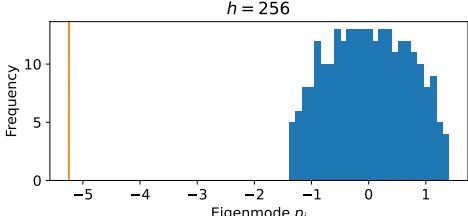

**Figure 4:** Numerical simulation of eigenvalue distributions of $\mathbf{W}$. In each figure, we generate $\mathbf{W}$ and $\mathbf{\Phi}$ by the initialization of Assumption 4, and illustrate the histogram of eigenmodes of $\mathbf{W}$. The vertical line indicates the value of $p_{\blacktriangle}^{(-)}$, the negative unstable equilibrium point of $p_j$-dynamics (8), computed by the binary search and numerical root finding. For parameters, we chose $\rho = 0.05$, $\sigma^2 = 1.0$, $d = 2048$, and $h \in \{64, 256\}$.

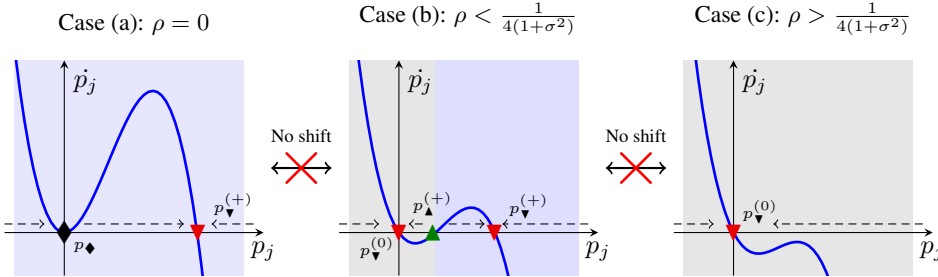

**Figure 5:** Schema of three eigenmode dynamics in the L2 loss case. Each figure illustrates the eigenmode corresponding fixed regularization strength $\rho$. The meaning of each mark ($\blacktriangle$, $\blacktriangledown$, $\blacklozenge$) and background colors can be found in the caption of Fig. 3. The figure borrows the illustration of [TCG21, Figure 4].

Let us review the dynamics of [TCG21]. We inherit Assumption 1 (symmetric projector), Assumption 2 (standard normal input), and Assumption 6 ($\mathbf{F}$ and $\mathbf{W}$ are commutative). Under this setup, [TCG21] analyzed the non-contrastive dynamics (4) with the L2 loss (1), and revealed that the eigenmodes of $\mathbf{W}$ and $\mathbf{F}$ (denoted by $p_j$ and $s_j$, respectively) asymptotically converges to the invariant parabola $s_j(t) = p_j^2(t)$ (see Eq. (7)), where the $p_j$-dynamics reads:

$$\dot{p}_j = p_j^2\{1 - (1 + \sigma^2)p_j\} - \rho p_j. \tag{9}$$

Compare the L2-loss dynamics (9) (third-order) and the cosine-loss dynamics (8) (sixth-order). Note that we omit the exponential moving average of the online representation in BYOL ($\tau = 1$) and use the same learning rate for the predictor and online nets ($\alpha = 1$) in [TCG21] for comparison.

The behaviors of the two dynamics are compared in Fig. 3 (cosine loss) and Fig. 5 (L2 loss). One of the most important differences is that the cosine loss dynamics has the regime shift depending on evolution of $N_\Phi$, $N_\Psi$, and $N_\times$, while the L2 loss dynamics does not have such a shift. Thus, the L2 loss dynamics and its time evolution are solely determined by a given regularization strength $\rho$ (see three plots in Fig. 5). That being said, if the L2 loss dynamics is regularized strongly such that $\rho > \frac{1}{4(1+\sigma^2)}$, there is no hope that the eigenmode stably converges without collapse to zero. On the contrary, a strong regularization with the cosine loss initially makes the dynamics fall into the Collapse regime, where no meaningful stable equilibrium exists, but the regime gradually shifts to Acute as the eigenmode (and the norms $N_\Phi$ and $N_\Psi$ accordingly) approaches zero. Such regime shift owes to feature normalization involved in the cosine loss.

### 5.4 NUMERICAL EXPERIMENTS

We conducted a simple numerical simulation of the SimSiam model using the official implementation available at `https://github.com/facebookresearch/simsiam`. We tested the linear model setup shown in Section 3, with linear representation net $\mathbf{\Phi}$ and linear projection head $\mathbf{W}$, and the representation dimension was set to $h = 64$. Data are generated from the 512-dimensional ($d = 512$) standard multivariate normal (Assumption 2) and data augmentation follows isotropic

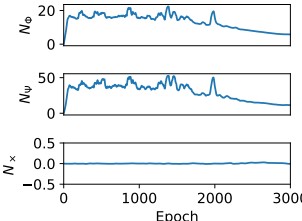 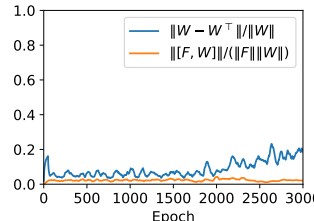 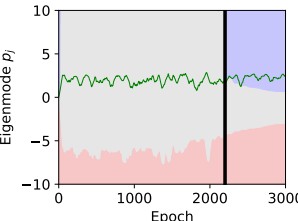

**Figure 6:** Numerical simulation of the SimSiam model. **(Left)** Time evolution of $N_\Phi$, $N_\Psi$, and $N_\times$. They overall remain stable (cf. Assumption 5). **(Center)** Asymmetry of the projection head $\mathbf{W}$ (measured by the relative error of $\mathbf{W} - \mathbf{W}^\top$) and non-commutativity of $\mathbf{F}$ and $\mathbf{W}$ (measured by the relative error of the commutator $[\mathbf{F}, \mathbf{W}]$). The relative errors stay close to zero during time evolution (cf. Assumptions 1 and 6). **(Right)** The leading eigenmode of the projection head $p_j$ (green line), with background colors illustrating three intervals where $p_j$ diverges , $p_j$ collapses , and $p_j$ stably converges at each epoch. The regime boundaries are numerically computed by the binary search and root finding of (8). Each color corresponds to those in Fig. 3. The vertical black line indicates the shift from Collapse (epoch $< 2200$) to Acute (epoch $> 2200$).

Gaussian noise $\mathcal{D}_{\mathbf{x}_0}^{\mathrm{aug}}$, with variance $\sigma^2 = 1.0$. The learning rate of the momentum SGD was initially set to $0.05$ and scheduled by the cosine annealing. The regularization strength was set to $\rho = 0.005$. For the other implementation details, we followed the official implementation.

The results are shown in Fig. 6. We first confirm how Assumption 5 is reasonable in practice by testing the values of $N_\Phi$, $N_\Psi$, and $N_\times$ during time evolution. Figure 6 (Left) shows that these three values, and $N_\times$ in particular, overall remain stable, with mild shrinkage of $N_\Phi$ and $N_\Psi$. Nevertheless, $N_\Phi$ and $N_\Psi$ occasionally have spikes. To take those behaviors into account, the local norm stability [WZZS21] would be useful in future analyses. Next, to confirm the validity of Assumptions 1 and 6, we plot asymmetry of the projection head $\mathbf{W}$ and commutativity of $\mathbf{F}$ and $\mathbf{W}$ in Fig. 6 (Center), which suggests that the assumptions are reasonable in general. Lastly, we empirically observe the regime shift in Fig. 6 (Right). The regularization strength $\rho = 0.005$ used in this experiment is rather larger than the default SimSiam regularization strength $\rho = 10^{-4}$, which leads to the Collapse regime initially (when epoch $< 2200$) but gradually shifts to the Acute regime (when epoch $> 2200$). Thus, we observed how the eigenmode escapes from the Collapse regime. More analyses (together with the other eigenmodes; additionally, the simulation with ResNet-18 encoder) can be found in Appendix D.

## 6 CONCLUSION

In this work, we questioned how to describe non-contrastive dynamics without eigenmode collapse. The existing theory (represented by [TCG21]) leverages the simplicity of the L2 loss to analytically derive the dynamics of the two-layer non-contrastive learning. However, the regularization severely affects eigenmode collapse: with too strong regularization, the dynamics has no way to escape from eigenmode collapse. This may indicate a drawback of the L2 loss analysis, though their theoretical model is transparent. Alternatively, we focused on the cosine loss, which involves feature normalization and derived the corresponding eigenmode dynamics. Despite that the dynamics may fall into the Collapse regime for too strong regularization, the shrinkage of the eigenmodes brings the regime into non-collapse ones. Thus, we witnessed the importance of feature normalization.

Technically, we leveraged the thermodynamical limit of the feature dimensions, which allows us to focus on high-dimensional concentrated feature norms. We believe that a similar device may enhance theoretical models of related learning problems and architectures, including self-supervised learning based on covariance regularization such as Barlow Twins [ZJM+21] and VICReg [BPL22].

This work is limited to the analysis of dynamics stability and refrains to answer why non-contrastive learning is appealing for many downstream tasks. While downstream performances of contrastive learning have been theoretically analyzed through the lens of the learning theoretic viewpoint [SPA+19, NS21, WZW+22, BNN22] and the smoothness of loss landscapes [LXLM23], we have far less understanding of non-contrastive learning for the time being. We hope that understanding the non-contrastive dynamics paves a road to the analysis of downstream tasks.

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

# APPENDIX

ON THE ROLE OF FEATURE NORMALIZATION IN NON-CONTRASTIVE SELF-SUPERVISED LEARNING

## A  TECHNICAL LEMMAS

### A.1  SUB-WEIBULL DISTRIBUTIONS

In this subsection, we give a brief introduction to *sub-Weibull distributions* [HAYWC19, VGNA20], which is a generalization of seminal sub-Gaussian and sub-exponential random variables. First, we define sub-Weibull distributions.

**Definition 1** ([HAYWC19]). *For $\beta > 0$, we define $X$ as a sub-Weibull random variable with the $\psi_\beta$-norm if it entails a bounded $\psi_\beta$-norm, defined as follows:*

$$\|X\|_{\psi_\beta} := \inf \left\{ C \in (0, \infty) \ \Big| \ \mathbb{E}[\exp(|X|^\beta / C^\beta)] \leq 2 \right\}.$$

We occasionally call $\beta$-*sub-Weibull* to specify the corresponding $\psi_\beta$-norm explicitly. Obviously, $\beta = 2$ and $\beta = 1$ recover sub-Gaussian and sub-exponential distributions, respectively. Among equivalent definitions of sub-Weibull distributions, we often use the following conditions.

**Proposition 3** ([VGNA20]). *Let $X$ be a sub-Weibull random variable. Then, the following conditions are equivalent:*

*1. The tails of $X$ satisfy*

$$\exists K_1 > 0 \quad such\ that \quad \mathbb{P}\{|X| \geq \varepsilon\} \leq 2\exp\left(-(\varepsilon/K_1)^\beta\right) \quad for\ all\ \varepsilon \geq 0.$$

*2. The moments of $X$ satisfy*

$$\exists K_2 > 0 \quad such\ that \quad \|X\|_{L^p} := \{\mathbb{E}\,|X|^p\}^{1/p} \leq K_2 p^{1/\beta} \quad for\ all\ p \geq 1.$$

*3. The moment-generating function (MGF) of $|X|^\beta$ is bounded at some point, namely,*

$$\exists K_3 > 0 \quad such\ that \quad \mathbb{E}\exp\left((|X|/K_3)^\beta\right) \leq 2.$$

*The parameters $K_1$, $K_2$, and $K_3$ differ from each other by at most an absolute constant factor.*

We are interested in sub-Weibull distributions because they admit a nice closure property, as shown below.

**Proposition 4** ([VGNA20]). *Let $X$ and $Y$ be $\beta$-sub-Weibull random variables. Then, $XY$ is $(\beta/2)$-sub-Weibull with $\|XY\|_{\psi_{\beta/2}} \leq \|X\|_{\psi_\beta} \|Y\|_{\psi_\beta}$. In addition, $X + Y$ is $\beta$-sub-Weibull with $\|X + Y\|_{\psi_\beta} \leq \|X\|_{\psi_\beta} + \|Y\|_{\psi_\beta}$.*

Note that Proposition 4 does not require the independence of two random variables $X$ and $Y$. Lastly, we show a corresponding concentration inequality for the sum of independent sub-Weibull random variables, which is a generalization of Hoeffding's and Bernstein's inequalities for sub-Gaussian and sub-exponential random variables, respectively.

**Proposition 5** ([HAYWC19]). *Let $X_1, \ldots, X_N$ be independent $\beta$-sub-Weibull random variables with $\|X_i\|_{\psi_\beta} \leq K$ for each $i \in [N]$. Then, there exists an absolute constant $C > 0$ only depending on $\beta$ such that for any $\delta \in (0, e^{-2})$,*

$$\left| \sum_{i=1}^N X_i - \mathbb{E}\left[\sum_{i=1}^N X_i\right] \right| \leq CK\left( \sqrt{N \log \frac{1}{\delta}} + \left(\log \frac{1}{\delta}\right)^{1/\beta} \right),$$

*with probability at least $1 - \delta$.*

For the proofs of these propositions, please refer to the corresponding references.

We additionally provide technical lemmas for random matrices whose element is sub-Weibull.

**Lemma 5.** *Let $\mathbf{G} \in \mathbb{R}^{h \times d}$ be a random matrix with each element being $\beta$-sub-Weibull such that $\|G_{ij}\|_{\psi_\beta} = \mathcal{O}(K(d, h))$ for some $\beta > 0$ and any $(i, j) \in [h] \times [d]$, where $K(d, h)$ may depend on $d$ and $h$. Then, $\frac{1}{d^2 h^2} \|\mathbf{G}^\top \mathbf{G}\|_{\mathrm{F}}^2 = \mathcal{O}_{\mathbb{P}}(K(d, h)/h)$.*

*Proof of Lemma 5.* Let $\mathbf{G}_i \in \mathbb{R}^h$ denote the $i$-th column vector of the matrix $\mathbf{G}$. We have the decomposition $\|\mathbf{G}^\top \mathbf{G}\|_{\mathrm{F}}^2 = \sum_{i,j=1}^d \langle \mathbf{G}_i, \mathbf{G}_j \rangle^2$. Let us focus on each $\langle \mathbf{G}_i, \mathbf{G}_j \rangle$ for fixed $i$ and $j$ first. We can decompose into $\langle \mathbf{G}_i, \mathbf{G}_j \rangle = \sum_{k=1}^h G_{ik} G_{jk}$, which is the sum of $(\beta/2)$-sub-Weibull random variable $G_{ik} G_{jk}$ with $\|G_{ik} G_{jk}\|_{\psi_{\beta/2}} = \mathcal{O}(K(d, h))$ (cf. Proposition 4). By using the closure property under addition (Proposition 4), the sum $\langle \mathbf{G}_i, \mathbf{G}_j \rangle$ is $(\beta/2)$-sub-Weibull again, with $\|\langle \mathbf{G}_i, \mathbf{G}_j \rangle\|_{\psi_{\beta/2}} = \mathcal{O}(hK(d, h))$.

Now, we move back to evaluation of $\|\mathbf{G}^\top \mathbf{G}\|_{\mathrm{F}}^2 = \sum_{i,j=1}^d \langle \mathbf{G}_i, \mathbf{G}_j \rangle^2$. By using the closure property under multiplication (Proposition 4), $\langle \mathbf{G}_i, \mathbf{G}_j \rangle^2$ is $(\beta/4)$-sub-Weibull with $\left\| \langle \mathbf{G}_i, \mathbf{G}_j \rangle^2 \right\|_{\psi_{\beta/4}} = \mathcal{O}(hK(d, h))$. Then, the closure property under addition implies that $\sum_{i,j=1}^d \langle \mathbf{G}_i, \mathbf{G}_j \rangle^2$ is $(\beta/4)$-sub-Weibull, with $\left\| \sum_{i,j=1}^d \langle \mathbf{G}_i, \mathbf{G}_j \rangle^2 \right\|_{\psi_{\beta/4}} = \mathcal{O}(d^2 h K(d, h))$. Hence, by using the sub-Weibull tails in Proposition 3,

$$\|\mathbf{G}^\top \mathbf{G}\|_{\mathrm{F}}^2 = \sum_{i,j=1}^d \langle \mathbf{G}_i, \mathbf{G}_j \rangle^2 = \mathcal{O}_{\mathbb{P}}(d^2 h K(d, h)),$$

from which we deduce that $\frac{1}{d^2 h^2} \|\mathbf{G}^\top \mathbf{G}\|_{\mathrm{F}}^2 = \mathcal{O}_{\mathbb{P}}(K(d, h)/h)$. $\qquad \square$

**Lemma 6.** *Let $\mathbf{G} \in \mathbb{R}^{h \times d}$ be a random matrix with each element being $\beta$-sub-Weibull such that $\|G_{ij}\|_{\psi_\beta} = \mathcal{O}(K(d, h))$ for some $\beta > 0$ and any $i, j \in [d]$, where $K(d, h)$ may depend on $d$ and $h$. Then, $\|\mathbf{G}\| = \mathcal{O}_{\mathbb{P}}((d^{1/\beta} + h^{1/\beta}) K(d, h))$.*

*Proof of Lemma 6.* The proof is akin to [Ver18, Theorem 4.4.5], which is a spectral norm deviation for sub-Gaussian random matrices. We leverage the $\varepsilon$-*net argument*: Using [Ver18, Corollary 4.2.13], we can find $\varepsilon$-nets $\mathcal{M}_d$ of $\mathbb{S}^{d-1}$ with $|\mathcal{M}_d| \leq 9^d$ and $\mathcal{M}_h$ of $\mathbb{S}^{h-1}$ with $|\mathcal{M}_h| \leq 9^h$, and $\|\mathbf{G}\| \leq 2 \max_{\mathbf{x} \in \mathcal{M}_d, \mathbf{y} \in \mathcal{M}_h} \langle \mathbf{G}\mathbf{x}, \mathbf{y} \rangle$. Hence, it is sufficient to control the quadratic form $\langle \mathbf{G}\mathbf{x}, \mathbf{y} \rangle$ for fixed $(\mathbf{x}, \mathbf{y}) \in \mathcal{M}_d \times \mathcal{M}_h$.

The quadratic form $\langle \mathbf{G}\mathbf{x}, \mathbf{y} \rangle = \sum_{i=1}^d \sum_{j=1}^h G_{ij} x_i y_j$ is the sum of $\beta$-sub-Weibull random variables. By the closure property (Proposition 4),

$$\|\langle \mathbf{G}\mathbf{x}, \mathbf{y} \rangle\|_{\psi_\beta}^2 \leq \sum_{i,j} \|G_{ij} x_i y_j\|_{\psi_\beta}^2 \leq \mathcal{O}(K(d, h)) \cdot \left( \sum_{i=1}^d x_i^2 \right) \left( \sum_{j=1}^h y_j^2 \right) = \mathcal{O}(K(d, h)).$$

Thus, sub-Weibull tails (Proposition 3) imply $\mathbb{P}\{\langle \mathbf{G}\mathbf{x}, \mathbf{y} \rangle \geq u\} \leq 2 \exp(-(u/K_1)^\beta)$ with $K_1 = \mathcal{O}(K(d, h))$. The union bound yields

$$\mathbb{P}\left\{ \max_{\mathbf{x} \in \mathcal{M}_d, \mathbf{y} \in \mathcal{M}_h} \langle \mathbf{G}\mathbf{x}, \mathbf{y} \rangle \geq u \right\} \leq 9^{d+h} \cdot 2 \exp(-(u/K_1)^\beta) \leq 2 \exp(-\delta^\beta),$$

where the last inequality is a consequence of the choice $u = C K_1 (d^{1/\beta} + h^{1/\beta} + \delta)$ with a sufficiently large absolute constant $C$. Hence, $\mathbb{P}\{\|\mathbf{G}\| \geq 2u\} \leq 2 \exp(-\delta^\beta)$ holds, namely, $\|\mathbf{G}\| = 2C(d^{1/\beta} + h^{1/\beta} + \delta) \cdot \mathcal{O}(K(d, h))$ holds with probability at least $1 - 2 \exp(-\delta^\beta)$. This completes the proof. $\qquad \square$

## A.2 INTEGRAL INEQUALITY

In this subsection, we briefly introduce the Grönwall–Bellman inequality [Bel43, GBLJ19] to solve functional inequalities represented by integrals. In subsequent analyses, we heavily use it to control the norm of certain random matrices during time evolution.

**Theorem 1** (Grönwall–Bellman inequality). *Let $\beta$ be a non-negative function and $\alpha$ a non-decreasing function. Let $u$ be a function defined on an interval $\mathcal{I} = [0, \infty)$ such that*

$$u(t) \le \alpha(t) + \int_0^t \beta(s) u(s) \mathrm{d}s, \quad \forall t \in \mathcal{I}.$$

*Then, we have*

$$u(t) \le \alpha(t) \exp\left(\int_0^t \beta(s) \mathrm{d}s\right), \quad \forall t \in \mathcal{I}.$$

## A.3 HELPER LEMMAS

**Lemma 7.** *Under the initialization of Assumption 4, we have the following results:*

1. $\frac{1}{h} \left\| \mathbf{\Phi}^\top \mathbf{\Phi}(0) \right\| = o_{\mathbb{P}}(1).$

2. $\frac{1}{h^2} \left\| \mathbf{\Phi}^\top \mathbf{W}^\top \mathbf{W} \mathbf{\Phi}(0) \right\| = o_{\mathbb{P}}(1).$

*Proof of Lemma 7.* To prove 1, we note that each element of the random matrix $\sqrt{d}\mathbf{\Phi}(0)$ is sub-Gaussian (namely, 2-sub-Weibull) with the $\psi_2$-norm being $\mathcal{O}(1)$, by the assumption on the parameter initialization (Assumption 4). Then, Lemma 6 implies $d \left\| \mathbf{\Phi}^\top \mathbf{\Phi}(0) \right\| = \left\| \sqrt{d}\mathbf{\Phi}(0) \right\|^2 = \mathcal{O}_{\mathbb{P}}(d)$. Finally, we have $\frac{1}{h} \left\| \mathbf{\Phi}^\top \mathbf{\Phi}(0) \right\| = \mathcal{O}_{\mathbb{P}}(1/h) = o_{\mathbb{P}}(1)$.

The identity 2 follows similarly. The $(i, j)$-th element of the random matrix $\sqrt{dh}\mathbf{W}\mathbf{\Phi}(0)$ can be expressed as $\langle \mathbf{w}_i, \mathbf{\Phi}_j \rangle$, where $\mathbf{w}_i$ is the $i$-th row vector of $\sqrt{h}\mathbf{W}(0)$ and $\mathbf{\Phi}_j$ is the $j$-th column vector of $\sqrt{d}\mathbf{\Phi}(0)$. Both $\mathbf{w}_i$ and $\mathbf{\Phi}_j$ are $h$-dimensional vectors with each element being standard normal. Hence, $\langle \mathbf{w}_i, \mathbf{\Phi}_j \rangle$ is the sum of $h$ sub-exponential random variables, being sub-exponential with $\| \langle \mathbf{w}_i, \mathbf{\Phi}_j \rangle \|_{\psi_1} = \mathcal{O}(h)$ (by using Proposition 4). This indicates that each element of $\sqrt{dh}\mathbf{W}\mathbf{\Phi}(0)$ is sub-exponential (namely, 1-sub-Weibull). Then, Lemma 6 implies $dh \left\| \mathbf{\Phi}^\top \mathbf{W}^\top \mathbf{W} \mathbf{\Phi}(0) \right\| = \left\| \sqrt{dh}\mathbf{W}\mathbf{\Phi}(0) \right\|^2 = \mathcal{O}_{\mathbb{P}}(d^2)$. Finally, we have $\frac{1}{h^2} \left\| \mathbf{\Phi}^\top \mathbf{W}^\top \mathbf{W} \mathbf{\Phi}(0) \right\| = \mathcal{O}_{\mathbb{P}}(1/h^2) = o_{\mathbb{P}}(1)$. $\square$

**Lemma 8.** *Under the initialization of Assumption 4, we have the following results:*

1. $\frac{1}{h^2} \left\| \mathbf{\Phi}^\top \mathbf{\Phi}(0) \right\|_{\mathrm{F}}^2 = o_{\mathbb{P}}(1).$

2. $\frac{1}{h^2} \left\| \mathbf{\Phi}^\top \mathbf{W}^\top \mathbf{W} \mathbf{\Phi}(0) \right\|_{\mathrm{F}}^2 = o_{\mathbb{P}}(1).$

3. $\frac{1}{h^2} \mathrm{tr}(\mathbf{W}^\top \mathbf{W}(0))^2 = \mathcal{O}_{\mathbb{P}}(1).$

*Proof of Lemma 8.* Let us prove 1. Again, each element of the random matrix $\sqrt{d}\mathbf{\Phi}(0)$ is 2-sub-Weibull (see the proof of Lemma 7). Thus, Lemma 5 implies $\frac{1}{h^2} \left\| \mathbf{\Phi}^\top \mathbf{\Phi}(0) \right\|_{\mathrm{F}}^2 = \frac{1}{d^2 h^2} \cdot d^2 \left\| \mathbf{\Phi}^\top \mathbf{\Phi}(0) \right\|_{\mathrm{F}}^2 = \mathcal{O}_{\mathbb{P}}(1/h) = o_{\mathbb{P}}(1)$.

The identity 2 follows similarly. Again, each element of the random matrix $\sqrt{dh}\mathbf{W}\mathbf{\Phi}(0)$ is 1-sub-Weibull (see the proof of Lemma 7) so that $\sqrt{dh}\mathbf{W}\mathbf{\Phi}(0)$ satisfies the assumption of Lemma 5, from which we deduce that

$$\frac{1}{h^2} \left\| \mathbf{\Phi}^\top \mathbf{W}^\top \mathbf{W} \mathbf{\Phi}(0) \right\|_{\mathrm{F}}^2 = \frac{1}{h^2} \cdot \frac{1}{d^2 h^2} \cdot d^2 h^2 \left\| \mathbf{\Phi}^\top \mathbf{W}^\top \mathbf{W} \mathbf{\Phi}(0) \right\|_{\mathrm{F}}^2$$
$$= \frac{1}{h^2} \mathcal{O}_{\mathbb{P}}(1)$$
$$= o_{\mathbb{P}}(1).$$

To prove 3, we see that $h \operatorname{tr}(\mathbf{W}^\top \mathbf{W}(0)) = h \|\mathbf{W}(0)\|_{\mathrm{F}}^2 = \sum_{i,j=1}^h (\sqrt{h} W(0)_{ij})^2$ is the sum of sub-exponential (namely, 1-sub-Weibull) random variables $(\sqrt{h} W(0)_{ij})^2$ with $\left\| \sqrt{h} W(0)_{ij} \right\|_{\psi_1} = \mathcal{O}(1)$ for $i, j \in [h]$. Hence, $h \operatorname{tr}(\mathbf{W}^\top \mathbf{W}(0))$ is 1-sub-Weibull with $\left\| h \operatorname{tr}(\mathbf{W}^\top \mathbf{W}(0)) \right\|_{\psi_1} = \mathcal{O}(h^2)$ from Proposition 4. By the closure property again, $h^2 \operatorname{tr}(\mathbf{W}^\top \mathbf{W}(0))^2$ is $\frac{1}{2}$-sub-Weibull with the corresponding norm being $\mathcal{O}(h^4)$. By using sub-Weibull tails in Proposition 3, we deduce that $\left| h^2 \operatorname{tr}(\mathbf{W}^\top \mathbf{W}(0))^2 \right| = \mathcal{O}_{\mathbb{P}}(h^2)$. Lastly, we obtain $\frac{1}{h^2} \operatorname{tr}(\mathbf{W}^\top \mathbf{W}(0))^2 = \mathcal{O}_{\mathbb{P}}(1)$. $\qquad \square$

**Lemma 9.** *Under Assumptions 2 and 4, we have the following consequences:*

1. $\frac{1}{h^2} \left\| \mathbf{\Phi}^\top \mathbf{\Phi}(0) \mathbf{x}_0 \right\|_2^2 = o_{\mathbb{P}}(1).$

2. $\frac{1}{h^4} \left\| \mathbf{\Phi}^\top \mathbf{W}^\top \mathbf{W} \mathbf{\Phi}(0) \mathbf{x}_0 \right\|_2^2 = o_{\mathbb{P}}(1).$

*Proof of Lemma 9.* Assumption 2 implies that $\mathbf{x}_0 \sim \mathcal{N}(\mathbf{0}, \mathbf{I}_d)$, from which we can verify that $\|\mathbf{x}_0\|_2^2 = \sum_{i=1}^d x_{0,i}^2$ is the sum of $d$ sub-exponential (i.e., 1-sub-Weibull) random variables and $\left\| \|\mathbf{x}_0\|_2^2 \right\|_{\psi_1} = \mathcal{O}(d)$ (Proposition 4). By sub-Weibull tails (Proposition 3), $\|\mathbf{x}_0\|_2^2 = \mathcal{O}_{\mathbb{P}}(d)$ entails.

To prove 1, we confirm that each element of $h \mathbf{\Phi}^\top \mathbf{\Phi}(0)$ is sub-exponential with the $\psi_1$-norm being $\mathcal{O}(1)$. To see this, we let $\mathbf{\Phi}_i$ denote the $i$-th column vector of $\sqrt{h} \mathbf{\Phi}(0)$. Assumption 4 indicates that $\mathbf{\Phi}_i$ is an $h$-dimensional standard normal random vector, and $\mathbb{E} \langle \mathbf{\Phi}_i, \mathbf{\Phi}_j \rangle = h \cdot [\![i = j]\!]$. Thus, Bernstein's inequality [Ver18, Corollary 2.8.3] yields $|\langle \mathbf{\Phi}_i, \mathbf{\Phi}_j \rangle - h \cdot [\![i = j]\!]| = \mathcal{O}_{\mathbb{P}}(1)$ (for sufficiently large $h$), which indicates that $h \mathbf{\Phi}^\top \mathbf{\Phi}(0) - h \mathbf{I}_d$ satisfies the assumption of Lemma 6 with $\beta = 1$ and $K(d, h) = 1$. Hence, by Lemma 6,

$$\left\| h \mathbf{\Phi}^\top \mathbf{\Phi}(0) \right\| \leq \left\| h \mathbf{\Phi}^\top \mathbf{\Phi}(0) - h \mathbf{I}_d \right\| + h \|\mathbf{I}_d\| = \mathcal{O}_{\mathbb{P}}(d) + h.$$

Combining this with $\|\mathbf{x}_0\|_2^2 = \mathcal{O}_{\mathbb{P}}(d)$, we obtain the following result:

$$\frac{1}{h^2} \left\| \mathbf{\Phi}^\top \mathbf{\Phi}(0) \mathbf{x}_0 \right\|_2^2 \leq \frac{1}{h^4} \cdot \left\| h \mathbf{\Phi}^\top \mathbf{\Phi}(0) \right\|^2 \cdot \|\mathbf{x}_0\|_2^2 = \frac{1}{h^4} \cdot \{\mathcal{O}_{\mathbb{P}}(d) + h\}^2 \cdot \mathcal{O}_{\mathbb{P}}(d) = \mathcal{O}_{\mathbb{P}}(h^{-1}),$$

which completes the proof.

To prove 2, we confirm that each element of $h^2 \mathbf{\Phi}^\top \mathbf{W}^\top \mathbf{W} \mathbf{\Phi}(0)$ is $\frac{1}{2}$-sub-Weibull with the $\psi_{\frac{1}{2}}$-norm being $\mathcal{O}(\sqrt{h})$. To see this, we let $\mathbf{\Psi}_i$ denote the $i$-th column vector of $h \mathbf{W}(0) \mathbf{\Phi}(0)$ (for $i \in [d]$). The $k$-th element of $\mathbf{\Psi}_i$ (for $k \in [h]$) is $\Psi_i^{(k)} := h \sum_{l=1}^h W(0)_{kl} \Phi(0)_{li}$, which is sub-exponential and mean zero from Assumption 4 and $\left| \Psi_i^{(k)} \right| = \left| h \sum_{l=1}^h W(0)_{kl} \Phi(0)_{li} \right| = \mathcal{O}_{\mathbb{P}}(1)$ (for sufficiently large $h$) from Bernstein's inequality. Here, each $(i, j)$-th element of $h^2 \mathbf{\Phi}^\top \mathbf{W}^\top \mathbf{W} \mathbf{\Phi}(0)$ is $\langle \mathbf{\Psi}_i, \mathbf{\Psi}_j \rangle = \sum_{k=1}^h \Psi_i^{(k)} \Psi_j^{(k)}$, which is the sum of $h$ products $\Psi_i^{(k)} \Psi_j^{(k)}$. Each $\Psi_i^{(k)} \Psi_j^{(k)}$ is $\frac{1}{2}$-sub-Weibull because of the closure property (Proposition 4), and hence the sum $\langle \mathbf{\Psi}_i, \mathbf{\Psi}_j \rangle$ is $\frac{1}{2}$-sub-Weibull with $\|\langle \mathbf{\Psi}_i, \mathbf{\Psi}_j \rangle\|_{\psi_{\frac{1}{2}}} = \mathcal{O}(h)$. Thus, we see the sub-Weibull property of $h^2 \mathbf{\Phi}^\top \mathbf{W}^\top \mathbf{W} \mathbf{\Phi}(0)$.

Hence, we can apply Lemma 6 to claim $\left\| h^2 \mathbf{\Phi}^\top \mathbf{W}^\top \mathbf{W} \mathbf{\Phi}(0) \right\| = \mathcal{O}_{\mathbb{P}}(d^2 h)$. Combining this with $\|\mathbf{x}_0\|_2^2 = \mathcal{O}_{\mathbb{P}}(d)$, we obtain the desired result:

$$\begin{aligned}
\frac{1}{h^4} \left\| \mathbf{\Phi}^\top \mathbf{W}^\top \mathbf{W} \mathbf{\Phi}(0) \mathbf{x}_0 \right\|_2^2 &\leq \frac{1}{h^8} \cdot \left\| h^2 \mathbf{\Phi}^\top \mathbf{W}^\top \mathbf{W} \mathbf{\Phi}(0) \right\|^2 \cdot \|\mathbf{x}_0\|_2^2 \\
&= \frac{1}{h^8} \cdot \mathcal{O}_{\mathbb{P}}(d^4 h^2) \cdot \mathcal{O}_{\mathbb{P}}(d) \\
&= \mathcal{O}_{\mathbb{P}}(h^{-1}).
\end{aligned}$$

$\qquad \square$

**Lemma 10.** *For any $t$, $\left\| \mathbf{\Phi}^\top \mathbf{\Phi}(t) \right\|_{\mathrm{F}} \leq \left( \left\| \mathbf{\Phi}^\top \mathbf{\Phi}(0) \right\|_{\mathrm{F}} + 4t \right) \exp(2\rho t).$*

*Proof of Lemma 10.* First, we use the fundamental theorem of calculus and the triangular inequality to decompose as follows:

$$
\begin{aligned}
\left\|\mathbf{\Phi}^\top\mathbf{\Phi}(t)\right\|_{\mathrm{F}} &= \left\|\mathbf{\Phi}^\top\mathbf{\Phi}(0) + \int_0^t \left\{ \dot{\mathbf{\Phi}}^\top\mathbf{\Phi}(\tau) + \mathbf{\Phi}^\top\dot{\mathbf{\Phi}}(\tau) \right\} \mathrm{d}\tau \right\|_{\mathrm{F}} \\
&\leq \left\|\mathbf{\Phi}^\top\mathbf{\Phi}(0)\right\|_{\mathrm{F}} + \int_0^t \left\|\dot{\mathbf{\Phi}}^\top\mathbf{\Phi}(\tau)\right\|_{\mathrm{F}} \mathrm{d}\tau + \int_0^t \left\|\mathbf{\Phi}^\top\dot{\mathbf{\Phi}}(\tau)\right\|_{\mathrm{F}} \mathrm{d}\tau \qquad (10) \\
&= \left\|\mathbf{\Phi}^\top\mathbf{\Phi}(0)\right\|_{\mathrm{F}} + 2\int_0^t \left\|\dot{\mathbf{\Phi}}^\top\mathbf{\Phi}(\tau)\right\|_{\mathrm{F}} \mathrm{d}\tau.
\end{aligned}
$$

The term $\dot{\mathbf{\Phi}}^\top\mathbf{\Phi}$ can be evaluated by using the dynamics derived in Lemma 1 as follows:

$$
\begin{aligned}
\dot{\mathbf{\Phi}}^\top\mathbf{\Phi} &= \left\{ \mathbf{W}^\top(\mathbf{W}^\top\dot{\mathbf{W}} + \rho\mathbf{W}\mathbf{W}^\top)(\mathbf{W}^\top)^\dagger(\mathbf{\Phi}^\top)^\dagger - \rho\mathbf{\Phi}\mathbf{\Phi}^\top\mathbf{W}^\top(\mathbf{W}^\top)^\dagger(\mathbf{\Phi}^\top)^\dagger \right\}^\top \mathbf{\Phi} \\
&= \left\{ \mathbf{W}^\top\mathbf{W}^\top\dot{\mathbf{W}}(\mathbf{W}^\dagger)^\top(\mathbf{\Phi}^\dagger)^\top + \rho\mathbf{W}^\top\mathbf{W}(\mathbf{\Phi}^\dagger)^\top - \rho\mathbf{\Phi} \right\}^\top \mathbf{\Phi} \\
&= \mathbf{\Phi}^\dagger\mathbf{W}^\dagger\dot{\mathbf{W}}^\top\mathbf{W}^2\mathbf{\Phi} + \rho\mathbf{\Phi}^\dagger\mathbf{W}^\top\mathbf{W}\mathbf{\Phi} - \rho\mathbf{\Phi}^\top\mathbf{\Phi} \\
&= \mathbf{\Phi}^\dagger\mathbf{W}^\dagger\{\mathbb{E}[\mathbf{z}'\boldsymbol{\omega}^\top - (\boldsymbol{\omega}^\top\mathbf{z}')\boldsymbol{\omega}\boldsymbol{\omega}^\top] - \rho\mathbf{W}\mathbf{W}^\top\}\mathbf{W}\mathbf{\Phi} + \rho\mathbf{\Phi}^\dagger\mathbf{W}^\top\mathbf{W}\mathbf{\Phi} - \rho\mathbf{\Phi}^\top\mathbf{\Phi} \\
&= \mathbf{\Phi}^\dagger\mathbf{W}^\dagger\,\mathbb{E}[\mathbf{z}'\boldsymbol{\omega}^\top - (\boldsymbol{\omega}^\top\mathbf{z}')\boldsymbol{\omega}\boldsymbol{\omega}^\top]\mathbf{W}\mathbf{\Phi} - \rho\mathbf{\Phi}^\top\mathbf{\Phi},
\end{aligned}
\qquad (11)
$$

whose Frobenius norm shall be bounded from above subsequently:

$$
\left\|\dot{\mathbf{\Phi}}^\top\mathbf{\Phi}\right\|_{\mathrm{F}} \leq \mathbb{E}\left\|\mathbf{\Phi}^\dagger\mathbf{W}^\dagger(\mathbf{z}'\boldsymbol{\omega}^\top)\mathbf{W}\mathbf{\Phi}\right\|_{\mathrm{F}} + \mathbb{E}\left\|\mathbf{\Phi}^\dagger\mathbf{W}^\dagger(\boldsymbol{\omega}\boldsymbol{\omega}^\top)\mathbf{W}\mathbf{\Phi}\right\|_{\mathrm{F}} + \rho\left\|\mathbf{\Phi}^\top\mathbf{\Phi}\right\|_{\mathrm{F}}.
$$

Note that we use $\left|\boldsymbol{\omega}^\top\mathbf{z}'\right| \leq 1$ because $\boldsymbol{\omega}, \mathbf{z}' \in \mathbb{S}^{h-1}$ in this bound. The norm $\left\|\mathbf{\Phi}^\dagger\mathbf{W}^\dagger(\mathbf{z}'\boldsymbol{\omega}^\top)\mathbf{W}\mathbf{\Phi}\right\|_{\mathrm{F}}$ is bounded as follows:

$$
\begin{aligned}
\left\|\mathbf{\Phi}^\dagger\mathbf{W}^\dagger(\mathbf{z}'\boldsymbol{\omega}^\top)\mathbf{W}\mathbf{\Phi}\right\|_{\mathrm{F}}^2 &= \left\langle \mathbf{\Phi}^\dagger\mathbf{W}^\dagger(\mathbf{z}'\boldsymbol{\omega}^\top)\mathbf{W}\mathbf{\Phi}, \mathbf{\Phi}^\dagger\mathbf{W}^\dagger(\mathbf{z}'\boldsymbol{\omega}^\top)\mathbf{W}\mathbf{\Phi} \right\rangle_{\mathrm{F}} \\
&= \mathrm{tr}(\mathbf{\Phi}^\top\mathbf{W}^\top\boldsymbol{\omega}(\mathbf{z}')^\top(\mathbf{W}^\dagger)^\top(\mathbf{\Phi}^\dagger)^\top\mathbf{\Phi}^\dagger\mathbf{W}^\dagger\mathbf{z}'\boldsymbol{\omega}^\top\mathbf{W}\mathbf{\Phi}) \\
&\overset{(*)}{=} \mathrm{tr}(\boldsymbol{\omega}(\mathbf{z}')^\top(\mathbf{W}^\dagger)^\top(\mathbf{\Phi}^\dagger)^\top\mathbf{\Phi}^\dagger\mathbf{W}^\dagger\mathbf{z}'\boldsymbol{\omega}^\top\mathbf{W}\mathbf{\Phi}\mathbf{\Phi}^\top\mathbf{W}^\top) \\
&\leq \left|\mathrm{tr}(\boldsymbol{\omega}(\mathbf{z}')^\top)\right| \cdot \left|\mathrm{tr}((\mathbf{W}^\dagger)^\top(\mathbf{\Phi}^\dagger)^\top\mathbf{\Phi}^\dagger\mathbf{W}^\dagger\mathbf{z}'\boldsymbol{\omega}^\top\mathbf{W}\mathbf{\Phi}\mathbf{\Phi}^\top\mathbf{W}^\top)\right| \qquad (12) \\
&\overset{(*)}{=} \left|\mathrm{tr}(\boldsymbol{\omega}(\mathbf{z}')^\top)\right| \cdot \left|\mathrm{tr}(\mathbf{z}'\boldsymbol{\omega}^\top)\right| \\
&= \|\boldsymbol{\omega}\|_2 \|\mathbf{z}'\|_2 \|\mathbf{z}'\|_2 \|\boldsymbol{\omega}\|_2 \\
&\leq 1,
\end{aligned}
$$

where the cyclic property of the trace $\mathrm{tr}(\mathbf{A}\mathbf{B}\mathbf{C}) = \mathrm{tr}(\mathbf{B}\mathbf{C}\mathbf{A})$ is used at the two identities (*). Because Eq. (12) relies solely on $\mathbf{z}', \boldsymbol{\omega} \in \mathbb{S}^{h-1}$, the same reasoning induces the upper bound $\left\|\mathbf{\Phi}^\dagger\mathbf{W}^\dagger(\boldsymbol{\omega}\boldsymbol{\omega}^\top)\mathbf{W}\mathbf{\Phi}\right\|_{\mathrm{F}} \leq 1$. By plugging everything back to Eq. (10), we obtain the following integral inequality for the norm $\left\|\mathbf{\Phi}^\top\mathbf{\Phi}(t)\right\|_{\mathrm{F}}$:

$$
\left\|\mathbf{\Phi}^\top\mathbf{\Phi}(t)\right\|_{\mathrm{F}} \leq \left\|\mathbf{\Phi}^\top\mathbf{\Phi}(0)\right\|_{\mathrm{F}} + 4t + 2\rho\int_0^t \left\|\mathbf{\Phi}^\top\mathbf{\Phi}(\tau)\right\|_{\mathrm{F}} \mathrm{d}\tau. \qquad (13)
$$

The form of Eq. (13) satisfies the assumption of the Grönwall–Bellman inequality (Theorem 1) with which the norm upper bound $\left\|\mathbf{\Phi}^\top\mathbf{\Phi}(t)\right\|_{\mathrm{F}} \leq (\left\|\mathbf{\Phi}^\top\mathbf{\Phi}(0)\right\|_{\mathrm{F}} + 4t)\exp(2\rho t)$ is derived. □

**Lemma 11.** *For any $t$, $\|\mathbf{\Phi}(t)\| \leq \sqrt{(\|\mathbf{\Phi}^\top\mathbf{\Phi}(0)\| + 4t)\exp(2\rho t)}$.*

*Proof of Lemma 11.* We evaluate $\left\|\mathbf{\Phi}^\top\mathbf{\Phi}(t)\right\| = \|\mathbf{\Phi}(t)\|^2$. By the fundamental theorem of calculus, we obtain the following decomposition:

$$
\left\|\mathbf{\Phi}^\top\mathbf{\Phi}(t)\right\| \leq \left\|\mathbf{\Phi}^\top\mathbf{\Phi}(0)\right\| + 2\int_0^t \left\|\dot{\mathbf{\Phi}}^\top\mathbf{\Phi}(\tau)\right\| \mathrm{d}\tau.
$$

By following the same derivation as the proof of Lemma 10, it is not difficult to see $\left\|\dot{\mathbf{\Phi}}^\top\mathbf{\Phi}\right\| \leq 2+\rho\left\|\mathbf{\Phi}^\top\mathbf{\Phi}\right\|$. Then, $\left\|\mathbf{\Phi}^\top\mathbf{\Phi}(t)\right\| \leq \left\|\mathbf{\Phi}^\top\mathbf{\Phi}(0)\right\| + 4t + 2\rho\int_0^t \left\|\mathbf{\Phi}^\top\mathbf{\Phi}(\tau)\right\| \mathrm{d}\tau$. This integral inequality can be solved via Theorem 1, and we have $\left\|\mathbf{\Phi}^\top\mathbf{\Phi}(t)\right\| \leq (\left\|\mathbf{\Phi}^\top\mathbf{\Phi}(0)\right\| + 4t)\exp(2\rho t)$. □

**Lemma 12.** *For* $\mathbf{W} \in \mathbb{S}\mathrm{ym}_h$*, for any* $t$*,* $\mathrm{tr}(\mathbf{W}^\top \mathbf{W}(t)) \le (\mathrm{tr}(\mathbf{W}^\top \mathbf{W}(0)) + 4t) \exp(2\rho t)$.

*Proof of Lemma 12.* By the fundamental theorem of calculus, we obtain the following decomposition:

$$\mathrm{tr}(\mathbf{W}^\top \mathbf{W}(t)) \le \mathrm{tr}(\mathbf{W}^\top \mathbf{W}(0)) + 2 \int_0^t \mathrm{tr}(\mathbf{W}^\top \dot{\mathbf{W}}(\tau)) \mathrm{d}\tau.$$

By using the dynamics in Lemma 1, we further obtain the bound of $\mathrm{tr}(\mathbf{W}^\top \dot{\mathbf{W}})$:

$$
\begin{aligned}
\mathrm{tr}(\mathbf{W}^\top \dot{\mathbf{W}}) &= \mathrm{tr}\left(\mathbb{E}[\mathbf{z}'\boldsymbol{\omega}^\top - (\boldsymbol{\omega}^\top \mathbf{z}')\boldsymbol{\omega}\boldsymbol{\omega}^\top] - \rho \mathbf{W}\mathbf{W}^\top\right) \\
&\le \mathbb{E}\,\mathrm{tr}(\mathbf{z}'\boldsymbol{\omega}^\top) + \mathbb{E}\,\mathrm{tr}(\boldsymbol{\omega}\boldsymbol{\omega}^\top) + \rho \,\mathrm{tr}(\mathbf{W}\mathbf{W}^\top) \\
&\le 2 + \rho \,\mathrm{tr}(\mathbf{W}^\top \mathbf{W}),
\end{aligned}
$$

where the trace evaluation of rank-1 matrices and the symmetry of $\mathbf{W}$ are used. Hence, we obtain the following integral inequality:

$$\mathrm{tr}(\mathbf{W}^\top \mathbf{W}(t)) \le \mathrm{tr}(\mathbf{W}^\top \mathbf{W}(0)) + 4t + 2\rho \int_0^t \mathrm{tr}(\mathbf{W}^\top \mathbf{W}(\tau)) \mathrm{d}\tau,$$

which is the same form as the integral inequality in Eq. (13), and can be solved in the same way. $\square$

**Lemma 13.** *For any* $t$*,* $\left\|\boldsymbol{\Phi}^\top \boldsymbol{\Phi}(t)\mathbf{x}_0\right\|_2^2 \le \left(\left\|\boldsymbol{\Phi}^\top \boldsymbol{\Phi}(0)\mathbf{x}_0\right\|_2^2 + 4 \left\|\mathbf{x}_0\right\|_2^2 t\right) \exp(2\rho t)$.

*Proof of Lemma 13.* First, we obtain

$$\left\|\boldsymbol{\Phi}^\top \boldsymbol{\Phi}(t)\mathbf{x}_0\right\|_2^2 \le \left\|\boldsymbol{\Phi}^\top \boldsymbol{\Phi}(0)\mathbf{x}_0\right\|_2^2 + \int_0^t \left\|\dot{\boldsymbol{\Phi}}^\top \boldsymbol{\Phi}(\tau)\mathbf{x}_0\right\|_2^2 \mathrm{d}\tau + \int_0^t \left\|\boldsymbol{\Phi}^\top \dot{\boldsymbol{\Phi}}(\tau)\mathbf{x}_0\right\|_2^2 \mathrm{d}\tau,$$

which is obtained in the same manner as Eq. (10) (in the proof of Lemma 10). We substitute the dynamics (Lemma 1), or Eq. (11) in the proof of Lemma 10, and simplify $\left\|\dot{\boldsymbol{\Phi}}^\top \boldsymbol{\Phi}(\tau)\mathbf{x}_0\right\|_2^2$ as follows:

$$
\begin{aligned}
\left\|\dot{\boldsymbol{\Phi}}^\top \boldsymbol{\Phi}\mathbf{x}_0\right\|_2^2 &= \left\|\boldsymbol{\Phi}^\dagger \mathbf{W}^\dagger \mathbb{E}[\mathbf{z}'\boldsymbol{\omega}^\top - (\boldsymbol{\omega}^\top \mathbf{z}')\boldsymbol{\omega}\boldsymbol{\omega}^\top]\mathbf{W}\boldsymbol{\Phi}\mathbf{x}_0 - \rho \boldsymbol{\Phi}^\top \boldsymbol{\Phi}\mathbf{x}_0\right\|_2^2 \\
&\le \mathbb{E}\left\|\boldsymbol{\Phi}^\dagger \mathbf{W}^\dagger (\mathbf{z}'\boldsymbol{\omega}^\top)\mathbf{W}\boldsymbol{\Phi}\mathbf{x}_0\right\|_2^2 + \mathbb{E}\left\|\boldsymbol{\Phi}^\dagger \mathbf{W}^\dagger (\boldsymbol{\omega}\boldsymbol{\omega}^\top)\mathbf{W}\boldsymbol{\Phi}\mathbf{x}_0\right\|_2^2 + \rho \left\|\boldsymbol{\Phi}^\top \boldsymbol{\Phi}\mathbf{x}_0\right\|_2^2,
\end{aligned}
$$

where $\left|\boldsymbol{\omega}^\top \mathbf{z}'\right| \le 1$ is used. The first term is bounded as follows:

$$
\begin{aligned}
\left\|\boldsymbol{\Phi}^\dagger \mathbf{W}^\dagger (\mathbf{z}'\boldsymbol{\omega}^\top)\mathbf{W}\boldsymbol{\Phi}\mathbf{x}_0\right\|_2^2 &= \mathrm{tr}\left(\boldsymbol{\Phi}^\dagger \mathbf{W}^\dagger (\mathbf{z}'\boldsymbol{\omega}^\top)\mathbf{W}\boldsymbol{\Phi}\mathbf{x}_0 \mathbf{x}_0^\top \boldsymbol{\Phi}^\top \mathbf{W}^\top (\boldsymbol{\omega}(\mathbf{z}')^\top)(\mathbf{W}^\dagger)^\top (\boldsymbol{\Phi}^\dagger)^\top\right) \\
&\overset{(*)}{=} \mathrm{tr}\left((\mathbf{z}'\boldsymbol{\omega}^\top)\mathbf{W}\boldsymbol{\Phi}\mathbf{x}_0 \mathbf{x}_0^\top \boldsymbol{\Phi}^\top \mathbf{W}^\top (\boldsymbol{\omega}(\mathbf{z}')^\top)(\mathbf{W}^\dagger)^\top (\boldsymbol{\Phi}^\dagger)^\top \boldsymbol{\Phi}^\dagger \mathbf{W}^\dagger\right) \\
&\overset{(\natural)}{\le} \left|\mathrm{tr}\left(\mathbf{W}\boldsymbol{\Phi}\mathbf{x}_0 \mathbf{x}_0^\top \boldsymbol{\Phi}^\top \mathbf{W}^\top (\boldsymbol{\omega}(\mathbf{z}')^\top)(\mathbf{W}^\dagger)^\top (\boldsymbol{\Phi}^\dagger)^\top \boldsymbol{\Phi}^\dagger \mathbf{W}^\dagger\right)\right| \\
&\overset{(*)}{=} \left|\mathrm{tr}\left((\boldsymbol{\omega}(\mathbf{z}')^\top)(\mathbf{W}^\dagger)^\top (\boldsymbol{\Phi}^\dagger)^\top \boldsymbol{\Phi}^\dagger \mathbf{W}^\dagger \mathbf{W}\boldsymbol{\Phi}\mathbf{x}_0 \mathbf{x}_0^\top \boldsymbol{\Phi}^\top \mathbf{W}^\top\right)\right| \\
&\overset{(\natural)}{\le} \left|\mathrm{tr}\left((\mathbf{W}^\dagger)^\top (\boldsymbol{\Phi}^\dagger)^\top \boldsymbol{\Phi}^\dagger \mathbf{W}^\dagger \mathbf{W}\boldsymbol{\Phi}\mathbf{x}_0 \mathbf{x}_0^\top \boldsymbol{\Phi}^\top \mathbf{W}^\top\right)\right| \\
&\overset{(*)}{=} \left|\mathrm{tr}\left(\boldsymbol{\Phi}^\dagger \mathbf{W}^\dagger \mathbf{W}\boldsymbol{\Phi}\mathbf{x}_0 \mathbf{x}_0^\top\right)\right| \\
&\le \left|\mathrm{tr}\left(\boldsymbol{\Phi}^\dagger \mathbf{W}^\dagger \mathbf{W}\boldsymbol{\Phi}\right) \cdot \mathrm{tr}\left(\mathbf{x}_0 \mathbf{x}_0^\top\right)\right| \\
&\overset{(*)}{=} \left|\mathrm{tr}\left(\mathbf{x}_0 \mathbf{x}_0^\top\right)\right| \\
&= \left\|\mathbf{x}_0\right\|_2^2,
\end{aligned}
$$

where we use the trace cyclic property at (*), and the Cauchy-Schwartz inequality and the trace property $\left|\mathrm{tr}(\mathbf{z}\boldsymbol{\omega}^\top)\right| = \left|\boldsymbol{\omega}^\top \mathbf{z}\right| \le 1$ for $\mathbf{z}, \boldsymbol{\omega} \in \mathbb{S}^{h-1}$ at ($\natural$). Similarly, $\left\|\boldsymbol{\Phi}^\dagger \mathbf{W}^\dagger (\boldsymbol{\omega}\boldsymbol{\omega}^\top)\mathbf{W}\boldsymbol{\Phi}\mathbf{x}_0\right\|_2^2 \le \left\|\mathbf{x}_0\right\|_2^2$. Thus, we have $\left\|\dot{\boldsymbol{\Phi}}^\top \boldsymbol{\Phi}\mathbf{x}_0\right\|_2^2 \le 2 \left\|\mathbf{x}_0\right\|_2^2 + \rho \left\|\boldsymbol{\Phi}^\top \boldsymbol{\Phi}\mathbf{x}_0\right\|_2^2$. By doing the same algebra again, we have $\left\|\boldsymbol{\Phi}^\top \dot{\boldsymbol{\Phi}}\mathbf{x}_0\right\|_2^2 \le 2 \left\|\mathbf{x}_0\right\|_2^2 + \rho \left\|\boldsymbol{\Phi}^\top \boldsymbol{\Phi}\mathbf{x}_0\right\|_2^2$ as well. By combining them,

$$\left\|\boldsymbol{\Phi}^\top \boldsymbol{\Phi}(t)\mathbf{x}_0\right\|_2^2 \le \left\|\boldsymbol{\Phi}^\top \boldsymbol{\Phi}(0)\mathbf{x}_0\right\|_2^2 + 4 \left\|\mathbf{x}_0\right\|_2^2 t + 2\rho \int_0^t \left\|\boldsymbol{\Phi}^\top \boldsymbol{\Phi}(\tau)\mathbf{x}_0\right\|_2^2 \mathrm{d}\tau$$

holds, to which the Grönwall–Bellman inequality (Theorem 1) can be used, and we deduce $\left\|\boldsymbol{\Phi}^\top\boldsymbol{\Phi}(t)\mathbf{x}_0\right\|_2^2 \le (\left\|\boldsymbol{\Phi}^\top\boldsymbol{\Phi}(0)\mathbf{x}_0\right\|_2^2 + 4\left\|\mathbf{x}_0\right\|_2^2 t)\exp(2\rho t)$. $\qquad\square$

**Lemma 14.** *For $\mathbf{W}\in\mathbb{S}\mathrm{ym}_h$, for any $t$, the following bound holds:*

$$\left\|\boldsymbol{\Phi}^\top\mathbf{W}^\top\mathbf{W}\boldsymbol{\Phi}(t)\right\|_\mathrm{F} \le \left\{\left\|\boldsymbol{\Phi}^\top\mathbf{W}^\top\mathbf{W}\boldsymbol{\Phi}(0)\right\|_\mathrm{F} + \frac{16\rho te^{2\rho t} + (2\rho I_0 - 8)(e^{2\rho t}-1)}{\rho^2}\right\} e^{4\rho t},$$

*where $I_0 \coloneqq \mathrm{tr}(\mathbf{W}^\top\mathbf{W}(0)) + \left\|\boldsymbol{\Phi}^\top\boldsymbol{\Phi}(0)\right\|_\mathrm{F}$.*

*Proof of Lemma 14.* By using the fundamental theorem of calculus and the triangular inequality, the Frobenius norm $\left\|\boldsymbol{\Phi}^\top\mathbf{W}^\top\mathbf{W}\boldsymbol{\Phi}(t)\right\|_\mathrm{F}$ is bounded:

$$
\begin{aligned}
&\left\|\boldsymbol{\Phi}^\top\mathbf{W}^\top\mathbf{W}\boldsymbol{\Phi}(t)\right\|_\mathrm{F}\\
&\le \left\|\boldsymbol{\Phi}^\top\mathbf{W}^\top\mathbf{W}\boldsymbol{\Phi}(0)\right\|_\mathrm{F} + 2\int_0^t \left\|\frac{\mathrm{d}(\mathbf{W}\boldsymbol{\Phi})(\tau)}{\mathrm{d}\tau}^\top (\mathbf{W}\boldsymbol{\Phi})(\tau)\right\|_\mathrm{F}\mathrm{d}\tau\\
&\le \left\|\boldsymbol{\Phi}^\top\mathbf{W}^\top\mathbf{W}\boldsymbol{\Phi}(0)\right\|_\mathrm{F} + 2\underbrace{\int_0^t \left\|\dot{\boldsymbol{\Phi}}^\top\mathbf{W}^\top\mathbf{W}\boldsymbol{\Phi}(\tau)\right\|_\mathrm{F}\mathrm{d}\tau}_{(\mathrm{A})} + 2\underbrace{\int_0^t \left\|\boldsymbol{\Phi}^\top\dot{\mathbf{W}}^\top\mathbf{W}\boldsymbol{\Phi}(\tau)\right\|_\mathrm{F}\mathrm{d}\tau}_{(\mathrm{B})}.
\end{aligned}
\tag{14}
$$

To bound (A) in Eq. (14), we proceed by plugging the dynamics (Lemma 1) in as follows:

$$
\begin{aligned}
\left\|\dot{\boldsymbol{\Phi}}^\top\mathbf{W}^\top\mathbf{W}\boldsymbol{\Phi}\right\|_\mathrm{F} &= \left\|(\boldsymbol{\Phi}^\dagger\mathbf{W}^\dagger\,\mathbb{E}[\boldsymbol{\omega}(\mathbf{z}')^\top - (\boldsymbol{\omega}^\top\mathbf{z}')\boldsymbol{\omega}\boldsymbol{\omega}^\top]\mathbf{W} - \rho\boldsymbol{\Phi}^\top)\mathbf{W}^\top\mathbf{W}\boldsymbol{\Phi}\right\|_\mathrm{F}\\
&\le \underbrace{\mathbb{E}\left\|\boldsymbol{\Phi}^\dagger\mathbf{W}^\dagger(\boldsymbol{\omega}(\mathbf{z}')^\top)\mathbf{W}\mathbf{W}^\top\mathbf{W}\boldsymbol{\Phi}\right\|_\mathrm{F}}_{(\clubsuit)} + \underbrace{\mathbb{E}\left\|\boldsymbol{\Phi}^\dagger\mathbf{W}^\dagger(\boldsymbol{\omega}\boldsymbol{\omega}^\top)\mathbf{W}\mathbf{W}^\top\mathbf{W}\boldsymbol{\Phi}\right\|_\mathrm{F}}_{(\diamondsuit)}\\
&\quad + \rho\left\|\boldsymbol{\Phi}^\top\mathbf{W}^\top\mathbf{W}\boldsymbol{\Phi}\right\|_\mathrm{F}.
\end{aligned}
\tag{15}
$$

We bound the squared $(\clubsuit)$ in Eq. (15) as follows:

$$
\begin{aligned}
&\left\|\boldsymbol{\Phi}^\dagger\mathbf{W}^\dagger(\boldsymbol{\omega}(\mathbf{z}')^\top)\mathbf{W}\mathbf{W}^\top\mathbf{W}\boldsymbol{\Phi}\right\|_\mathrm{F}^2\\
&= \mathrm{tr}\left(\boldsymbol{\Phi}^\top\mathbf{W}^\top\mathbf{W}\mathbf{W}^\top(\mathbf{z}'\boldsymbol{\omega}^\top)(\mathbf{W}^\dagger)^\top(\boldsymbol{\Phi}^\dagger)^\top \cdot \boldsymbol{\Phi}^\dagger\mathbf{W}^\dagger(\boldsymbol{\omega}(\mathbf{z}')^\top)\mathbf{W}\mathbf{W}^\top\mathbf{W}\boldsymbol{\Phi}\right)\\
&\overset{(*\natural)}{\le} \left|\mathrm{tr}\left((\mathbf{W}^\dagger)^\top(\boldsymbol{\Phi}^\dagger)^\top\boldsymbol{\Phi}^\dagger\mathbf{W}^\dagger(\boldsymbol{\omega}(\mathbf{z}')^\top)\mathbf{W}\mathbf{W}^\top\mathbf{W}\boldsymbol{\Phi}\cdot\boldsymbol{\Phi}^\top\mathbf{W}^\top\mathbf{W}\mathbf{W}^\top\right)\right|\\
&\overset{(*)}{=} \left|\mathrm{tr}\left((\boldsymbol{\Phi}^\dagger)^\top\boldsymbol{\Phi}^\dagger\mathbf{W}^\dagger(\boldsymbol{\omega}(\mathbf{z}')^\top)\mathbf{W}\mathbf{W}^\top\mathbf{W}\boldsymbol{\Phi}\boldsymbol{\Phi}^\top\mathbf{W}^\top\mathbf{W}\right)\right|\\
&\overset{(*\natural)}{\le} \left|\mathrm{tr}\left(\mathbf{W}\mathbf{W}^\top\mathbf{W}\boldsymbol{\Phi}\boldsymbol{\Phi}^\top\mathbf{W}^\top\mathbf{W}\cdot(\boldsymbol{\Phi}^\dagger)^\top\boldsymbol{\Phi}^\dagger\mathbf{W}^\dagger\right)\right|\\
&\overset{(*)}{=} \left|\mathrm{tr}\left(\boldsymbol{\Phi}^\dagger\mathbf{W}^\top\mathbf{W}\boldsymbol{\Phi}\cdot\boldsymbol{\Phi}^\top\mathbf{W}^\top\mathbf{W}(\boldsymbol{\Phi}^\dagger)^\top\right)\right|\\
&\le \left|\mathrm{tr}(\boldsymbol{\Phi}^\dagger\mathbf{W}^\top\mathbf{W}\boldsymbol{\Phi})\cdot\mathrm{tr}(\boldsymbol{\Phi}^\top\mathbf{W}^\top\mathbf{W}(\boldsymbol{\Phi}^\dagger)^\top)\right|\\
&\overset{(*)}{=} \mathrm{tr}(\mathbf{W}^\top\mathbf{W})^2,
\end{aligned}
$$

where we use the trace cyclic property at (*), and use the trace cyclic property, the Cauchy-Schwartz inequality, and the trace evaluation of rank-1 matrices at (*♮), as we do in the proof of Lemma 13. By using the same techniques, the squared $(\diamondsuit)$ in Eq. (15) can be bounded by $\mathrm{tr}(\mathbf{W}^\top\mathbf{W})$ as well. Hence, we obtain the bound of Eq. (15) as $\left\|\dot{\boldsymbol{\Phi}}^\top\mathbf{W}^\top\mathbf{W}\boldsymbol{\Phi}\right\|_\mathrm{F} \le 2\,\mathrm{tr}(\mathbf{W}^\top\mathbf{W}) + \rho\left\|\boldsymbol{\Phi}^\top\mathbf{W}^\top\mathbf{W}\boldsymbol{\Phi}\right\|_\mathrm{F}$. To bound (B) in Eq. (14), the dynamics (Lemma 1) is plugged in again:

$$
\begin{aligned}
\left\|\boldsymbol{\Phi}^\top\dot{\mathbf{W}}^\top\mathbf{W}\boldsymbol{\Phi}\right\|_\mathrm{F} &= \left\|\boldsymbol{\Phi}^\top\,\mathbb{E}[\boldsymbol{\omega}(\mathbf{z}')^\top - (\boldsymbol{\omega}^\top\mathbf{z}')\boldsymbol{\omega}\boldsymbol{\omega}^\top]\boldsymbol{\Phi} - \rho\boldsymbol{\Phi}^\top\mathbf{W}\mathbf{W}^\top\boldsymbol{\Phi}^\top\right\|\\
&\le \underbrace{\mathbb{E}\left\|\boldsymbol{\Phi}^\top(\boldsymbol{\omega}(\mathbf{z}')^\top)\boldsymbol{\Phi}\right\|_\mathrm{F}}_{(\heartsuit)} + \underbrace{\mathbb{E}\left\|\boldsymbol{\Phi}^\top(\boldsymbol{\omega}\boldsymbol{\omega}^\top)\boldsymbol{\Phi}\right\|_\mathrm{F}}_{(\spadesuit)} + \rho\left\|\boldsymbol{\Phi}^\top\mathbf{W}\mathbf{W}^\top\boldsymbol{\Phi}\right\|_\mathrm{F},
\end{aligned}
\tag{16}
$$

where the squared ($\heartsuit$) is bounded as follows:

$$
\begin{aligned}
\left\| \boldsymbol{\Phi}^{\top}(\boldsymbol{\omega}(\mathbf{z}')^{\top})\boldsymbol{\Phi} \right\|_{\mathrm{F}}^{2} &= \mathrm{tr}\left( \boldsymbol{\Phi}^{\top}(\mathbf{z}'\boldsymbol{\omega}^{\top})\boldsymbol{\Phi} \cdot \boldsymbol{\Phi}^{\top}(\boldsymbol{\omega}(\mathbf{z}')^{\top})\boldsymbol{\Phi} \right) \\
&\overset{(*\natural)}{\leq} \left| \mathrm{tr}\left( \boldsymbol{\Phi}\boldsymbol{\Phi}^{\top}(\boldsymbol{\omega}(\mathbf{z}')^{\top})\boldsymbol{\Phi}\boldsymbol{\Phi}^{\top} \right) \right| \\
&\overset{(*\natural)}{\leq} \left| \mathrm{tr}\left( \boldsymbol{\Phi}\boldsymbol{\Phi}^{\top}\boldsymbol{\Phi}\boldsymbol{\Phi}^{\top} \right) \right| \\
&= \left\| \boldsymbol{\Phi}\boldsymbol{\Phi}^{\top} \right\|_{\mathrm{F}}^{2} \\
&= \left\| \boldsymbol{\Phi}^{\top}\boldsymbol{\Phi} \right\|_{\mathrm{F}}^{2}.
\end{aligned}
$$

The squared ($\spadesuit$) is bounded by $\left\| \boldsymbol{\Phi}^{\top}\boldsymbol{\Phi} \right\|_{\mathrm{F}}$ as well. Hence, we obtain the bound of Eq. (16) as $\left\| \boldsymbol{\Phi}^{\top}\dot{\mathbf{W}}^{\top}\mathbf{W}\boldsymbol{\Phi} \right\|_{\mathrm{F}} \leq 2\left\| \boldsymbol{\Phi}^{\top}\boldsymbol{\Phi} \right\|_{\mathrm{F}} + \rho\left\| \boldsymbol{\Phi}^{\top}\mathbf{W}\mathbf{W}^{\top}\boldsymbol{\Phi} \right\|_{\mathrm{F}}$. Eventually, we obtain the following bound from Eq. (14) (which requires the symmetry of $\mathbf{W}$):

$$
\begin{aligned}
\left\| \boldsymbol{\Phi}^{\top}\mathbf{W}^{\top}\mathbf{W}\boldsymbol{\Phi}(t) \right\|_{\mathrm{F}} &\leq \left\| \boldsymbol{\Phi}^{\top}\mathbf{W}^{\top}\mathbf{W}\boldsymbol{\Phi}(0) \right\|_{\mathrm{F}} + 4\int_{0}^{t} \mathrm{tr}(\mathbf{W}^{\top}\mathbf{W}(\tau))\mathrm{d}\tau \\
&\quad + 4\int_{0}^{t} \left\| \boldsymbol{\Phi}^{\top}\boldsymbol{\Phi}(\tau) \right\|_{\mathrm{F}} \mathrm{d}\tau + 4\int_{0}^{t} \rho\left\| \boldsymbol{\Phi}^{\top}\mathbf{W}^{\top}\mathbf{W}\boldsymbol{\Phi}(\tau) \right\|_{\mathrm{F}} \mathrm{d}\tau \\
&\leq \left\| \boldsymbol{\Phi}^{\top}\mathbf{W}^{\top}\mathbf{W}\boldsymbol{\Phi}(0) \right\|_{\mathrm{F}} + 4\int_{0}^{t} \rho\left\| \boldsymbol{\Phi}^{\top}\mathbf{W}^{\top}\mathbf{W}\boldsymbol{\Phi}(\tau) \right\|_{\mathrm{F}} \mathrm{d}\tau \\
&\quad + 4\int_{0}^{t} \left\{ I_0 \exp(2\rho\tau) + 8\tau\exp(2\rho\tau) \right\} \mathrm{d}\tau \\
&\leq \left\| \boldsymbol{\Phi}^{\top}\mathbf{W}^{\top}\mathbf{W}\boldsymbol{\Phi}(0) \right\|_{\mathrm{F}} + 4\int_{0}^{t} \rho\left\| \boldsymbol{\Phi}^{\top}\mathbf{W}^{\top}\mathbf{W}\boldsymbol{\Phi}(\tau) \right\|_{\mathrm{F}} \mathrm{d}\tau \\
&\quad + \frac{16\rho t e^{2\rho t} + (2\rho I_0 - 8)(e^{2\rho t} - 1)}{\rho^2},
\end{aligned}
$$

where Lemmas 10 and 12 are used at the second inequality and integration by parts is used in the third inequality. This integral inequality can be solved by the Grönwall–Bellman inequality (Theorem 1), and we can obtain the conclusion. $\qquad\square$

**Lemma 15.** *For* $\mathbf{W} \in \mathbb{S}\mathrm{ym}_h$, *for any* $t$, *the following bound holds:*

$$
\|\mathbf{W}\boldsymbol{\Phi}(t)\| \leq \sqrt{\left\{ \left\| \boldsymbol{\Phi}^{\top}\mathbf{W}^{\top}\mathbf{W}\boldsymbol{\Phi}(0) \right\| + \frac{16\rho t e^{2\rho t} + (2\rho I_0 - 8)(e^{2\rho t} - 1)}{\rho^2} \right\} e^{4\rho t}},
$$

*where* $I_0$ *is defined in Lemma 14.*

*Proof of Lemma 15.* We evaluate $\left\| \boldsymbol{\Phi}^{\top}\mathbf{W}^{\top}\mathbf{W}\boldsymbol{\Phi}(t) \right\| = \left\| \mathbf{W}\boldsymbol{\Phi}(t) \right\|^2$. By the fundamental theorem of calculus, we obtain the following decomposition:

$$
\left\| \boldsymbol{\Phi}^{\top}\mathbf{W}^{\top}\mathbf{W}\boldsymbol{\Phi}(t) \right\| \leq \left\| \boldsymbol{\Phi}^{\top}\mathbf{W}^{\top}\mathbf{W}\boldsymbol{\Phi}(0) \right\| + 2\int_{0}^{t} \left\| \left(\frac{\mathrm{d}\mathbf{W}\boldsymbol{\Phi}}{\mathrm{d}\tau}\right)^{\top}\mathbf{W}\boldsymbol{\Phi}(\tau) \right\| \mathrm{d}\tau.
$$

By following the same derivation as the proof of Lemma 14, it is not difficult to see the following upper bound:

$$
\left\| \left(\frac{\mathrm{d}\mathbf{W}\boldsymbol{\Phi}}{\mathrm{d}\tau}\right)^{\top}\mathbf{W}\boldsymbol{\Phi} \right\| \leq 2\,\mathrm{tr}(\mathbf{W}^{\top}\mathbf{W}) + 2\left\| \boldsymbol{\Phi}^{\top}\boldsymbol{\Phi} \right\|_{\mathrm{F}} + 2\rho\left\| \boldsymbol{\Phi}^{\top}\mathbf{W}^{\top}\mathbf{W}\boldsymbol{\Phi} \right\|.
$$

By plugging the results of Lemmas 10 and 12 into $\mathrm{tr}(\mathbf{W}^{\top}\mathbf{W}(\tau))$ and $\left\| \boldsymbol{\Phi}^{\top}\boldsymbol{\Phi}(\tau) \right\|_{\mathrm{F}}$, we obtain the integral inequality:

$$
\begin{aligned}
\left\| \boldsymbol{\Phi}^{\top}\mathbf{W}^{\top}\mathbf{W}\boldsymbol{\Phi}(t) \right\| &\leq \left\| \boldsymbol{\Phi}^{\top}\mathbf{W}^{\top}\mathbf{W}\boldsymbol{\Phi}(0) \right\| + 4\rho\int_{0}^{t} \left\| \boldsymbol{\Phi}^{\top}\mathbf{W}^{\top}\mathbf{W}\boldsymbol{\Phi}(\tau) \right\| \mathrm{d}\tau \\
&\quad + \frac{16\rho t e^{2\rho t} + (2\rho I_0 - 8)(e^{2\rho t} - 1)}{\rho^2}.
\end{aligned}
$$

This can be solved via Theorem 1. $\qquad\square$

**Lemma 16.** *For $\mathbf{W} \in \mathbb{S}\mathrm{ym}_h$, for any $t$, the following bound holds:*

$$\left\|\mathbf{\Phi}^\top \mathbf{W}^\top \mathbf{W}\mathbf{\Phi}(t)\mathbf{x}_0\right\|_2^2$$
$$\leq \left\{\left\|\mathbf{\Phi}^\top \mathbf{W}^\top \mathbf{W}\mathbf{\Phi}(0)\mathbf{x}_0\right\|_2^2 + \Xi_1 \left\|\mathbf{x}_0\right\|_2^2 + \Xi_2 \left\|\mathbf{\Phi}^\top \mathbf{\Phi}(0)\mathbf{x}_0\right\|_2^2\right\} \exp(2\rho t),$$

*where*

$$\Xi_1 := T_0^2 \frac{e^{4\rho t}-1}{\rho} + 2T_0 \frac{e^{4\rho t}(4\rho t - 1)+1}{\rho^2} + 2\frac{e^{4\rho t}(8\rho^2 t^2 - 4\rho t + 1)-1}{\rho^3} + 4\frac{e^{2\rho t}(2\rho t - 1)+1}{\rho^2},$$

$$\Xi_2 := 2\frac{e^{2\rho t}-1}{\rho},$$

*and $T_0 := \mathrm{tr}(\mathbf{W}^\top \mathbf{W}(0))$.*

*Proof of Lemma 16.* By using the fundamental theorem of calculus, $\left\|\mathbf{\Phi}^\top \mathbf{W}^\top \mathbf{W}\mathbf{\Phi}(t)\mathbf{x}_0\right\|_2^2$ is bounded as follows:

$$\left\|\mathbf{\Phi}^\top \mathbf{W}^\top \mathbf{W}\mathbf{\Phi}(t)\mathbf{x}_0\right\|_2^2$$
$$\leq \left\|\mathbf{\Phi}^\top \mathbf{W}^\top \mathbf{W}\mathbf{\Phi}(0)\mathbf{x}_0\right\|_2^2 + 2\int_0^t \left\|\left(\frac{\mathrm{d}\mathbf{W}\mathbf{\Phi}}{\mathrm{d}\tau}\right)^\top \mathbf{W}\mathbf{\Phi}(\tau)\mathbf{x}_0\right\|_2^2 \mathrm{d}\tau$$
$$\leq \left\|\mathbf{\Phi}^\top \mathbf{W}^\top \mathbf{W}\mathbf{\Phi}(0)\mathbf{x}_0\right\|_2^2 \tag{17}$$
$$+ \underbrace{2\int_0^t \left\|\dot{\mathbf{\Phi}}^\top \mathbf{W}^\top \mathbf{W}\mathbf{\Phi}(\tau)\mathbf{x}_0\right\|_2^2 \mathrm{d}\tau}_{(A)} + \underbrace{2\int_0^t \left\|\mathbf{\Phi}^\top \dot{\mathbf{W}}^\top \mathbf{W}\mathbf{\Phi}(\tau)\mathbf{x}_0\right\|_2^2 \mathrm{d}\tau}_{(B)}.$$

To bound (A) in Eq. (17), we follow almost the same calculation as Eq. (15) in the proof of Lemma 13 (therefore omitted) and obtain $\left\|\dot{\mathbf{\Phi}}^\top \mathbf{W}^\top \mathbf{W}\mathbf{\Phi}\mathbf{x}_0\right\|_2^2 \leq \mathrm{tr}(\mathbf{W}^\top \mathbf{W})^2 \left\|\mathbf{x}_0\right\|_2^2$. To bound (B) in Eq. (17), we follow almost the same calculation as Eq. (16) in the proof of Lemma 13 (therefore omitted) and obtain $\left\|\mathbf{\Phi}^\top \dot{\mathbf{W}}^\top \mathbf{W}\mathbf{\Phi}\mathbf{x}_0\right\|_2^2 \leq 2\left\|\mathbf{\Phi}^\top \mathbf{\Phi}\mathbf{x}_0\right\|_2^2 + \rho \left\|\mathbf{\Phi}^\top \mathbf{W}^\top \mathbf{W}\mathbf{\Phi}\mathbf{x}_0\right\|_2^2$. Here, the symmetry of $\mathbf{W}$ is used. By substituting them back into (A) and (B) in Eq. (17), we obtain the following bound:

$$\left\|\mathbf{\Phi}^\top \mathbf{W}^\top \mathbf{W}\mathbf{\Phi}(t)\mathbf{x}_0\right\|_2^2 \leq \left\|\mathbf{\Phi}^\top \mathbf{W}^\top \mathbf{W}\mathbf{\Phi}(0)\mathbf{x}_0\right\|_2^2 + 2\rho \int_0^t \left\|\mathbf{\Phi}^\top \mathbf{W}^\top \mathbf{W}\mathbf{\Phi}(\tau)\mathbf{x}_0\right\|_2^2 \mathrm{d}\tau$$
$$+ 4\left\|\mathbf{x}_0\right\|_2^2 \underbrace{\int_0^t \mathrm{tr}(\mathbf{W}^\top \mathbf{W}(\tau))^2 \mathrm{d}\tau}_{(\clubsuit)} + 4\underbrace{\int_0^t \left\|\mathbf{\Phi}^\top \mathbf{\Phi}(\tau)\mathbf{x}_0\right\|_2^2 \mathrm{d}\tau}_{(\diamondsuit)}.$$

The term ($\clubsuit$) can be evaluated by Lemma 12 and integration by parts as follows:

$$(\clubsuit) \leq \int_0^t (T_0 + 4\tau)^2 \exp(4\rho\tau)\mathrm{d}\tau$$
$$= \int_0^t (T_0^2 + 8T_0\tau + 16\tau^2)\exp(4\rho\tau)\mathrm{d}\tau$$
$$= T_0^2 \frac{e^{4\rho t}-1}{4\rho} + T_0 \frac{e^{4\rho t}(4\rho t - 1)+1}{2\rho^2} + \frac{e^{4\rho t}(8\rho^2 t^2 - 4\rho t + 1)-1}{2\rho^3},$$

The term ($\diamondsuit$) can be evaluated by Lemma 13 and integration by parts as follows:

$$(\diamondsuit) \leq \int_0^t \left\{\left\|\mathbf{\Phi}^\top \mathbf{\Phi}(0)\mathbf{x}_0\right\|_2^2 + 4\left\|\mathbf{x}_0\right\|_2^2 \tau\right\} e^{2\rho\tau}\mathrm{d}\tau$$
$$= \left\|\mathbf{\Phi}^\top \mathbf{\Phi}(0)\mathbf{x}_0\right\|_2^2 \frac{e^{2\rho t}-1}{2\rho} + \left\|\mathbf{x}_0\right\|_2^2 \frac{e^{2\rho t}(2\rho t - 1)+1}{\rho^2}.$$

Hence, we obtain the following integral inequality:

$$\left\|\mathbf{\Phi}^\top\mathbf{W}^\top\mathbf{W}\mathbf{\Phi}(t)\mathbf{x}_0\right\|_2^2 \le \left\|\mathbf{\Phi}^\top\mathbf{W}^\top\mathbf{W}\mathbf{\Phi}(0)\mathbf{x}_0\right\|_2^2 + \Xi_1\left\|\mathbf{x}_0\right\|_2^2 + \Xi_2\left\|\mathbf{\Phi}^\top\mathbf{\Phi}(0)\mathbf{x}_0\right\|_2^2$$
$$+ 2\rho\int_0^t\left\|\mathbf{\Phi}^\top\mathbf{W}^\top\mathbf{W}\mathbf{\Phi}(\tau)\mathbf{x}_0\right\|_2^2\mathrm{d}\tau,$$

which can be solved by the Grönwall–Bellman inequality (Theorem 1). As a result, the desired bound on $\left\|\mathbf{\Phi}^\top\mathbf{W}^\top\mathbf{W}\mathbf{\Phi}(t)\mathbf{x}_0\right\|_2^2$ can be obtained. $\qquad\square$

## B  MISSING PROOFS

**Lemma 1.** *Parameter matrices $\mathbf{W}$ and $\mathbf{\Phi}$ evolve as follows:*

$$\mathbf{W}^\top\dot{\mathbf{W}} = \mathbf{H} - \rho\mathbf{W}\mathbf{W}^\top,$$
$$\dot{\mathbf{\Phi}}\mathbf{\Phi}^\top\mathbf{W}^\top = \mathbf{W}^\top\mathbf{H} - \rho\mathbf{\Phi}\mathbf{\Phi}^\top\mathbf{W}^\top,\tag{4}$$

*where $\mathbf{H} := \mathbb{E}[\mathbf{z}'\boldsymbol{\omega}^\top - (\boldsymbol{\omega}^\top\mathbf{z}')\boldsymbol{\omega}\boldsymbol{\omega}^\top]$, $\mathbf{z} := \mathbf{\Phi}\mathbf{x}'/\|\mathbf{\Phi}\mathbf{x}'\|_2$, and $\boldsymbol{\omega} := \mathbf{W}\mathbf{\Phi}\mathbf{x}/\|\mathbf{W}\mathbf{\Phi}\mathbf{x}\|_2$. The expectation in $\mathbf{H}$ is taken over $\mathbf{x}_0, \mathbf{x}$, and $\mathbf{x}'$.*

*Proof of Lemma 1.* To derive the $\mathbf{W}$-dynamics, we begin with calculating the gradient $\nabla_{\mathbf{W}}\mathcal{L}_{\cos}$.

$$-\nabla_{\mathbf{W}}\mathcal{L}_{\cos} = \mathbb{E}\left[\frac{1}{\|\mathbf{\Phi}\mathbf{x}'\|_2}\frac{\|\mathbf{W}\mathbf{\Phi}\mathbf{x}\|_2\nabla_{\mathbf{W}}(\mathbf{x}^\top\mathbf{\Phi}^\top\mathbf{W}^\top\mathbf{\Phi}\mathbf{x}') - \mathbf{x}^\top\mathbf{\Phi}^\top\mathbf{W}^\top\mathbf{\Phi}\mathbf{x}'\nabla_{\mathbf{W}}\|\mathbf{W}\mathbf{\Phi}\mathbf{x}\|_2}{\|\mathbf{W}\mathbf{\Phi}\mathbf{x}\|_2^2}\right]$$

$$= \mathbb{E}\left[\frac{\nabla_{\mathbf{W}}(\mathbf{x}^\top\mathbf{\Phi}^\top\mathbf{W}^\top\mathbf{z}') - \boldsymbol{\omega}^\top\mathbf{z}'\nabla_{\mathbf{W}}\|\mathbf{W}\mathbf{\Phi}\mathbf{x}\|_2}{\|\mathbf{W}\mathbf{\Phi}\mathbf{x}\|_2}\right]$$

$$= \mathbb{E}\left[\frac{\mathbf{z}'\mathbf{x}^\top\mathbf{\Phi}^\top - (\boldsymbol{\omega}^\top\mathbf{z}')\frac{\mathbf{W}\mathbf{\Phi}\mathbf{x}\mathbf{x}^\top\mathbf{\Phi}^\top}{\|\mathbf{W}\mathbf{\Phi}\mathbf{x}\|_2}}{\|\mathbf{W}\mathbf{\Phi}\mathbf{x}\|_2}\right]$$

$$= \mathbb{E}\left[\mathbf{z}'\frac{\mathbf{x}^\top\mathbf{\Phi}^\top}{\|\mathbf{W}\mathbf{\Phi}\mathbf{x}\|_2} - (\boldsymbol{\omega}^\top\mathbf{z}')\boldsymbol{\omega}\frac{\mathbf{x}^\top\mathbf{\Phi}^\top}{\|\mathbf{W}\mathbf{\Phi}\mathbf{x}\|_2}\right].$$

Here, $\mathbf{W}$ follows the dynamics $\dot{\mathbf{W}} = -\nabla_{\mathbf{W}}\mathcal{L}_{\cos} - \rho\mathbf{W}$, and hence we obtain $\dot{\mathbf{W}}\mathbf{W}^\top = \mathbb{E}[\mathbf{z}'\boldsymbol{\omega}^\top - (\boldsymbol{\omega}^\top\mathbf{z}')\boldsymbol{\omega}\boldsymbol{\omega}^\top] - \rho\mathbf{W}\mathbf{W}^\top$.

To derive the $\mathbf{\Phi}$-dynamics, we calculate the gradient $\nabla_{\mathbf{\Phi}}\mathcal{L}_{\cos}$.

$$-\nabla_{\mathbf{\Phi}}\mathcal{L}_{\cos}$$

$$= \mathbb{E}\left[\frac{1}{\|\mathbf{\Phi}\mathbf{x}'\|_2}\frac{\|\mathbf{W}\mathbf{\Phi}\mathbf{x}\|_2\nabla_{\mathbf{\Phi}}(\mathbf{x}^\top\mathbf{\Phi}^\top\mathbf{W}^\top\mathrm{StopGrad}(\mathbf{\Phi})\mathbf{x}') - \mathbf{x}^\top\mathbf{\Phi}^\top\mathbf{W}^\top\mathbf{\Phi}\mathbf{x}'\nabla_{\mathbf{\Phi}}\|\mathbf{W}\mathbf{\Phi}\mathbf{x}\|_2}{\|\mathbf{W}\mathbf{\Phi}\mathbf{x}\|_2^2}\right]$$

$$= \mathbb{E}\left[\frac{1}{\|\mathbf{\Phi}\mathbf{x}'\|_2}\frac{\|\mathbf{W}\mathbf{\Phi}\mathbf{x}\|_2\mathbf{W}^\top\mathbf{\Phi}\mathbf{x}'\mathbf{x}^\top - \mathbf{x}^\top\mathbf{\Phi}^\top\mathbf{W}^\top\mathbf{\Phi}\mathbf{x}'\frac{\mathbf{W}^\top\mathbf{W}\mathbf{\Phi}\mathbf{x}\mathbf{x}^\top}{\|\mathbf{W}\mathbf{\Phi}\mathbf{x}\|_2}}{\|\mathbf{W}\mathbf{\Phi}\mathbf{x}\|_2^2}\right]$$

$$= \mathbf{W}^\top\mathbb{E}\left[\frac{\mathbf{z}'\mathbf{x}^\top - (\boldsymbol{\omega}^\top\mathbf{z}')\boldsymbol{\omega}\mathbf{x}^\top}{\|\mathbf{W}\mathbf{\Phi}\mathbf{x}\|_2}\right],$$

from which $(-\nabla_{\mathbf{\Phi}}\mathcal{L}_{\cos})\mathbf{\Phi}^\top\mathbf{W}^\top = \mathbf{W}^\top\mathbb{E}[\mathbf{z}'\boldsymbol{\omega}^\top - (\boldsymbol{\omega}^\top\mathbf{z}')\boldsymbol{\omega}\boldsymbol{\omega}^\top]$ follows. Thus, the dynamics $\dot{\mathbf{\Phi}} = -\nabla_{\mathbf{\Phi}}\mathcal{L}_{\cos} - \rho\mathbf{\Phi}$ can be written as $\dot{\mathbf{\Phi}}\mathbf{\Phi}^\top\mathbf{W}^\top = \mathbf{W}^\top\mathbb{E}[\mathbf{z}'\boldsymbol{\omega}^\top - (\boldsymbol{\omega}^\top\mathbf{z}')\boldsymbol{\omega}\boldsymbol{\omega}^\top] - \rho\mathbf{\Phi}\mathbf{\Phi}^\top\mathbf{W}^\top$. $\quad\square$

**Lemma 2.** *Under Assumptions 1 to 4, for a fixed $\mathbf{x}_0$, the norms of $\mathbf{\Phi}\mathbf{x}$ and $\mathbf{W}\mathbf{\Phi}\mathbf{x}$ (as well as $\mathbf{\Phi}\mathbf{x}'$ and $\mathbf{W}\mathbf{\Phi}\mathbf{x}'$) are concentrated:*

$$\left\|\frac{1}{\sqrt{h\sigma^2}}\mathbf{\Phi}\mathbf{x}\right\|_2^2 = \left\|\frac{1}{\sqrt{h}}\mathbf{\Phi}\right\|_{\mathrm{F}}^2 + \left\|\frac{1}{\sqrt{h\sigma^2}}\mathbf{\Phi}\mathbf{x}_0\right\|_2^2 + o_{\mathbb{P}}(1),$$

$$\left\|\frac{1}{\sqrt{h^2\sigma^2}}\mathbf{W}\mathbf{\Phi}\mathbf{x}\right\|_2^2 = \left\|\frac{1}{\sqrt{h^2}}\mathbf{W}\mathbf{\Phi}\right\|_{\mathrm{F}}^2 + \left\|\frac{1}{\sqrt{h^2\sigma^2}}\mathbf{W}\mathbf{\Phi}\mathbf{x}_0\right\|_2^2 + o_{\mathbb{P}}(1).$$

*Proof of Lemma 2.* We will show concentration of $\left\|\frac{1}{\sqrt{h\sigma^2}}\mathbf{\Phi}\mathbf{x}\right\|_2^2$ and $\left\|\frac{1}{\sqrt{h^2\sigma^2}}\mathbf{W}\mathbf{\Phi}\mathbf{x}\right\|_2^2$:

**Concentration of** $\|\mathbf{\Phi}\mathbf{x}\|_2^2$ : We begin with showing the first concentration.

$$\left\|\frac{1}{\sqrt{h\sigma^2}}\mathbf{\Phi}\mathbf{x}\right\|_2^2 = \left\|\frac{1}{\sqrt{h}}\left(\mathbf{\Phi}\frac{\mathbf{x}-\mathbf{x}_0}{\sigma}+\mathbf{\Phi}\frac{\mathbf{x}_0}{\sigma}\right)\right\|_2^2$$

$$= \underbrace{\left\|\frac{1}{\sqrt{h}}\mathbf{\Phi}\frac{\mathbf{x}-\mathbf{x}_0}{\sigma}\right\|_2^2}_{(A)} + 2\sigma^{-1}\underbrace{\left\langle\frac{1}{\sqrt{h}}\mathbf{\Phi}\frac{\mathbf{x}-\mathbf{x}_0}{\sigma},\frac{1}{\sqrt{h}}\mathbf{\Phi}\mathbf{x}_0\right\rangle}_{(B)} + \left\|\frac{1}{\sqrt{h}}\mathbf{\Phi}\frac{\mathbf{x}_0}{\sigma}\right\|_2^2. \quad (18)$$

To deal with (A), which is a Gaussian chaos (namely, a quadratic form with standard normal vectors), we invoke the Hanson–Wright inequality [Ver18, Theorem 6.3.2]. Note that $\frac{\mathbf{x}-\mathbf{x}_0}{\sigma}$ follows the standard normal distribution. Then, the following inequality holds with probability at least $1-\delta$ (over the sampling of $\mathbf{x}$):

$$\left|\left\|\frac{1}{\sqrt{h}}\mathbf{\Phi}\frac{\mathbf{x}-\mathbf{x}_0}{\sigma}\right\|_2 - \left\|\frac{1}{\sqrt{h}}\mathbf{\Phi}\right\|_F\right| \le \sqrt{\frac{C_0\|\mathbf{\Phi}\|^2\log\frac{2}{\delta}}{h}}, \quad (19)$$

where the expectation is taken over $\mathbf{x}\sim\mathcal{N}(\mathbf{x}_0,\sigma^2\mathbf{I}_d)$, and $C_0$ is an absolute constant irrelevant to $d$ and $h$. Now, we evaluate the deviation term and show it vanishes as $d,h\to\infty$. Since the deviation term contains $\|\mathbf{\Phi}\|^2$ and it depends on the time $t$, we need to carefully evaluate its order in $d$ and $h$ along with time evolution. For this purpose, Lemma 11 is used to obtain $\|\mathbf{\Phi}(t)\|^2 \le (\|\mathbf{\Phi}^\top\mathbf{\Phi}(0)\|+4t)\exp(2\rho t)$. Lastly, the Gaussian initialization of $\mathbf{\Phi}$ (Assumption 4) induces $\frac{1}{h}\|\mathbf{\Phi}^\top\mathbf{\Phi}(0)\| = o_{\mathbb{P}}(1)$ (by Lemma 7). Thus, the deviation term of Eq. (19) is bounded from above as follows:

$$\sqrt{\frac{C_0(\|\mathbf{\Phi}^\top\mathbf{\Phi}(0)\|+4t)\exp(2\rho t)\log\frac{2}{\delta}}{h}} = o_{\mathbb{P}}(1),$$

from which we conclude as follows:

$$\left\|\frac{1}{\sqrt{h}}\mathbf{\Phi}\frac{\mathbf{x}-\mathbf{x}_0}{\sigma}\right\|_2^2 = \left\|\frac{1}{\sqrt{h}}\mathbf{\Phi}\right\|_F^2 + o_{\mathbb{P}}(1).$$

Next, we deal with (B) in Eq. (18). The term (B) is equivalent to $\left\langle\frac{1}{h}\mathbf{\Phi}^\top\mathbf{\Phi}\mathbf{x}_0,\frac{\mathbf{x}-\mathbf{x}_0}{\sigma}\right\rangle$, which is a linear combination of the standard normal random variables. Its concentration (to mean 0) can be established by the general Hoeffding's inequality [Ver18, Theorem 2.6.3] as follows: With probability at least $1-\delta$ (over the sampling of $\mathbf{x}$),

$$(B) = \left|\left\langle\frac{1}{h}\mathbf{\Phi}^\top\mathbf{\Phi}\mathbf{x}_0,\frac{\mathbf{x}-\mathbf{x}_0}{\sigma}\right\rangle\right| \le \sqrt{\frac{C_1\|\mathbf{\Phi}^\top\mathbf{\Phi}\mathbf{x}_0\|_2^2\log\frac{2}{\delta}}{h^2}}, \quad (20)$$

where $C_1$ is an absolute constant irrelevant to $d$ and $h$. We need to evaluate $\|\mathbf{\Phi}^\top\mathbf{\Phi}(t)\mathbf{x}_0\|_2^2$ by noting its time dependency again. For this purpose, Lemma 13 is used to obtain $\|\mathbf{\Phi}^\top\mathbf{\Phi}(t)\mathbf{x}_0\|_2^2 \le (\|\mathbf{\Phi}^\top\mathbf{\Phi}(0)\mathbf{x}_0\|_2^2 + 4\|\mathbf{x}_0\|_2^2 t)\exp(2\rho t)$. Here, $\frac{1}{h^2}\|\mathbf{\Phi}^\top\mathbf{\Phi}(0)\mathbf{x}_0\|_2^2 = o_{\mathbb{P}}(1)$ (Lemma 9) holds. In addition, $\mathbf{x}_0\sim\mathcal{N}(\mathbf{0},\mathbf{I})$ (Assumption 2) indicates that $\|\mathbf{x}_0\|_2^2$ is the sum of independent zero-mean sub-exponential random variables, from which Bernstein's inequality claim $\|\mathbf{x}_0\|_2^2 = \mathcal{O}_{\mathbb{P}}(d)$ [Ver18, Corollary 2.8.3]. Plugging them into the upper bound of $\|\mathbf{\Phi}^\top\mathbf{\Phi}(t)\mathbf{x}_0\|_2^2$, we deduce

$$(B) \le \sqrt{C_1\log\frac{2}{\delta}\left(\frac{\|\mathbf{\Phi}^\top\mathbf{\Phi}(0)\mathbf{x}_0\|_2^2}{h^2}+4t\frac{\|\mathbf{x}_0\|_2^2}{h^2}\right)e^{2\rho t}} = \sqrt{o_{\mathbb{P}}(1)+\mathcal{O}_{\mathbb{P}}(\alpha h^{-1})} = o_{\mathbb{P}}(1).$$

Eventually, the concentration of (A) and (B) is established and the conclusion follows from Eq. (18).

**Concentration of** $\|\mathbf{W}\mathbf{\Phi}\mathbf{x}\|_2^2$ : In the same manner as Eq. (18), we have the following decomposition:

$$\left\|\frac{1}{\sqrt{h^2\sigma^2}}\mathbf{W}\mathbf{\Phi}\mathbf{x}\right\|_2^2 = \left\|\frac{1}{h}\mathbf{W}\mathbf{\Phi}\frac{\mathbf{x}-\mathbf{x}_0}{\sigma}\right\|_2^2 + \frac{2}{\sigma}\left\langle\frac{1}{h}\mathbf{W}\mathbf{\Phi}\frac{\mathbf{x}-\mathbf{x}_0}{\sigma},\frac{1}{h}\mathbf{W}\mathbf{\Phi}\mathbf{x}_0\right\rangle + \left\|\frac{1}{h}\mathbf{W}\mathbf{\Phi}\frac{\mathbf{x}_0}{\sigma}\right\|_2^2. \quad (21)$$

The subsequent analysis follows in a very similar way to the analysis of $\left\|\frac{1}{\sqrt{h\sigma^2}}\boldsymbol{\Phi}\mathbf{x}\right\|_2^2$. Indeed, we can obtain the following inequalities (each of them with probability at least $1-\delta$, respectively):

$$\left|\left\|\frac{1}{h}\mathbf{W}\boldsymbol{\Phi}\frac{\mathbf{x}-\mathbf{x}_0}{\sigma}\right\|_2 - \left\|\frac{1}{h}\mathbf{W}\boldsymbol{\Phi}\right\|_{\mathrm{F}}\right| \leq \sqrt{\frac{C_2\left\|\mathbf{W}\boldsymbol{\Phi}\right\|^2\log\frac{2}{\delta}}{h^2}}, \tag{22}$$

$$\left|\left\langle\frac{1}{h}\mathbf{W}\boldsymbol{\Phi}\frac{\mathbf{x}-\mathbf{x}_0}{\sigma}, \frac{1}{h}\mathbf{W}\boldsymbol{\Phi}\mathbf{x}_0\right\rangle\right| \leq \sqrt{\frac{C_3\left\|\boldsymbol{\Phi}^\top\mathbf{W}^\top\mathbf{W}\boldsymbol{\Phi}\mathbf{x}_0\right\|_2^2\log\frac{2}{\delta}}{h^4}}, \tag{23}$$

where $C_2$ and $C_3$ are absolute constants (see Eqs. (19) and (20)).

To deal with Eq. (22), we control the spectral norm $\|\mathbf{W}\boldsymbol{\Phi}(t)\|$ along time evolution by using Lemma 15, and obtain the following bound:

$$\|\mathbf{W}\boldsymbol{\Phi}(t)\|^2 \leq \left\{\left\|\boldsymbol{\Phi}^\top\mathbf{W}^\top\mathbf{W}\boldsymbol{\Phi}(0)\right\| + \frac{16\rho t e^{2\rho t} + (2\rho I_0 - 8)(e^{2\rho t}-1)}{\rho^2}\right\}e^{4\rho t},$$

where $I_0 := \mathrm{tr}(\mathbf{W}^\top\mathbf{W}(0)) + \left\|\boldsymbol{\Phi}^\top\boldsymbol{\Phi}(0)\right\|_{\mathrm{F}}$. By plugging this bound back into Eq. (22) and using Lemmas 7 and 8, we obtain

$$\left\|\frac{1}{h}\mathbf{W}\boldsymbol{\Phi}\frac{\mathbf{x}-\mathbf{x}_0}{\sigma}\right\|_2^2 = \left\|\frac{1}{h}\mathbf{W}\boldsymbol{\Phi}\right\|_{\mathrm{F}}^2 + o_{\mathbb{P}}(1).$$

Next, we deal with Eq. (23) by controlling the L2 norm $\left\|\boldsymbol{\Phi}^\top\mathbf{W}^\top\mathbf{W}\boldsymbol{\Phi}(t)\mathbf{x}_0\right\|_2^2$ along time evolution. By using Lemma 16, we obtain the following bound:

$$\left\|\boldsymbol{\Phi}^\top\mathbf{W}^\top\mathbf{W}\boldsymbol{\Phi}(t)\mathbf{x}_0\right\|_2^2$$
$$\leq \left\{\left\|\boldsymbol{\Phi}^\top\mathbf{W}^\top\mathbf{W}\boldsymbol{\Phi}(0)\mathbf{x}_0\right\|_2^2 + \mathcal{O}(\left\|\boldsymbol{\Phi}^\top\boldsymbol{\Phi}(0)\mathbf{x}_0\right\|_2^2) + \|\mathbf{x}_0\|_2^2\,\mathcal{O}(\mathrm{tr}(\mathbf{W}^\top\mathbf{W}(0))^2)\right\}e^{2\rho t},$$

where the order term $\mathcal{O}(\mathrm{tr}(\mathbf{W}^\top\mathbf{W}(0))^2)$ hides the dependency on $t$. We now combine Lemmas 8 and 9 and the consequence of Bernstein's inequality $\|\mathbf{x}_0\|_2^2 = \mathcal{O}_{\mathbb{P}}(d)$ and substitute them into Eq. (23). Then, we obtain

$$\left|\left\langle\frac{1}{h}\mathbf{W}\boldsymbol{\Phi}\frac{\mathbf{x}-\mathbf{x}_0}{\sigma}, \frac{1}{h}\mathbf{W}\boldsymbol{\Phi}\mathbf{x}_0\right\rangle\right|$$
$$\leq \sqrt{C_3'\left\{\frac{\left\|\boldsymbol{\Phi}^\top\mathbf{W}^\top\mathbf{W}\boldsymbol{\Phi}(0)\mathbf{x}_0\right\|_2^2}{h^4} + \frac{\mathcal{O}(\left\|\boldsymbol{\Phi}^\top\boldsymbol{\Phi}(0)\mathbf{x}_0\right\|_2^2)}{h^4} + \frac{\|\mathbf{x}_0\|_2^2\,\mathcal{O}(\mathrm{tr}(\mathbf{W}^\top\mathbf{W}(0))^2)}{h^4}\right\}}$$
$$= \sqrt{o_{\mathbb{P}}(1) + o_{\mathbb{P}}(1)\cdot h^{-2} + \mathcal{O}_{\mathbb{P}}(d)\cdot o_{\mathbb{P}}(1)\cdot h^{-2}}$$
$$= o_{\mathbb{P}}(1),$$

where $C_3' := C_3 e^{2\rho t}\log\frac{2}{\delta}$.

Hence, the concentration result for $\left\|\frac{1}{\sqrt{h^2\sigma^2}}\mathbf{W}\boldsymbol{\Phi}\mathbf{x}\right\|_2^2$ is established by substituting Eqs. (22) and (23) back into Eq. (21). $\qquad\square$

**Lemma 3.** *Under Assumptions 1 to 4, the following concentrations are established:*

$$\left\|\frac{1}{\sqrt{h\sigma^2}}\boldsymbol{\Phi}\mathbf{x}_0\right\|_2 = \left\|\frac{1}{\sqrt{h\sigma^2}}\boldsymbol{\Phi}\right\|_{\mathrm{F}} + o_{\mathbb{P}}(1), \qquad \left\|\frac{1}{\sqrt{h^2\sigma^2}}\mathbf{W}\boldsymbol{\Phi}\mathbf{x}_0\right\|_2 = \left\|\frac{1}{\sqrt{h^2\sigma^2}}\mathbf{W}\boldsymbol{\Phi}\mathbf{x}_0\right\|_{\mathrm{F}} + o_{\mathbb{P}}(1).$$

*Proof of Lemma 3.* To establish concentration of $\|\boldsymbol{\Phi}\mathbf{x}_0\|_2$, we invoke the Hanson–Wright inequality [Ver18, Theorem 6.3.2]: For an absolute constant $C_0$,

$$\left|\left\|\frac{1}{\sqrt{h\sigma^2}}\boldsymbol{\Phi}\mathbf{x}_0\right\|_2 - \left\|\frac{1}{\sqrt{h\sigma^2}}\boldsymbol{\Phi}\right\|_{\mathrm{F}}\right| \leq \sqrt{\frac{C_0\|\boldsymbol{\Phi}\|^2\log\frac{2}{\delta}}{h\sigma^2}},$$

with probability at least $1 - \delta$. Here, we further derive the upper bound of the right-hand side by Lemma 11:

$$\frac{\|\mathbf{\Phi}(t)\|^2}{h} \leq \frac{(\|\mathbf{\Phi}^\top \mathbf{\Phi}(0)\| + 4t)\exp(2\rho t)}{h} = o_\mathbb{P}(1),$$

where the last identity follows from Lemma 7. Thus, the concentration of $\|\mathbf{\Phi}\mathbf{x}_0\|_2$ is shown.

To establish concentration of $\|\mathbf{W}\mathbf{\Phi}\mathbf{x}_0\|$, we invoke the Hanson–Wright inequality again: For an absolute constant $C_1$,

$$\left| \left\| \frac{1}{\sqrt{h^2\sigma^2}}\mathbf{W}\mathbf{\Phi}\mathbf{x}_0 \right\| - \left\| \frac{1}{\sqrt{h^2\sigma^2}}\mathbf{W}\mathbf{\Phi} \right\|_\mathrm{F} \right| \leq \sqrt{\frac{C_1 \|\mathbf{W}\mathbf{\Phi}\|^2 \log \frac{2}{\delta}}{h^2\sigma^2}},$$

with probability at least $1 - \delta$. We can show $\frac{1}{h^2}\|\mathbf{W}\mathbf{\Phi}(t)\|^2 = o_\mathbb{P}(1)$ in the same way as in the proof of Lemma 2. $\qquad\square$

**Lemma 4.** *Let* $\mathbf{\Psi} := \mathbf{W}\mathbf{\Phi}$. *Assume that* $\|\mathbf{\Phi}\|_\mathrm{F}$ *and* $\|\mathbf{\Psi}\|_\mathrm{F}$ *are bounded away from zero. Under Assumptions 1 to 4,* $\mathbf{H}$ *can be expressed as follows:*

$$\mathbf{H} = \frac{1}{1 + \sigma^2}\left\{ \tilde{\mathbf{\Phi}}\tilde{\mathbf{\Psi}}^\top - 2\tilde{\mathbf{\Psi}}\tilde{\mathbf{\Phi}}^\top\tilde{\mathbf{\Psi}}\tilde{\mathbf{\Psi}}^\top - \mathrm{tr}(\tilde{\mathbf{\Phi}}^\top\tilde{\mathbf{\Psi}})\tilde{\mathbf{\Psi}}\tilde{\mathbf{\Psi}}^\top \right\} + o_\mathbb{P}(1),$$

*where* $\tilde{\mathbf{\Phi}} := \mathbf{\Phi}/\|\mathbf{\Phi}\|_\mathrm{F}$ *and* $\tilde{\mathbf{\Psi}} := \mathbf{\Psi}/\|\mathbf{\Psi}\|_\mathrm{F}$.

*Proof of Lemma 4.* To evaluate $\mathbf{H} = \mathbb{E}[\mathbf{z}'\boldsymbol{\omega}^\top - (\boldsymbol{\omega}^\top\mathbf{z}')\boldsymbol{\omega}\boldsymbol{\omega}^\top] := \mathbf{H}_1 - \mathbf{H}_2$, where $\mathbf{H}_1 := \mathbb{E}[\mathbf{z}'\boldsymbol{\omega}^\top]$ and $\mathbf{H}_2 = \mathbb{E}[(\boldsymbol{\omega}^\top\mathbf{z}')\boldsymbol{\omega}\boldsymbol{\omega}^\top]$, we evaluate the normalizers $\|\mathbf{\Phi}\mathbf{x}'\|_2^{-1}$ and $\|\mathbf{W}\mathbf{\Phi}\mathbf{x}\|_2^{-1}$ first. By Lemmas 2 and 3,

$$\frac{1}{\|\mathbf{\Phi}\mathbf{x}'\|_2} = \frac{1}{\sqrt{h\sigma^2}} \cdot \left\{ \left\| \frac{1}{\sqrt{h}}\mathbf{\Phi} \right\|_\mathrm{F}^2 + \left\| \frac{1}{\sqrt{h\sigma^2}}\mathbf{\Phi} \right\|_\mathrm{F}^2 + o_\mathbb{P}(1) \right\}^{-1/2}$$

$$= \frac{1}{\sqrt{h\sigma^2}} \cdot \frac{1}{\sqrt{1 + \sigma^{-2}}\left\| \frac{1}{\sqrt{h}}\mathbf{\Phi} \right\|_\mathrm{F} + o_\mathbb{P}(1)}$$

$$\overset{(\clubsuit)}{=} \frac{1}{\sqrt{h\sigma^2}} \cdot \left\{ \frac{1}{\sqrt{1 + \sigma^{-2}}\left\| \frac{1}{\sqrt{h}}\mathbf{\Phi} \right\|_\mathrm{F}} + o_\mathbb{P}(1) \right\}$$

$$= \frac{1}{\sqrt{1 + \sigma^2}} \cdot \frac{1}{\|\mathbf{\Phi}\|_\mathrm{F}} + o_\mathbb{P}(1),$$

where $(\clubsuit)$ is due to the first-order Taylor expansion $f(\varepsilon) = \frac{1}{x + \varepsilon} \approx \frac{1}{x} - \frac{\varepsilon}{x^2}$ around $\varepsilon = 0$. Similarly, we have

$$\frac{1}{\|\mathbf{W}\mathbf{\Phi}\mathbf{x}\|_2} = \frac{1}{\|\mathbf{\Psi}\mathbf{x}\|_2} = \frac{1}{\sqrt{1 + \sigma^2}} \cdot \frac{1}{\|\mathbf{\Psi}\|_\mathrm{F}} + o_\mathbb{P}(1).$$

Next, we evaluate $\mathbf{H}_1$.

$$\mathbf{H}_1 = \underset{\mathbf{x}_0}{\mathbb{E}}\,\underset{\mathbf{x},\mathbf{x}'}{\mathbb{E}}\left[ \frac{\mathbf{\Phi}\mathbf{x}'}{\|\mathbf{\Phi}\mathbf{x}'\|_2}\left( \frac{\mathbf{\Psi}\mathbf{x}}{\|\mathbf{\Psi}\mathbf{x}\|_2} \right)^\top \right]$$

$$= \underset{\mathbf{x}_0}{\mathbb{E}}\left[ \frac{1}{(1 + \sigma^2)\|\mathbf{\Phi}\|_\mathrm{F}\|\mathbf{\Psi}\|_\mathrm{F}}\,\underset{\mathbf{x},\mathbf{x}'}{\mathbb{E}}[\mathbf{\Phi}\mathbf{x}'\mathbf{x}^\top\mathbf{\Psi}^\top] \right] + o_\mathbb{P}(1)$$

$$= \frac{1}{1 + \sigma^2}\frac{\mathbf{\Phi}}{\|\mathbf{\Phi}\|_\mathrm{F}}\frac{\mathbf{\Psi}^\top}{\|\mathbf{\Psi}\|_\mathrm{F}} + o_\mathbb{P}(1),$$

where we used $\mathbb{E}_{\mathbf{x}_0}\mathbb{E}_{\mathbf{x},\mathbf{x}'}[\mathbf{x}'\mathbf{x}^\top] = \mathbb{E}_{\mathbf{x}_0}[\mathbf{x}_0\mathbf{x}_0^\top] = \mathbf{I}_d$ at the last identity. We can evaluate $\mathbf{H}_2$ similarly.

$$\mathbf{H}_2 = \underset{\mathbf{x}_0}{\mathbb{E}}\,\underset{\mathbf{x},\mathbf{x}'}{\mathbb{E}}\left[ \frac{(\mathbf{x}^\top\mathbf{\Psi}^\top\mathbf{\Phi}\mathbf{x}')\mathbf{\Psi}\mathbf{x}\mathbf{x}^\top\mathbf{\Psi}^\top}{\|\mathbf{\Phi}\mathbf{x}'\|_2\|\mathbf{\Psi}\mathbf{x}\|_2^3} \right]$$

$$= \underset{\mathbf{x}_0}{\mathbb{E}}\left[ \frac{1}{(1 + \sigma^2)^2\|\mathbf{\Phi}\|_\mathrm{F}\|\mathbf{\Psi}\|_\mathrm{F}^3}\mathbf{\Psi}\,\underset{\mathbf{x},\mathbf{x}'}{\mathbb{E}}[(\mathbf{x}^\top\mathbf{\Psi}^\top\mathbf{\Phi}\mathbf{x}')\mathbf{x}\mathbf{x}^\top]\mathbf{\Psi}^\top \right] + o_\mathbb{P}(1),$$

where the inner expectation $\mathbb{E}[(\mathbf{x}^\top \mathbf{\Psi}^\top \mathbf{\Phi} \mathbf{x}') \mathbf{x} \mathbf{x}^\top]$ requires the moment evaluations of Gaussian:

$$
\begin{aligned}
\mathop{\mathbb{E}}_{\mathbf{x}_0} \mathop{\mathbb{E}}_{\mathbf{x},\mathbf{x}'}[(\mathbf{x}^\top \mathbf{\Psi}^\top \mathbf{\Phi} \mathbf{x}') \mathbf{x} \mathbf{x}^\top] &= \mathop{\mathbb{E}}_{\mathbf{x}|\mathbf{x}_0} [\mathbf{x} \mathbf{x}^\top \mathbf{A} \mathbf{x}_0 \mathbf{x}^\top] && \triangleleft \mathbf{A} := \mathbf{\Psi}^\top \mathbf{\Phi} \\
&= \sigma^2 \mathbb{E}[\mathbf{A} \mathbf{x}_0 \mathbf{x}_0^\top] + \sigma^2 \mathbb{E}[\mathbf{x}_0 \mathbf{x}_0^\top \mathbf{A}] \\
&\quad + \mathbb{E}[\mathbf{x}_0 \mathbf{x}_0^\top \mathbf{A} \mathbf{x}_0 \mathbf{x}_0^\top] + \sigma^2 \mathbb{E}[\mathbf{x}^\top \mathbf{A} \mathbf{x}_0] \mathbf{I}_d && \triangleleft \text{[PP12, §8.2.3]} \\
&= 2\sigma^2 \mathbf{A} + \mathbb{E}[\mathbf{x}_0 \mathbf{x}_0^\top \mathbf{A} \mathbf{x}_0 \mathbf{x}_0^\top] + \sigma^2 \operatorname{tr}(\mathbf{A}) \mathbf{I}_d && \triangleleft \text{[PP12, §8.2.2]} \\
&= 2\sigma^2 \mathbf{A} + \{2\mathbf{A} + \operatorname{tr}(\mathbf{A}) \mathbf{I}_d\} + \sigma^2 \operatorname{tr}(\mathbf{A}) \mathbf{I}_d && \triangleleft \text{[PP12, §8.2.4]} \\
&= (1+\sigma^2)\{2\mathbf{\Psi}^\top \mathbf{\Phi} + \operatorname{tr}(\mathbf{\Psi}^\top \mathbf{\Phi}) \mathbf{I}_d\}.
\end{aligned}
$$

Note that $\mathbf{\Psi}^\top \mathbf{\Phi} = \mathbf{A} = \mathbf{A}^\top = \mathbf{\Phi}^\top \mathbf{\Psi}$ under Assumption 1. By plugging this back,

$$
\mathbf{H}_2 = \frac{1}{1+\sigma^2} \left\{ 2\tilde{\mathbf{\Psi}} \tilde{\mathbf{\Phi}}^\top \tilde{\mathbf{\Psi}} \tilde{\mathbf{\Psi}}^\top + \operatorname{tr}(\tilde{\mathbf{\Psi}}^\top \tilde{\mathbf{\Phi}}) \tilde{\mathbf{\Psi}} \tilde{\mathbf{\Psi}}^\top \right\} + o_{\mathbb{P}}(1).
$$

The desired expression of $\mathbf{H} = \mathbf{H}_1 - \mathbf{H}_2$ is thereby obtained. $\qquad \square$

**Proposition 1.** *Suppose $\mathbf{W}$ is non-singular. Under the dynamics* (4) *with $\mathbf{H} = \hat{\mathbf{H}}$, the commutator $\mathbf{L}(t) := [\mathbf{F}, \mathbf{W}] := \mathbf{F}\mathbf{W} - \mathbf{W}\mathbf{F}$ satisfies $\frac{\mathrm{d}\operatorname{vec}(\mathbf{L}(t))}{\mathrm{d}t} = -\mathbf{K}(t)\operatorname{vec}(\mathbf{L}(t))$, where*

$$
\mathbf{K}(t) := 2\frac{\mathbf{W} \oplus \mathbf{W}\mathbf{F}\mathbf{W} + \mathbf{W}^2(\mathbf{F}\mathbf{W} \oplus \mathbf{I}_d)}{(1+\sigma^2)N_\Phi N_\Psi^3} + \frac{(\mathbf{W}^{-1}) \oplus \mathbf{F} - (\mathbf{W} - N_\times \mathbf{W}^2) \oplus \mathbf{I}_d}{(1+\sigma^2)N_\Phi N_\Psi} + 3\rho \mathbf{I}_d,
$$

*and $\mathbf{A} \oplus \mathbf{B} := \mathbf{A} \otimes \mathbf{B} + \mathbf{B} \otimes \mathbf{A}$ denotes the sum of the two Kronecker products.*

*If $\inf_{t \geq 0} \lambda_{\min}(\mathbf{K}(t)) \geq \lambda_0 > 0$ for some $\lambda_0 > 0$, then $\|\mathbf{L}(t)\|_{\mathrm{F}} \to 0$ as $t \to \infty$.*

In the proof, we leverage the elementary properties of commutators.

**Lemma 17.** *For matrices $\mathbf{A}$, $\mathbf{B}$, and $\mathbf{C}$ with the same size, we have the following identities.*

1. $[\mathbf{A}, \mathbf{A}] = \mathbf{O}$.

2. $[\mathbf{A}, \mathbf{B}] = -[\mathbf{B}, \mathbf{A}]$.

3. $[\mathbf{A}, \mathbf{B}\mathbf{C}] = [\mathbf{A}, \mathbf{B}]\mathbf{C} + \mathbf{B}[\mathbf{A}, \mathbf{C}]$.

4. $[\mathbf{A}\mathbf{B}, \mathbf{C}] = \mathbf{A}[\mathbf{B}, \mathbf{C}] + [\mathbf{A}, \mathbf{C}]\mathbf{B}$.

*Proof of Proposition 1.* First, compute the time derivative $\dot{\mathbf{L}} = \mathbf{F}\dot{\mathbf{W}} - \dot{\mathbf{W}}\mathbf{F} + \dot{\mathbf{F}}\mathbf{W} - \mathbf{W}\dot{\mathbf{F}}$:

$$
\begin{aligned}
\mathbf{F}\dot{\mathbf{W}} - \dot{\mathbf{W}}\mathbf{F} &= \mathbf{F}\mathbf{H}^\top \mathbf{W}^{-1} - \mathbf{W}^{-1}\mathbf{H}\mathbf{F} - \rho\mathbf{L}, \\
\dot{\mathbf{F}}\mathbf{W} - \mathbf{W}\dot{\mathbf{F}} &= \mathbf{W}\mathbf{H} - \mathbf{H}^\top \mathbf{W} + \mathbf{W}^{-1}\mathbf{H}^\top \mathbf{W}^2 - \mathbf{W}^2 \mathbf{H}\mathbf{W}^{-1} - 2\rho\mathbf{L},
\end{aligned}
$$

which implies

$$
\dot{\mathbf{L}} = (\mathbf{F}\mathbf{H}^\top \mathbf{W}^{-1} - \mathbf{W}^{-1}\mathbf{H}\mathbf{F}) + (\mathbf{W}\mathbf{H} - \mathbf{H}^\top \mathbf{W}) + (\mathbf{W}^{-1}\mathbf{H}^\top \mathbf{W}^2 - \mathbf{W}^2 \mathbf{H}\mathbf{W}^{-1}) - 3\rho\mathbf{L}. \quad (24)
$$

We substitute $\mathbf{H} = \hat{\mathbf{H}}$. Then,

$$
\mathbf{W}\mathbf{H} - \mathbf{H}^\top \mathbf{W} = -2\frac{\mathbf{W}^2 \mathbf{F}\mathbf{W}\mathbf{F}\mathbf{W} - \mathbf{W}\mathbf{F}\mathbf{W}\mathbf{F}\mathbf{W}^2}{(1+\sigma^2)N_\Phi N_\Psi^3} - N_\times \frac{\mathbf{W}^2 \mathbf{F}\mathbf{W} - \mathbf{W}\mathbf{F}\mathbf{W}^2}{(1+\sigma^2)N_\Phi N_\Psi},
$$

which can be simplified by Lemma 17 as follows:

$$
\begin{cases}
\mathbf{W}^2 \mathbf{F}\mathbf{W}\mathbf{F}\mathbf{W} - \mathbf{W}\mathbf{F}\mathbf{W}\mathbf{F}\mathbf{W}^2 = [\mathbf{W}, \mathbf{W}\mathbf{F}\mathbf{W}\mathbf{F}]\mathbf{W} = -(\mathbf{L}\mathbf{W}\mathbf{F} + \mathbf{F}\mathbf{W}\mathbf{L})\mathbf{W}, \\
\mathbf{W}^2 \mathbf{F}\mathbf{W} - \mathbf{W}\mathbf{F}\mathbf{W}^2 = [\mathbf{W}, \mathbf{W}\mathbf{F}\mathbf{W}] = -\mathbf{W}\mathbf{L}\mathbf{W}.
\end{cases}
$$

With the same technique, Eq. (24) can be simplified as follows:

$$
\begin{aligned}
\dot{\mathbf{L}} = {}& \frac{(\mathbf{L}\mathbf{W} + \mathbf{W}\mathbf{L}) - (\mathbf{F}\mathbf{L}\mathbf{W}^{-1} + \mathbf{W}^{-1}\mathbf{L}\mathbf{F})}{(1+\sigma^2)N_\Phi N_\Psi} \\
& - 2\frac{(\mathbf{W}\mathbf{F}\mathbf{W}\mathbf{L}\mathbf{W} + \mathbf{W}\mathbf{L}\mathbf{W}\mathbf{F}\mathbf{W}) + \mathbf{W}^2(\mathbf{F}\mathbf{W}\mathbf{L} + \mathbf{L}\mathbf{W}\mathbf{F})}{(1+\sigma^2)N_\Phi N_\Psi^3} \\
& - N_\times \frac{\mathbf{L}\mathbf{W}^2 + \mathbf{W}^2 \mathbf{L}}{(1+\sigma^2)N_\Phi N_\Psi} - 3\rho\mathbf{L}.
\end{aligned}
$$

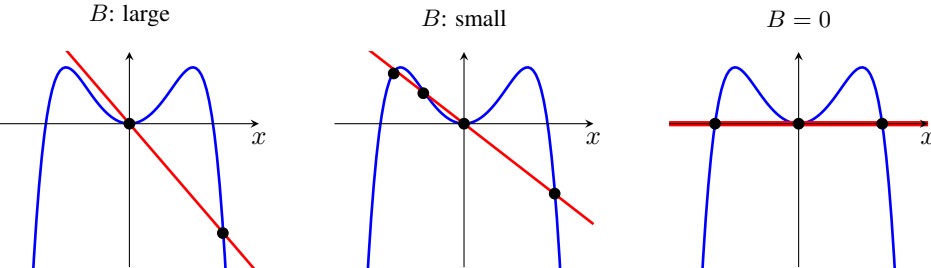

**Figure 7:** Plots of $g(x) = -Ax^6 + x^2$ (blue) and $h(x) = -Bx$ (red). **(Left)** $(A, B) = (1.5, 0.6)$ **(Center)** $(A, B) = (1.5, 0.4)$ **(Right)** $(A, B) = (1.5, 0)$

By using $\mathrm{vec}(\mathbf{ALB} + \mathbf{BLA}) = (\mathbf{B} \otimes \mathbf{A} + \mathbf{A} \otimes \mathbf{B})\mathrm{vec}(\mathbf{L}) = (\mathbf{A} \oplus \mathbf{B})\mathrm{vec}(L)$ for $\mathbf{A}, \mathbf{B} \in \mathbb{Sym}_d$, we obtain $\frac{\mathrm{d}\mathrm{vec}(\mathbf{L})}{\mathrm{d}t} = -\mathbf{K}\mathrm{vec}(\mathbf{L})$.

Finally, by applying [TCG21, Lemma 2], the dynamics of $\mathbf{L}(t)$ satisfies $\|\mathrm{vec}(\mathbf{L}(t))\|_2 \leq e^{-2\lambda_0 t} \|\mathrm{vec}(\mathbf{L}(0))\|_2 \to 0$ under the assumption $\inf_{t \geq 0} \lambda_{\min}((\mathbf{K}(t))) \geq \lambda_0 > 0$. $\qquad\square$

**Proposition 2.** *Suppose $\mathbf{W}$ is non-singular. Under the dynamics* (4) *with $\mathbf{H} = \hat{\mathbf{H}}$, we have $\dot{\mathbf{U}} = \mathbf{O}$.*

*Proof of Proposition 2.* The proof mostly follows the discussion of [TCG21, Appendix B.1]. To apply their discussion, all we need to check is the existence of diagonal matrices $\mathbf{G}_1$ and $\mathbf{G}_2$ such that $\dot{\mathbf{W}} = \mathbf{UG}_1\mathbf{U}^\top$ and $\dot{\mathbf{F}} = \mathbf{UG}_2\mathbf{U}^\top$ under the dynamics Eq. (4) with $\mathbf{H} = \hat{\mathbf{H}}$.

For $\dot{\mathbf{W}}$, invertibility of $\mathbf{W}$ implies $\dot{\mathbf{W}} = \mathbf{W}^{-1}\hat{\mathbf{H}} - \rho\mathbf{W}$ from the dynamics Eq. (4). With simultaneous diagonalization $\mathbf{W} = \mathbf{U}\boldsymbol{\Lambda}_W\mathbf{U}^\top$ and $\mathbf{F} = \mathbf{U}\boldsymbol{\Lambda}_F\mathbf{U}^\top$, we have $\mathbf{W}^{-1} = \mathbf{U}\boldsymbol{\Lambda}_W^{-1}\mathbf{U}^\top$ and $\hat{\mathbf{H}} = \mathbf{U}\boldsymbol{\Lambda}_{\hat{H}}\mathbf{U}^\top$ for some diagonal matrix $\boldsymbol{\Lambda}_{\hat{H}}$. Hence, $\dot{\mathbf{W}} = \mathbf{UG}_1\mathbf{U}^\top$ for some diagonal matrix $\mathbf{G}_1$.

In the same manner, we can verify $\dot{\mathbf{F}} = \mathbf{UG}_2\mathbf{U}^\top$ for some diagonal matrix $\mathbf{G}_2$. $\qquad\square$

## C    ANALYSIS OF REGIME SHIFT

In Section 5.2, we claimed that the $p_j$-dynamics (8) entails the three regimes, mainly based on categorization of the numerical plots with different values of $(N_\Phi, N_\Psi, \rho)$ in Fig. 2. Here, we show that the equilibrium point sets with different parameter values can indeed be classified into the three regimes.

First, we need slight approximation because the $p_j$-dynamics (8) is sixth-order and extremely challenging to deal with analytically in general. We choose to set $N_\times (= \mathrm{tr}(\tilde{\boldsymbol{\Phi}}^\top \tilde{\boldsymbol{\Psi}})) \approx 0$. This can be confirmed in our simple numerical experiments in Fig. 6. Then, the $p_j$-dynamics reads:

$$\dot{p}_j \approx \frac{1}{(1+\sigma^2)N_\Phi N_\Psi} \underbrace{\left\{ -\frac{2}{N_\Psi^2}p_j^6 + p_j^2 - \rho(1+\sigma^2)N_\Phi N_\Psi p_j \right\}}_{=f(p_j)}.$$

Let us write $f(x) = -Ax^6 + x^2 - Bx$ with $A \coloneqq 2/N_\Psi^2 > 0$ and $B \coloneqq \rho(1+\sigma^2)N_\Phi N_\Psi \geq 0$. Now, we focus on finding the roots of $f(x) = 0$, which are the equilibrium points of the $p_j$-dynamics. In Fig. 7, we show the graphs of $g(x) = -Ax^6 + x^2$ and $h(x) = -Bx$ with different $B$. When $B = 0$, we can analytically find the roots of $f(x) = g(x) = 0$ by $g(x) = -Ax^2(x^2+A^{-1/2})(x+A^{-1/4})(x-A^{-1/4})$ and $x = 0, \pm A^{-1/4}$. This corresponds to the Stable regime in Fig. 3. When $B$ is larger than zero and as $h(x) = -Bx$ tilts towards negative slightly, we have four roots as seen in Fig. 7 (Center). This corresponds to the Acute regime in Fig. 3. Finally, when $B$ is significantly larger than zero, we have only two roots as seen in Fig. 7 (Left), which corresponds to the Collapse regime in Fig. 3. These three cases are interpolated smoothly as $B \propto \rho N_\Phi N_\Psi$ changes; to put it differently, as regularization strength $\rho$ and norms $N_\Phi, N_\Psi$ decrease, the regime approaches the Stable. Note again that we will never perfectly attain the Stable regime because the $p_j$-dynamics diverges as $N_\Phi, N_\Psi \to 0$.

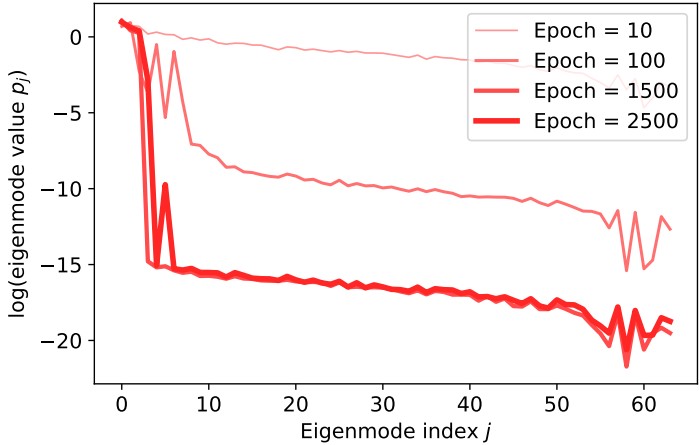

**Figure 8:** Time evolution of the eigenmodes. At each epoch, the projection head eigenmode $p_j$ for each $j \in \{1, 2, \ldots, 64\}$ is plotted. The eigenmode values are uniformly averaged within $[\text{epoch} - 50, \text{epoch} + 50]$ to avoid visual clutter due to eigenmode fluctuation.

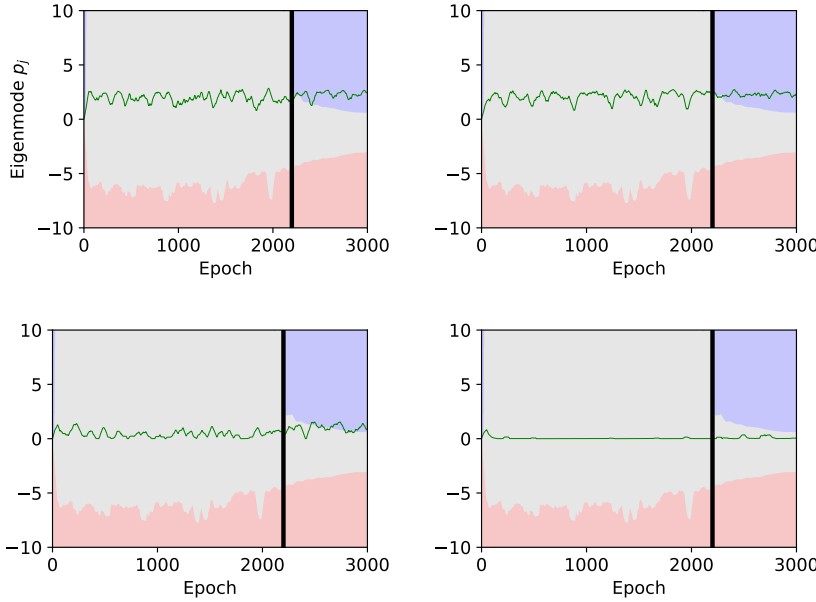

**Figure 9:** The eigenmodes of the projection head $p_j$ are plotted, with background colors illustrating three intervals where $p_j$ diverges , $p_j$ collapses , and $p_j$ stably converges at each epoch. Each color corresponds to those in Fig. 3. The vertical black line indicates the shift from Collapse (epoch < 2200) to Acute (epoch > 2200). The eigenmode values are uniformly averaged within $[\text{epoch} - 50, \text{epoch} + 50]$ to avoid visual clutter due to eigenmode fluctuation. **(Top left)** $j = 1$ (the largest eigenmode); **(Top center)** $j = 2$ (the second largest eigenmode); **(Top center)** $j = 3$ (the third largest eigenmode); **(Bottom left)** $j = 4$ (the fourth largest eigenmode);

# D    ADDITIONAL NUMERICAL EXPERIMENTS

## D.1    FULL DETAIL OF LINEAR ENCODER SETUP

We further analyze the numerical experiments in Section 5.4. In Section 5.4, we focused on illustration of the leading eigenmode $p_j$ of the projection head, which is shown in Fig. 6. Here, we

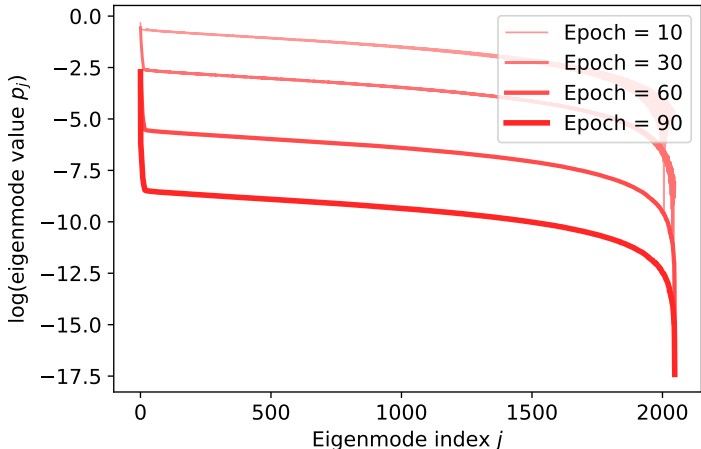

**Figure 10:** Time evolution of the eigenmodes (trained with the nonlinear encoder). At each epoch, the projection head eigenmode $p_j$ for each $j \in \{1, 2, \ldots, 2048\}$ is plotted.

investigate the other eigenmodes. Throughout the analysis, we focus on the absolute value of the eigenmode $|p_j|$ because the eigendecomposition is non-unique; indeed, due to the decomposition $\mathbf{W} = \sum_{j=1}^{64} p_j \mathbf{u}_j \mathbf{u}_j^\top$ ($\mathbf{u}_j$ is the eigenvector), the eigenmode signs are irrelevant to the norms of the eigenvectors. Flipping the sign does not affect the orthonormality of the eigenvectors, keeping $\mathbf{U}$ to be a orthogonal matrix. After taking the absolute values, all eigenmodes are sorted in the descending order, where $j = 1$ and $j = 64$ correspond to the largest and smallest, respectively.

Figure 8 illustrates time evolution of the eigenmode values of the projection head. Initially (epoch $=$ 10), the eigenmode distribution mildly concentrates around the origin, which can be seen in the initialization of the eigenvalue distribution in Fig. 4 as well. As time evolves, the distribution quickly concentrates at zero very sharply, whereas a few positive eigenmodes that are significantly larger than zero remains.

Next, we investigate time evolution of each eigenmode individually. Figure 9 shows time evolution of the largest ($j = 1$), second largest ($j = 2$), third largest ($j = 3$), and fourth largest ($j = 4$), using the same illustration as Fig. 6. The top left figure ($j = 1$) is the same one as in Fig. 6. As can be seen in this case, only $p_1$, $p_2$, and $p_3$ remain positive and all the other eigenmodes (including $5 \le j \le 64$ omitted from Fig. 9) converges to nearly zero. In our theoretical analysis, we argued that there are only two stable equilibrium in the Acute regime ($p_j = 0$ and $p_j = p_{\blacktriangledown}^{(+)}$ in Fig. 3). Given this, the convergences of $p_{\{1,2,3\}}$ to positive values (that even fall in the stable interval) and $\{p_j\}_{j=3}^{63}$ to zero are reasonable in terms of the dynamics. Moreover, this convergence avoids the complete collapse $\mathbf{W} \to \mathbf{O}$; the complete collapse is avoided if several (but not necessarily all) eigenmodes remain to be non-zero.

### D.2 SIMULATION WITH NONLINEAR ENCODER

Here, we complement our analysis by conducting the numerical simulation of the SimSiam model using a nonlinear encoder. As in Section 5.4, we use the official implementation of SimSiam. The implementation differences from the official code are listed below:

- Dataset: CIFAR-10
- The feature encoder: ResNet-18, but the last fully-connected layers being replaced with linear $\mathbf{\Phi}$
- The projection head: linear $\mathbf{W} \in \mathbb{R}^{2048 \times 2048}$ without bias ($h = 2048$)
- Parameter initialization: following Assumption 4 and $\mathbf{W}$ are symmetrized by $(\mathbf{W} + \mathbf{W}^\top)/2$
- Optimizer: the momentum SGD with the initial learning rate 0.005
- Regularization strength: $\rho = 0.008$

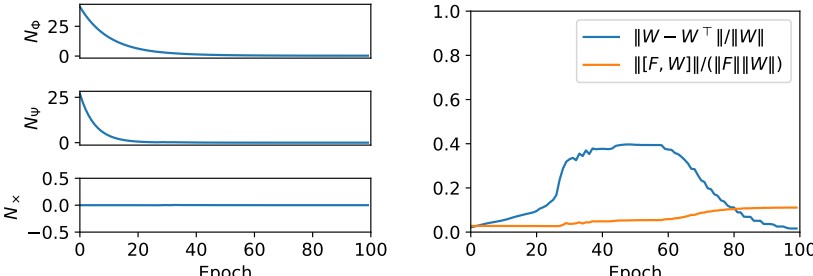

**Figure 11:** Numerical simulation of the SimSiam model with the nonlinear encoder. **(Left)** Time evolution of $N_\Phi$, $N_\Psi$, and $N_\times$. **(Right)** Asymmetry of the projection head $\mathbf{W}$ (measured by the relative error of $\mathbf{W} - \mathbf{W}^\top$) and non-commutativity of $\mathbf{F}$ and $\mathbf{W}$ (measured by the relative error of the commutator $[\mathbf{F}, \mathbf{W}]$).

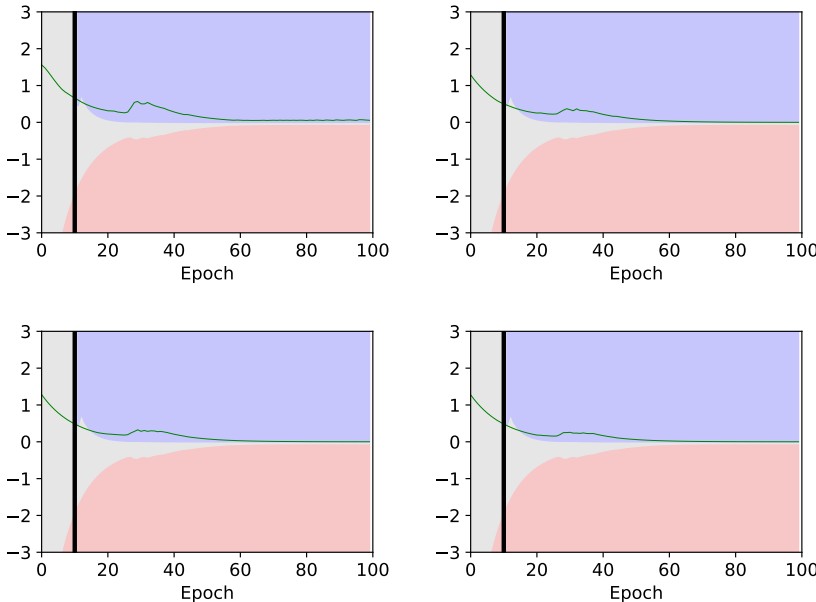

**Figure 12:** The eigenmodes of the projection head $p_j$ are plotted (trained with the nonlinear encoder), with background colors illustrating three intervals where $p_j$ diverges , $p_j$ collapses , and $p_j$ stably converges at each epoch. Each color corresponds to those in Fig. 3. The vertical black line indicates the shift from Collapse (epoch $< 10$) to Acute (epoch $> 10$). **(Top left)** $j = 1$ (the largest eigenmode); **(Top center)** $j = 2$ (the second largest eigenmode); **(Top center)** $j = 3$ (the third largest eigenmode); **(Bottom left)** $j = 4$ (the fourth largest eigenmode);

- Epochs: 100

We used the same data augmentation applied to the ImageNet dataset in the official implementation. The other details remain to be the same as the official implementation.

To see how the nonlinear setup aligns with Assumption 1 (symmetry of $\mathbf{W}$), Assumption 6 (commutativity of $\mathbf{W}$ and $\mathbf{F}$), and Assumption 5 (constancy of $N_\Phi$, $N_\Psi$, and $N_\times$), we show them in Fig. 11. The norm parameters $N_\Phi$, $N_\Psi$, and $N_\times$ remains relatively stable, which aligns with Assumption 5 well. During the training epochs, $\mathbf{W}$ becomes relatively asymmetry, but converges to a symmetric matrix. This point needs to be carefully addressed in future work. We can suppose that $\mathbf{W}$ and $\mathbf{F}$ remain to be commutative.

The time evolution of the eigenvalues of the linear projection head $\mathbf{W}$ is shown in Fig. 10, and each eigenvalue ($j = 1, 2, 3, 4$) is shown in Fig. 12. Each background color in Fig. 12 indicates whether

$p_j$ diverges (red), collapses (gray), and stably converges (blue). The boundaries of these intervals are computed by numerical root finding of the $p_j$-dynamics (8). We observe that only a few number of eigenvalues remain to be non-zero while most of them degenerate to zero; general trend observed in the synthetic case using the linear encoder (Appendix C). Moreover, we can see that the initial Collapse regime ($\text{epoch} < 10$) is lifted to the Acute regime ($\text{epoch} > 10$) in Fig. 12. The (non-zero) eigenvalues eventually converge to the values in the (blue) stable interval.

