# OpenReview forum: "Feature Normalization Prevents Collapse of Non-contrastive Learning Dynamics"
_ICLR.cc/2024/Conference — Submitted to ICLR 2024_

### Official Review · Reviewer_JjF9 · 2023-10-28

**Soundness:** 3 good
**Presentation:** 3 good
**Contribution:** 3 good
**Rating:** 6
**Confidence:** 3

**Summary:**

This paper represents an extension of prior work in the field of Self-Supervised Learning (SSL) theory, with a specific emphasis on elucidating how non-contrastive SSL methods prevent the issue of feature collapse. The paper's primary focus centers on the examination of the final feature normalization step and its role in the underlying dynamics. The authors furnish compelling evidence concerning the dynamics of the underlying eigenmodes, and the theory finds support through numerical simulations.

**Strengths:**

1. This paper addresses an important and relatively underexplored issue regarding the role of feature normalization in non-contrastive Self-Supervised Learning (SSL). The authors demonstrate that the normalization step introduces sixth-order dynamics, resulting in the dynamic emergence of a stable equilibrium, even when dealing with initially collapsed solutions.

2. The authors present compelling evidence, and their underlying assumptions appear to be quite reasonable.

3. Numerical simulations validate the predictions made by the theory.

**Weaknesses:**

I would anticipate the theoretical framework to align with the behavior observed in real datasets. However, the paper does not investigate the dynamics in more complex scenarios.

**Questions:**

The authors mentioend BarlowTwins and VICReg. They effectively are still contrastive. How do you think their 'feature normalization' behavior is related?

---

> ### Author Response · Authors · 2023-11-19
> **Response**
>
> We are glad to see the reviewer acknowledges our contributions positively. Our responses to your concerns and questions are as follows:
>
> **More complex scenarios:** Thank you for the suggestion. In the newly added Appendix D.2, we provide the similar experiments to the synthetic case with the ResNet-18 encoder. Overall, we can see a similar trend, and moreover, the regime transition can be observed as well (with properly chosen parameters).
>
> **Barlow Twins and VICReg:** Compared to the conventional contrastive learning pushing anchors away from negative examples, Barlow Twins and VICReg are still different because their variance regularizer is to regularize the _dimension-wise_ variance, not _sample-wise_. Thus, their underlying mechanism to avoid the collapse is still an interesting problem to study, particularly under the presence of feature normalization, which has not been dealt with.

---

### Official Review · Reviewer_HYk4 · 2023-10-29

**Soundness:** 3 good
**Presentation:** 3 good
**Contribution:** 2 fair
**Rating:** 5
**Confidence:** 4

**Summary:**

This paper follows the previous setting in Tian et al. (2021) which explores simplified modeling for non-contrastive learning. It posits the representation model as an identity function, with both the projection layer and prediction layer streamlined into linear components. What distinguishes this study from its predecessors is the exploration of the commonly used cosine loss in practical applications. By applying these simplifications and introducing additional assumptions, the authors demonstrate that the norms tend to concentrate around some constants, which helps to simplify the learning dynamics with feature normalization. With further assumptions, the paper disentangles the learning dynamics into the sixth-order eigenmode dynamics in which a stable equilibrium emerges even if there is no stable equilibrium with the initial parametrization and regularization strength.

**Strengths:**

- The paper is well-written and easy to follow.
- This work proves that the feature norm concentrates around a constant with proper parameter initialization.

**Weaknesses:**

1. Some of the assumptions are quite stringent, especially since this paper is not pioneering work, and they may not provide much reference value for practical non-contrastive learning with negative pairs.
2. Assumptions 2 and 3 in section 4 are rather strict. Assumption 2 requires that the input data follow an isotropic Gaussian distribution, which is hard to accept in practical situations. Perhaps a mixture of isotropic Gaussians could be considered. Assumption 3 pertains to the width-infinite limit.
3. In section 5, the authors consider the norms of these linear layers as constants (Assumption 5). This assumption, however, is still far from providing a real dynamic analysis for the cosine loss. Since feature normalization may not guarantee convexity, smoothness, and Lipschitzness, its dynamic analysis should focus on proving the convergence rather than simplifying its complexity to obtain closed-formed dynamics. The existing conclusions do not provide much contribution and insight to understanding non-contrastive learning dynamics.
4. The relevant numerical results still do not fully validate the reasonableness of these assumptions, such as the increasing error between $W$ and $W^\top$, and the decrease in $N_{\phi}$ and $N_{\psi}$. Therefore, while I appreciate the authors for using Hanson-Wright inequality to demonstrate that some norms concentrate, it is still not particularly remarkable.

**Questions:**

Please see weakness

---

> ### Author Response · Authors · 2023-11-19
> **Response**
>
> Thank you for carefully reviewing our manuscript. Let us discuss your concerns about the assumptions. We hope the discussion sounds reasonable and supports our contribution further.
>
> **Assumption of the width-infinite limit:** This assumption is relevant to the norm concentration, which is essentially the central limit theorem. Numerically, the concentration can be established with the order of 100 dimensions; this is not unrealistic limit, unlike some generalization analysis of neural nets requiring exponentially more dimensions than samples.
>
> **Assumption on the norm constancy:** This assumption is only relevant to disentangle the matrix dynamics (4) into the eigenmode dynamics (6). We do not need the norm constancy globally (for all time $t$) if we are concerned about this disentanglement at each fixed time. Globally, the norm values may of course change; however, the shapes of dynamics at each time shown in Figures 2 and 3 remain qualitatively the same.
>
> **General:** Dynamics analysis generally requires strong assumptions such as (isotropic) Gaussian distributions and linear-algebraic assumptions—they are often stronger than practical scenarios. However, this does not mean that these analyses are pointless; as long as they can serve as proxies to some real scenarios. How about our analysis? Isotropic Gaussian—yes, it is quite strong, yet we can establish the regime transition and go beyond the L2-loss analysis (that fails with excessively strong regularization) at least in this specific scenario. The symmetry assumption of $W$ as well: as our simulation actually observes the eigenmodes converge to the stable interval, this can be seen as corroborating evidence that our model behaves as a reasonable proxy. Everyone anticipates the extension to general setups, but not all assumptions can be lifted all at once—our scientific journey is a continuous step to build a better model little by little. To this end, our work contribute to taking feature normalization into account and relating it to complete collapse in a synthetic setup first.

---

> > ### Comment · Reviewer_HYk4 · 2023-11-23
> >
> > I appreciate the authors for responding. However, the provided responses did not sufficiently address my concerns, specifically, the mentioned assumptions do not effectively support the core of the paper, "Feature Normalization Prevents Collapse of Non-contrastive Learning Dynamics." Consequently, I will maintain my original score.

---

> > > ### Author Response · Authors · 2023-11-23
> > > **Further response**
> > >
> > > Again, thank you so much for providing your opinion further.
> > >
> > > While we have already tried to clarify "our work contributes to taking feature normalization into account and relating it to complete collapse in a synthetic setup first" in [the last response](https://openreview.net/forum?id=RlfD5cE1ep&noteId=ZDEsg0kN7S), the reviewer claims "the mentioned assumptions do not effectively support the core of the paper, 'Feature Normalization Prevents Collapse of Non-contrastive Learning Dynamics.'" We suppose that this is a philosophical issue whether we allow _step-by-step_ lifting of assumptions or do only allow lifting all assumptions all at once. Thus, we let the area chair take the decision.

---

> > > > ### Comment · Reviewer_HYk4 · 2023-11-23
> > > > **Thank you!**
> > > >
> > > > I do not harbor doubts regarding the step-by-step lifting of assumptions in the paper. However, my concern lies in the oversimplified nature of the assumptions used, particularly assumption 5. This particular assumption appears to disconnect the entire paper from the practicalities of feature normalization.

---

> > > > > ### Author Response · Authors · 2023-11-23
> > > > > **Further clarification**
> > > > >
> > > > > We appreciate the reviewer for clarifying the misunderstanding in my previous post. Regarding Assumption 5, we have the following claim in the initial response:
> > > > >
> > > > > > **Assumption on the norm constancy:** This assumption is only relevant to disentangle the matrix dynamics (4) into the eigenmode dynamics (6). We do not need the norm constancy globally (for all time) if we are concerned about this disentanglement at each fixed time. Globally, the norm values may of course change; however, the shapes of dynamics at each time shown in Figures 2 and 3 remain qualitatively the same.
> > > > >
> > > > > To state differently, we _do not_ need the global norm constancy for all time $t$ if we want to characterize the shape of the dynamics at a fixed time $t$; of course, this fixed dynamics cannot be directly tracked along time, but qualitative behaviors of the equilibriums remain the same. We will try to improve the writing of this part to avoid the confusion.

---

### Official Review · Reviewer_sECc · 2023-10-30

**Soundness:** 3 good
**Presentation:** 3 good
**Contribution:** 2 fair
**Rating:** 8
**Confidence:** 4

**Summary:**

This article proposes an extension of the theory of non-contrastive learning (BYOL, SimSiam) to consider the cosine loss rather than the L2 loss, showing how feature normalization changes from third-order dynamics to sixth-order.
They show that three regimes exist depending on the norms of the layers, which results in a shift between the three regimes as the norms decrease until the stable regime, where the eigenmodes converge.

**Strengths:**

This article presents an improvement other than the theoretical framework of non-contrastive learning using solely the Euclidian loss.
The paper is well-written and easy to follow. The assumptions taken are relatively well justified and allow for an interesting analysis.

**Weaknesses:**

**Previous literature** There have been recent contributions to the literature of non-contrastive learning which do take into account the cosine loss, and which are not referenced in this article. In particular, Halvagal et al., Implicit variance regularization in non-contrastive SSL, 2023. The eigenmode dynamics seem extremely similar (after some changes in the notation) and it seems extremely important to me that the authors compare themselves to this article. The authors also do not seem to have a similar conclusion on the implicit variance regularization that Halvagal et al. focus on.

**Regimes** The three regimes found in Section 5.2. seem to have been found solely by categorizing the regimes experimentally shown in Figure 2 while reading, which seems like a weak justification for those regimes. A clearer analysis of the equilibrium points at least in the Appendix seems necessary.

The authors claim that as the norms decrease, the regimes fall to the stable one. However, in the stable regime, the norms will increase as the eigenmodes increase to the saddle point $p^+$. Is there a risk of the acute and stable regimes alternating between each other?

**Experiments** Numerical experiments on the SimSiam model remain on linear networks in Section 5.4. A similar Figure to Figure 6 for a real network such as ResNet (maybe only focusing on the linear projection head) would help confirm the theoretical findings in the linear case. Otherwise, the link with a real SimSiam network remains relatively limited, except for the weight decay argument.

**Figure 6c** I also find Figure 6.c. hard to read. Are the intervals the theoretical intervals using the values of the norms? In this case, what is the theoretical value of the saddle point? Having the values of a single eigenmode gives very little information on the values of the spectrum of $W$. Do all the values stay relatively constant like here?

**Questions:**

The notion of Thermodynamic limit is novel to me in optimization and needs to be further explained. How is it different from the Neural Tangent Kernel regime, is it the constant ratio $\alpha$?

Do the authors have more intuition on the role of the exponential moving average in BYOL with their new findings?

Small remarks:
* Intro: "Folklore says" is a somewhat strange way to quote an article.
* Sec 4. Assumption 2. $Σ = I $ seems superfluous to add after $D = N(0, I)$.
* After Lemma 3: $\Phi x'$ is not defined.
* Equation (5): $\hat H$ is not defined.
* After Equation (9): "unite learning rate"

---

> ### Author Response · Authors · 2023-11-19
> **Response**
>
> We are grateful for carefully reviewing our manuscript and providing extremely insightful comments. Please see our response below. Typos have been addressed already.
>
> **Previous literature:** A good catch! It’s a shame that we had been aware of Halvagal et al. (2023) only one week after the submission… They have successfully established the mechanism of the implicit variance regularization of the eigenvalues, which would not be elicitable from our analysis because we assume the standard normal input, regrettably. By contrast, their implicit regularization holds _only for non-collapsed eigenvalues ($\\lambda_k > 0$)_, as seen in their Eq. (11). Thus, they do not explain how the complete collapse ($\\lambda_k = 0$ for all $k=0$) is avoided; our work contributes to this end. The clarification has been added in the updated draft.
>
> **Regimes:** Based on your suggestion, we attempted to give a slightly more formal derivation of the regimes in newly added Appendix C, though $p\_j$-dynamics is sixth-order and cannot be solved analytically in general.
>
> Regarding the Acute and Stable regimes, we will not have alternation of them numerically because the Stable regime is conceptual and cannot be attained exactly; the unstable interval (gray in Figure 3) could be arbitrarily small as the norms decrease, but will not completely disappear.
>
> **Experiments:**  Thank you for the suggestion. In the newly added Appendix D.2, we provide the similar experiments to the synthetic case with the ResNet-18 encoder. Overall, we can see a similar trend, and moreover, the regime transition can be observed as well (with properly chosen parameters).
>
> **Figure 6c:** We improved the visualization by adding smoothing along the time axis and added a description to explain how the regime intervals are calculated: they are calculated theoretically by using the $p\_j$-dynamics (8), but solving the sixth-order equation by numerical root finding. In addition, we expanded the analysis in newly added Appendix D.1 to see not only the largest but also the other eigenvalues. Those eigenvalues converging to non-zero values tend to be in the stable (blue) interval, and all the other converges to zero. More discussion can be found there.
>
> **Thermodynamic limit:** It is relevant to the NTK regime, but the proportional limit with ratio $\\alpha$ is important. Such a proportional limit has been often used to study the generalization error under the double descent (for controlling random matrices). For example, Pennington & Worah (2017) “Nonlinear random matrix theory for deep learning” leverages it.
>
> **Findings about EMA:** The most general analysis of non-contrastive learning with the exponential moving average is challenging, so Tian et al. (2021) chose to approximate it by the _proportional EMA_ (their Assumption 1): representing the target representation net by $\\Phi_a(t) = \tau(t) \\Phi(t)$. While we can do the same analysis with our setup, the proportional ratio $\\tau$ is eventually cancelled out due to the feature normalization.

---

> ### Comment · Reviewer_sECc · 2023-11-23
>
> We thank the authors for their detailed response.
>
> **Previous literature** Indeed, the complete collapse seems possible in their work, which this submission tackles. I may be mistaken but I do not see why the implicit variance regularization is not applicable to this work, since it is due to the $\lambda_m$ term in Eq. 9 of in Halvagal et al., which seems to correspond to the $p_j^2$ term in Eq. 9 in the authors' work, and is not particularly related to the assumption on the input? This may be a misunderstanding of the two works on my part. In any case, I mainly think that the comparison should be discussed in the final version.
>
> **Regimes** Thank you for the clarifications. These points should be made clearer in the final version.
>
> **Experiments / Fig 6c** The description added improves the readability of the Figure. The experiments on the ResNet-18 help ground the results in a more realistic setting. However, the claim that the norms remain stable becomes hard to maintain. Similarly, the curve of asymmetry of $W$ is quite surprising, but is less important for the results. (Small mistake: 'W becomes relatively asymmetry")
>
> **Thermodynamic limit** Thank you for the clarification. The term in itself was a novelty for me, and should maybe be cited from a source using it similarly, such as Pacelli et al., A statistical mechanics framework for Bayesian deep neural networks beyond the infinite-width limit.
>
> The authors have answered some of my concerns and improved their article with some additional experiments and clarifications. Thus, I'm increasing my rating from a 6 to a 8.

---

> > ### Author Response · Authors · 2023-11-23
> > **Response**
> >
> > Thank you so much for the reevaluation and acknowledging our additional work based on your feedback. We certainly feel that our work becomes better after the revision based on your comments. Your additional feedback related to further clarification and typo will be updated.
> >
> > ----
> >
> > Regarding to Halgaval et al., it seems to be better focusing on their Eq. (11) (the cosine loss dynamics) instead of Eq. (9) (the L2 loss dynamics). In their cosine loss dynamics, different non-collapsed eigenvalues $\lambda\_k$ and $\lambda\_m$ approaches at the equilibrium state, which is called the implicit variance regularization by them. In our theoretical analysis, this can hardly be observed because the standard normal input is assumed, as we described; indeed, this assumption is crucial to disentangle the matrix dynamics of our Eq. (5) to the eigenmode dynamics of our Eq. (6). The disentangled eigenmode dynamics are the same for every eigenmode just because the input covariance is isotropic. This simplification has been used since Saxe et al. (2014) "Exact solutions to the nonlinear dynamics of learning in deep linear neural networks." Without the isotropic covariance, the eigenmodes are generally entangled, and we may be able to see a similar observation to Halgaval et al., while analyzing such entangled dynamics is much more intricate. This is why we stated "which would not be elicitable from our analysis because we assume the standard normal input."

---

### Official Review · Reviewer_ZH8r · 2023-11-01

**Soundness:** 3 good
**Presentation:** 3 good
**Contribution:** 2 fair
**Rating:** 5
**Confidence:** 2

**Summary:**

This paper studies the dynamics of non-contrastive self-supervised learning (e.g. BYOL, SimSiam etc.) and shows how feature normalization can play a role in preventing the collapse of all representations to a single point. By studying this in the infinite dimensional limit, the paper shows, that with the cosine loss, the training dynamics are different from that with L2 loss.

**Strengths:**

1. The technical analysis appears rigorous and reasonably clear to follow.

**Weaknesses:**

1. Considering the majority of the analysis assumes norms of all features are nearly same, due to the high dimensional limit, I do not see how this analysis can show the effects of feature normalization on non-contrastive learning.

2. Moreover, the notions of "6-th order dynamics"and "3rd order dynamics" are not sufficiently explained in the paper.

3. Most importantly, I'm not convinced this an interesting problem to study in the context of prior work providing key understanding regarding how non-contrastive SSL training dynamics work.

**Questions:**

1. Considering the majority of the analysis assumes norms of all features are nearly same, due to the high dimensional limit, I do not see how this analysis can show the effects of feature normalization on non-contrastive learning. Can the authors explain why they believe this analysis is showing anything meaningful about feature normalization.

2. Practically, are there any differences in the conclusions of the training dynamics of the cosine loss and the L2 loss? (while they may be of "different orders").

---

> ### Author Response · Authors · 2023-11-19
> **Response**
>
> Thank you for raising important questions to our work! We address them subsequently. We appreciate it if you could rethink the evaluation of our work based on the updates.
>
> **Assumption on the norm constancy:** This assumption is only relevant to disentangle the matrix dynamics (4) into the eigenmode dynamics (6). We do not need the norm constancy globally (for all time $t$) if we are concerned about this disentanglement at each fixed time. Globally, the norm values may of course change; however, the shapes of dynamics at each time shown in Figures 2 and 3 remain qualitatively the same.
>
> **Importance of the problem studied here:** Many recent studies have started to theoretically investigate non-contrastive learning and its implicit bias from the perspective of the L2 loss, which may represent different behaviors from the cosine loss as we show in this work. To elicit more useful insights in the future, we believe that our work can definitely contribute to driving the community into focusing on the cosine loss and providing a cornerstone for the analysis.
>
> **6th order vs. 3rd order dynamics:** As can be seen in the dynamics (8) and (9), they are 6th- and 3rd-order formula in $p\_j$, respectively, which are called the dynamics with respective orders. One of the practical differences is admissible hyperparameter ranges; for example, while regularization (weight decay) cannot be too large for the L2 loss, it is possible for the cosine loss, as confirmed in our pilot study (Figure 1).

---

### Meta-Review · Area_Chair_VzE8 · 2023-12-05

**Metareview:**

The ability of BYOL/SimSiam to learn good representations without negative pairs has been a mystery that attracts many recent interests. Following the seminal work of Tian et al, this work aims to extend their analysis from L2 similarity loss to cosine similarity loss that is actually used in practice. They claim that this difference is very important and worths studying, which, however, is not fully demonstrated yet. Following similar assumptions as in Tian et al’s framework, they analyze the dynamics of parameters, and use the assumptions on thermodynamical limit to make the normalizer easier to deal with. They also utilize an assumption on the stable norm to simplify the formula and arrive at the dynamics of the singular values. They then characterize different learning regimes with a loose examination of the dynamics, and verify some of them on synthetic data.

The main strength of this work is that it performs a rigorous analysis of the cosine similarity version of the loss. However, though the authors seem trying very hard to present the logic, the paper is still highly technical and hard to follow for non-experts. As pointed out by some reviewers, is also unclear what is the significance or new insights of the new derivations for understanding non-contrastive learning methods. The extension from L2 similarity to cosine similarity is not well justified, since the authors do not show that this gap makes a lot of differences to the actual behaviors of non-contrastive learning, especially its collapsing. More critically, as emphasized by Reviewer HYk4, Assumption 5 that assumes the stable norm of features (if so, there is no need for feature normalization) goes against the main goal of this work for analyzing feature normalization, and it seems to play a necessary part in the theoretical results.

Summarizing these opinions, I would recommend the rejection of this paper, although it has some good merits. The authors are encouraged to put more efforts to establish more realistic assumptions, to highlight the influences between different objectives, and to write this paper in a more digestible way (and to put more formal details in the appendix), and submit it to a future venue.

**Justification For Why Not Higher Score:**

This paper lacks enough significance for a better understanding of non-contrastive learning and it relies on strong assumptions that go against the primary interest of this subject.

**Justification For Why Not Lower Score:**

N/A

---

### Decision · Program_Chairs · 2024-01-16

Reject